# Energy-based Hopfield Boosting for Out-of-Distribution Detection

**Claus Hofmann** [1]    **Simon Schmid** [2]    **Bernhard Lehner** [3]

**Daniel Klotz** [4]    **Sepp Hochreiter** [1]

[1] Institute for Machine Learning, JKU LIT SAL IWS Lab,
Johannes Kepler University, Linz, Austria
[2] Software Competence Center Hagenberg GmbH, Austria
[3] Silicon Austria Labs, JKU LIT SAL IWS Lab, Linz, Austria
[4] Department of Computational Hydrosystems,
Helmholtz Centre for Environmental Research–UFZ, Leipzig, Germany

`hofmann@ml.jku.at`

## Abstract

Out-of-distribution (OOD) detection is critical when deploying machine learning models in the real world. Outlier exposure methods, which incorporate auxiliary outlier data in the training process, can drastically improve OOD detection performance compared to approaches without advanced training strategies. We introduce Hopfield Boosting, a boosting approach, which leverages modern Hopfield energy to sharpen the decision boundary between the in-distribution and OOD data. Hopfield Boosting encourages the model to focus on hard-to-distinguish auxiliary outlier examples that lie close to the decision boundary between in-distribution and auxiliary outlier data. Our method achieves a new state-of-the-art in OOD detection with outlier exposure, improving the FPR95 from 2.28 to 0.92 on CIFAR-10, from 11.76 to 7.94 on CIFAR-100, and from 50.74 to 36.60 on ImageNet-1K.

## 1 Introduction

Out-of-distribution (OOD) detection is crucial when using machine learning systems in the real world (Ruff et al., 2021; Yang et al., 2021; Liu et al., 2021). Deployed models will — sooner or later — encounter inputs that deviate from the training distribution. For example, a system trained to recognize music genres might also hear a sound clip of construction site noise. In the best case, a naive deployment can then result in overly confident predictions. In the worst case, we will get erratic model behavior and completely wrong predictions (Hendrycks & Gimpel, 2017). The purpose of OOD detection is to classify these inputs as OOD, such that the system can then, for instance, notify users that no prediction is possible. In this paper we propose Hopfield Boosting, a novel OOD detection method that leverages the energy component of modern Hopfield networks (MHNs; Ramsauer et al., 2021) and advances the state-of-the-art of OOD detection. This energy represents a measure of dissimilarity between a set of data instances $X$ and a query instance $\xi$. It is therefore a natural fit for doing OOD detection (as shown in Zhang et al., 2023a).

Hopfield Boosting uses an auxiliary outlier data set (AUX) to *boost* the model's OOD detection capacity. This allows the training process to learn a boundary around the in-distribution (ID) data, improving the performance at the OOD detection task. In summary, our contributions are as follows:

38th Conference on Neural Information Processing Systems (NeurIPS 2024).

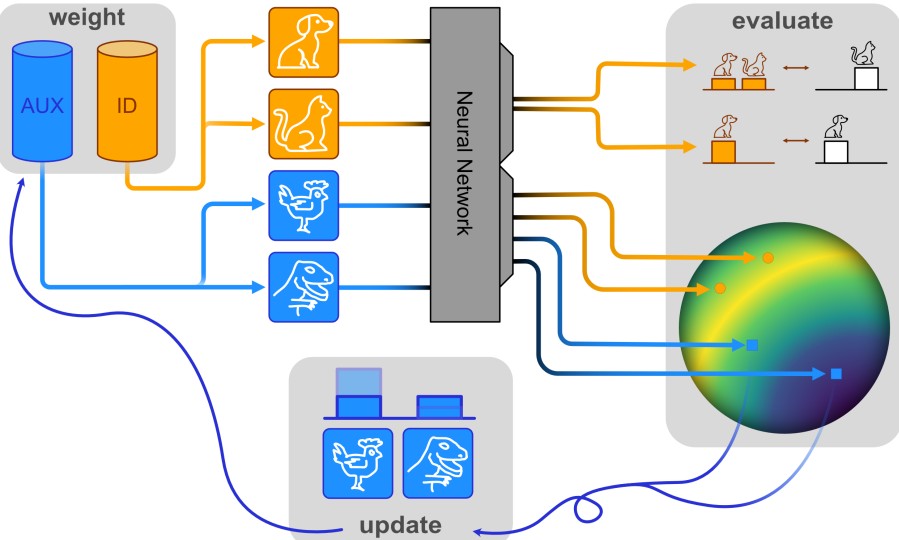

Figure 1: The Hopfield Boosting concept. The first step (weight) creates weak learners by firstly choosing in-distribution samples (ID, orange), and by secondly choosing auxiliary outlier samples (AUX, blue) according to their assigned probabilities; the second step (evaluate) computes the losses for the resulting predictions (Section 3); and the third step (update) assigns new probabilities to the AUX samples according to their position on the hypersphere (see Figure 2).

1. We propose Hopfield Boosting, an OOD detection approach that samples weak learners by using the MHE (Ramsauer et al., 2021).

2. Hopfield Boosting achieves a new state-of-the-art in OOD detection. It improves the average false positive rate at 95% true positives (FPR95) from 2.28 to 0.92 on CIFAR-10, from 11.38 to 7.94 on CIFAR-100, and from 50.74 to 36.60 on ImageNet-1K.

3. We provide theoretical background that motivates Hopfield Boosting for OOD detection.

## 2    Related Work

Some authors (e.g., Bishop, 1994; Roth et al., 2022; Yang et al., 2022) distinguish between anomalies, outliers, and novelties. These distinctions reflect different goals within applications (Ruff et al., 2021). For example, when an anomaly is found, it will usually be removed from the training pipeline. However, when a novelty is found it should be studied. We focus on the detection of samples that are not part of the training distribution and consider sample categorization as a downstream task.

**Post-hoc OOD detection.**    A common and straightforward OOD detection approach is to use a post-hoc strategy, where one employs statistics obtained from a classifier. The perhaps most well known and simplest approach in this class is the Maximum Softmax Probability (MSP; Hendrycks & Gimpel, 2017), where one utilizes $p(\,y \mid \boldsymbol{x}\,)$ of the most likely class $y$ given a feature vector $\boldsymbol{x} \in \mathbb{R}^D$ to estimate whether a sample is OOD. Despite good empirical performances, this view is intrinsically limited, since OOD detection should focus on $p(\boldsymbol{x})$ (Morteza & Li, 2022). A wide range of post-hoc OOD detection approaches have been proposed to address the shortcomings of MSP (e.g., Lee et al., 2018; Hendrycks et al., 2019a; Liu et al., 2020; Sun et al., 2021, 2022; Wang et al., 2022; Zhang et al., 2023a; Djurisic et al., 2023; Liu et al., 2023; Xu et al., 2024). Most related to Hopfield Boosting is the work of Zhang et al. (2023a) — to our knowledge, they are the first to apply the MHE to OOD detection. Specifically, they use the ID data set to produce stored patterns and then use a modified version of MHE as the OOD score. While post-hoc approaches can be deployed out of the box on any model, a crucial limitation is that their performance heavily depends on the employed model itself.

**Training methods.** In contrast to post-hoc strategies, training-based methods modify the training process to improve the model's OOD detection capability (e.g., Hendrycks et al., 2019c; Tack et al., 2020; Sehwag et al., 2021; Du et al., 2022; Hendrycks et al., 2022; Wei et al., 2022a; Ming et al., 2023; Tao et al., 2023; Lu et al., 2024). For example, Self-Supervised Outlier Detection (SSD; Sehwag et al., 2021) leverages contrastive self-supervised learning to train a model for OOD detection.

**Auxiliary outlier data and outlier exposure.** A third group of OOD detection approaches are outlier exposure (OE) methods. Like Hopfield Boosting, they incorporate AUX data in the training process to improve the detection of OOD samples (e.g., Hendrycks et al., 2019b; Liu et al., 2020; Ming et al., 2022; Zhang et al., 2023b; Wang et al., 2023a; Zhu et al., 2023; Jiang et al., 2024). We provide more detailed discussions on a range of OE methods in Appendix C.1. As far as we know, all OE approaches optimize an objective ($\mathcal{L}_{\text{OOD}}$), which aims at improving the model's discriminative power between ID and OOD data using the AUX data set as a stand-in for the OOD case. Hendrycks et al. (2019b) were the first to use the term OE to describe a more restrictive OE concept. Since their approach uses the MSP for incorporating the AUX data we refer to it as MSP-OE. Further, we refer to the OE approach introduced in Liu et al. (2020) as EBO-OE (to differentiate it from EBO, their post-hoc approach). In general, OE methods conceptualize the AUX data set as a large and diverse data set (e.g., ImageNet for vision tasks). As a consequence, usually, only a small subset of the samples bear semantic similarity to the ID data set — most data points are easily distinguishable from the ID data. Recent approaches therefore actively try to find informative samples for the training. The aim is to refine the decision boundary, ensuring the ID data is more tightly encapsulated (e.g., Chen et al., 2021; Ming et al., 2022). For example, Posterior Sampling-based Outlier Mining (POEM; Ming et al., 2022) selects samples close to the decision boundary using Thompson sampling: They first sample a linear decision boundary between ID and AUX data and then select those data instances which are closest to the sampled decision boundary. Hopfield Boosting also makes use of samples close to the boundary by giving them higher weights for the boosting step.

**Continuous modern Hopfield networks.** MHNs are energy-based associative memory networks. They advance conventional Hopfield networks (Hopfield, 1984) by introducing continuous queries and states with the MHE as a new energy function. MHE leads to exponential storage capacity, while retrieval is possible with a one-step update (Ramsauer et al., 2021). The update rule of MHNs coincides with attention as it is used in the Transformer (Vaswani et al., 2017). Examples for successful applications of MHNs are Widrich et al. (2020); Fürst et al. (2022); Sanchez-Fernandez et al. (2022); Paischer et al. (2022); Schäfl et al. (2022); Schimunek et al. (2023) and Auer et al. (2023). Section 3.2 gives an introduction to MHE for OOD detection. For further details on MHNs, we refer to Appendix A.

**Boosting for classification.** Boosting, in particular, AdaBoost (Freund & Schapire, 1995), is an ensemble learning technique for classification. It is designed to focus ensemble members toward data instances that are hard to classify by assigning them higher weights. These challenging instances often lie near the maximum margin hyperplane (Rätsch et al., 2001), akin to support vectors in support vector machines (SVMs; Cortes & Vapnik, 1995). Popular boosting methods include Gradient boosting (Breiman, 1997), LogitBoost (Friedman et al., 2000), and LPBoost (Demiriz et al., 2002).

**Radial basis function networks.** Radial basis function networks (RBF networks; Moody & Darken, 1989) are function approximators of the form

$$\varphi(\boldsymbol{\xi}) = \sum_{i=1}^{N} \omega_i \exp\left(-\frac{||\boldsymbol{\xi} - \boldsymbol{\mu}_i||_2^2}{2\sigma_i^2}\right), \tag{1}$$

where $\omega_i$ are linear weights, $\boldsymbol{\mu}_i$ are the component means and $\sigma_i^2$ are the component variances. RBF networks can be described as a weighted linear superposition of $N$ radial basis functions and have previously been used as hypotheses for boosting (Rätsch et al., 2001). If the linear weights are strictly positive, RBF networks can be viewed as an unnormalized weighted mixture of Gaussian distributions $p_i(\boldsymbol{\xi}) = \mathcal{N}(\boldsymbol{\xi}; \boldsymbol{\mu}_i, \sigma_i^2 \boldsymbol{I})$ with $i = \{1, \ldots, N\}$. Appendix H.1 explores the connection between RBF networks and MHNs via Gaussian mixtures in more depth. We refer to Bishop (1995) and Müller et al. (1997) for more general information on RBF networks.

## 3 Method

This section presents Hopfield Boosting: First, we formalize the OOD detection task. Second, we give an overview of the MHE and why it is suitable for OOD detection. Finally, we introduce the AUX-based boosting framework. Figure 1 shows a summary of the Hopfield Boosting concept.

### 3.1 Classification and OOD Detection

Consider a multi-class classification task denoted as $(\boldsymbol{X}^{\mathcal{D}}, \boldsymbol{Y}^{\mathcal{D}}, \mathcal{Y})$, where $\boldsymbol{X}^{\mathcal{D}} \in \mathbb{R}^{D \times N}$ represents a set of $N$ $D$-dimensional feature vectors $(\boldsymbol{x}_1^{\mathcal{D}}, \boldsymbol{x}_2^{\mathcal{D}}, \ldots, \boldsymbol{x}_N^{\mathcal{D}})$, which are i.i.d. samples $\boldsymbol{x}_i^{\mathcal{D}} \sim p_{\text{ID}}$. $\boldsymbol{Y}^{\mathcal{D}} \in \mathcal{Y}^N$ denotes the labels associated with these feature vectors, and $\mathcal{Y}$ is a set containing possible classes ($\|\mathcal{Y}\| = K$ signifies the number of distinct classes). We consider observations $\boldsymbol{\xi}^{\mathcal{D}} \in \mathbb{R}^D$ that deviate considerably from the data generation $p_{\text{ID}}(\boldsymbol{\xi}^{\mathcal{D}})$ that defines the "normality" of our data as OOD. Following Ruff et al. (2021), an observation is OOD if it pertains to the set

$$\mathbb{O} = \{\boldsymbol{\xi}^{\mathcal{D}} \in \mathbb{R}^D \mid p_{\text{ID}}(\boldsymbol{\xi}^{\mathcal{D}}) < \epsilon\} \text{ where } \epsilon \geq 0. \tag{2}$$

Since the probability density of the data generation $p_{\text{ID}}$ is in general not known, one needs to estimate $p_{\text{ID}}(\boldsymbol{\xi}^{\mathcal{D}})$. In practice, it is common to define an outlier score $s(\boldsymbol{\xi})$ that uses an encoder $\phi$, where $\boldsymbol{\xi} = \phi(\boldsymbol{\xi}^{\mathcal{D}})$. The outlier score should — in the best case — preserve the density ranking. In contrast to a density estimation, the score $s(\boldsymbol{\xi})$ does not have to fulfill all requirements of a probability density (like proper normalization or non-negativity). Given $s(\boldsymbol{\xi})$ and $\phi$, OOD detection can be formulated as a binary classification task with the classes ID and OOD:

$$\hat{B}(\boldsymbol{\xi}^{\mathcal{D}}, \gamma) = \begin{cases} \text{ID} & \text{if } s(\phi(\boldsymbol{\xi}^{\mathcal{D}})) \geq \gamma \\ \text{OOD} & \text{if } s(\phi(\boldsymbol{\xi}^{\mathcal{D}})) < \gamma \end{cases}. \tag{3}$$

It is common to choose the threshold $\gamma$ so that a portion of 95% of ID samples from a previously unseen validation set are correctly classified as ID. However, metrics like the area under the receiver operating characteristic (AUROC) can be directly computed on $s(\boldsymbol{\xi})$ without specifying $\gamma$ since the AUROC computation sweeps over the threshold.

### 3.2 Modern Hopfield Energy

The log-sum-exponential (lse) function is defined as

$$\text{lse}(\beta, \boldsymbol{z}) = \beta^{-1} \log \left( \sum_{i=1}^{N} \exp(\beta z_i) \right), \tag{4}$$

where $\beta$ is the inverse temperature and $\boldsymbol{z} \in \mathbb{R}^N$ is a vector. The lse can be seen as a soft approximation to the maximum function: As $\beta \to \infty$, the lse approaches $\max_i z_i$.

Given a set of $N$ $d$-dimensional stored patterns $(\boldsymbol{x}_1, \boldsymbol{x}_2, \ldots, \boldsymbol{x}_N)$ arranged in a data matrix $\boldsymbol{X}$, and a $d$-dimensional query $\boldsymbol{\xi}$, the MHE is defined as

$$E(\boldsymbol{\xi}; \boldsymbol{X}) = -\text{lse}(\beta, \boldsymbol{X}^T \boldsymbol{\xi}) + \frac{1}{2} \boldsymbol{\xi}^T \boldsymbol{\xi} + C, \tag{5}$$

where $C = \beta^{-1} \log N + \frac{1}{2} M^2$ and $M$ is the largest norm of a pattern: $M = \max_i \|x_i\|$. $\boldsymbol{X}$ is also called the memory of the MHN. Intuitively, Equation (5) can be explained as follows: The dot-product within the lse computes a similarity for a given $\boldsymbol{\xi} \in \mathbb{R}^d$ to all patterns in the memory $\boldsymbol{X} \in \mathbb{R}^{d \times N}$. The lse function aggregates the similarities to form a single value, where the $\beta$ parameterizes the aggregation operation: If $\beta \to \infty$, the maximum similarity of $\boldsymbol{\xi}$ to the patterns in $\boldsymbol{X}$ is returned.

To use the MHE for OOD detection, Hopfield Boosting acquires the memory patterns $\boldsymbol{X}$ by feeding raw data instances $(\boldsymbol{x}_1^{\mathcal{D}}, \boldsymbol{x}_2^{\mathcal{D}}, \ldots, \boldsymbol{x}_N^{\mathcal{D}})$ of the ID data set arranged in the data matrix $\boldsymbol{X}^{\mathcal{D}} \in \mathbb{R}^{D \times N}$ to an encoder $\phi : \mathbb{R}^D \to \mathbb{R}^d$ (e.g., ResNet): $\boldsymbol{x}_i = \phi(\boldsymbol{x}_i^{\mathcal{D}})$. We denote the component-wise application of $\phi$ to the patterns in $\boldsymbol{X}^{\mathcal{D}}$ as $\boldsymbol{X} = \phi(\boldsymbol{X}^{\mathcal{D}})$. Similarly, a raw query $\boldsymbol{\xi}^{\mathcal{D}} \in \mathbb{R}^D$ is fed through the encoder to obtain the query pattern: $\boldsymbol{\xi} = \phi(\boldsymbol{\xi}^{\mathcal{D}})$. One can now use $E(\boldsymbol{\xi}; \boldsymbol{X})$ to estimate whether $\boldsymbol{\xi}$ is ID or OOD: A low energy indicates $\boldsymbol{\xi}$ is ID, and a high energy signifies that $\boldsymbol{\xi}$ is OOD.

### 3.3 Boosting Framework

**Sampling of informative outlier data.** Hopfield Boosting uses AUX data to learn a decision boundary between the ID and OOD region during the training. Similar to Chen et al. (2021) and Ming et al. (2022), Hopfield Boosting selects informative outliers close to the ID-OOD decision boundary. For this selection, Hopfield Boosting weights the AUX data similar to AdaBoost (Freund & Schapire, 1995) by sampling data instances close to the decision boundary more frequently. We consider samples close to the decision boundary as weak learners — their nearest neighbors consist of samples from their own class as well as from the foreign class. An individual weak learner represents a classifier that is only slightly better than random guessing (Figure 6). Vice versa, a strong learner can be created by forming an ensemble of a set of weak learners (Figure 2).

We denote the matrix containing the raw AUX data instances $(\boldsymbol{o}_1^{\mathcal{D}}, \boldsymbol{o}_2^{\mathcal{D}}, \ldots, \boldsymbol{o}_N^{\mathcal{D}})$ as $\boldsymbol{O}^{\mathcal{D}} \in \mathbb{R}^{D \times M}$, and the memory containing the encoded AUX patterns as $\boldsymbol{O} = \phi(\boldsymbol{O}^{\mathcal{D}})$. The boosting process proceeds as follows: There exists a weight $(w_1, w_2, \ldots, w_N)$ for each data point in $\boldsymbol{O}^{\mathcal{D}}$ and the individual weights are aggregated into the weight vector $\boldsymbol{w}_t$. Hopfield Boosting uses these weights to draw mini-batches $\boldsymbol{O}_s^{\mathcal{D}}$ from $\boldsymbol{O}^{\mathcal{D}}$ for training, where weak learners are sampled more frequently.

We introduce an MHE-based energy function which Hopfield Boosting uses to determine how weak a specific learner $\boldsymbol{\xi}$ is (with higher energy indicating a weaker learner):

$$\mathrm{E}_b(\boldsymbol{\xi}; \boldsymbol{X}, \boldsymbol{O}) = -2\,\mathrm{lse}(\beta, (\boldsymbol{X} \| \boldsymbol{O})^T \boldsymbol{\xi}) + \mathrm{lse}(\beta, \boldsymbol{X}^T \boldsymbol{\xi}) + \mathrm{lse}(\beta, \boldsymbol{O}^T \boldsymbol{\xi}), \tag{6}$$

where $\boldsymbol{X} \in \mathbb{R}^{d \times N}$ contains ID patterns, $\boldsymbol{O} \in \mathbb{R}^{d \times M}$ contains AUX patterns, and $(\boldsymbol{X} \| \boldsymbol{O}) \in \mathbb{R}^{d \times (N+M)}$ denotes the concatenated data matrix containing the patterns from both $\boldsymbol{X}$ and $\boldsymbol{O}$. Before computing $\mathrm{E}_b$, we normalize the feature vectors in $\boldsymbol{X}$, $\boldsymbol{O}$, and $\boldsymbol{\xi}$ to unit length. Figure 3 displays the energy landscape of $\mathrm{E}_b(\boldsymbol{\xi}; \boldsymbol{X}, \boldsymbol{O})$ using exemplary data on a 3-dimensional sphere. $\mathrm{E}_b$ is maximal at the decision boundary between ID and AUX data and decreases with increasing distance from the decision boundary in both directions.

As we show in our theoretical discussion in Appendix G, when modeling the class-conditional densities of the ID and AUX data set as mixtures of Gaussian distributions

$$p(\boldsymbol{\xi} \mid \mathrm{ID}) = \frac{1}{N} \sum_{i=1}^{N} \mathcal{N}(\boldsymbol{\xi}; \boldsymbol{x}_i, \beta^{-1} \boldsymbol{I}), \tag{7}$$

$$p(\boldsymbol{\xi} \mid \mathrm{AUX}) = \frac{1}{N} \sum_{i=1}^{N} \mathcal{N}(\boldsymbol{\xi}; \boldsymbol{o}_i, \beta^{-1} \boldsymbol{I}), \tag{8}$$

with equal class priors $p(\mathrm{ID}) = p(\mathrm{AUX}) = 1/2$ and normalized patterns $||\boldsymbol{x}_i|| = 1$ and $||\boldsymbol{o}_i|| = 1$, we obtain $\mathrm{E}_b(\boldsymbol{\xi}; \boldsymbol{X}, \boldsymbol{O}) \overset{C}{=} \beta^{-1} \log(p(\mathrm{ID} \mid \boldsymbol{\xi}) \cdot p(\mathrm{AUX} \mid \boldsymbol{\xi}))$, where $\overset{C}{=}$ denotes equality up to an irrelevant additive constant. The exponential of $\mathrm{E}_b$ is the variance of a Bernoulli random variable with the outcomes $\{\mathrm{ID}, \mathrm{AUX}\}$ conditioned on $\boldsymbol{\xi}$. Thus, according to $\mathrm{E}_b$, the weak learners are situated at locations where the model defined in Equations (7) and (8) is uncertain.

Given a set of query values $(\boldsymbol{\xi}_1, \boldsymbol{\xi}_2, \ldots, \boldsymbol{\xi}_n)$ assembled in a query matrix $\boldsymbol{\Xi} \in \mathbb{R}^{d \times n}$, we denote a vector of energies $\boldsymbol{e} \in \mathbb{R}^n$ with $e_i = \mathrm{E}_b(\boldsymbol{\xi}_i; \boldsymbol{X}, \boldsymbol{O})$ as

$$\boldsymbol{e} = \mathrm{E}_b(\boldsymbol{\Xi}; \boldsymbol{X}, \boldsymbol{O}). \tag{9}$$

To calculate the weights $\boldsymbol{w}_{t+1}$, we use the memory of AUX patterns as a query matrix $\boldsymbol{\Xi} = \boldsymbol{O}$ and compute the respective energies $\mathrm{E}_b$ of those patterns. The resulting energy vector $\mathrm{E}_b(\boldsymbol{\Xi}; \boldsymbol{X}, \boldsymbol{O})$ is then normalized by a $\mathrm{softmax}$. This computation provides the updated weights:

$$\boldsymbol{w}_{t+1} = \mathrm{softmax}(\beta \mathrm{E}_b(\boldsymbol{\Xi}; \boldsymbol{X}, \boldsymbol{O})). \tag{10}$$

Appendix J provides theoretical background on how informative samples close to the decision boundary are beneficial for training an OOD detector.

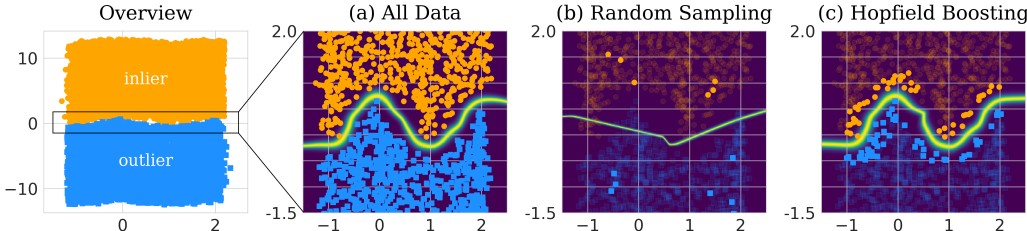

Figure 2: Synthetic example of the adaptive resampling mechanism. Hopfield Boosting forms a strong learner by sampling and combining a set of weak learners close to the decision boundary. The heatmap on the background shows $\exp(\beta \mathrm{E}_b(\boldsymbol{\xi}; \boldsymbol{X}, \boldsymbol{O}))$, where $\beta$ is 60. Only the sampled (i.e., highlighted) points serve as memories $\boldsymbol{X}$ and $\boldsymbol{O}$.

**Training the model with MHE.**   In this section, we introduce how Hopfield Boosting uses the sampled weak learners to improve the detection of patterns outside the training distribution. We follow the established training method for OE (Hendrycks et al., 2019b; Liu et al., 2020; Ming et al., 2022): Train a classifier on the ID data using the standard cross-entropy loss and add an OOD loss that uses the AUX data set to sharpen the decision boundary between the ID and OOD regions. Formally, this yields the loss

$$\mathcal{L} = \mathcal{L}_{\mathrm{CE}} + \lambda \mathcal{L}_{\mathrm{OOD}}, \tag{11}$$

where $\lambda$ is a hyperparamter indicating the relative importance of $\mathcal{L}_{\mathrm{OOD}}$. Hopfield Boosting explicitly minimizes $\mathrm{E}_b$ (which is also the energy function Hopfield Boosting uses to sample weak learners). Given the weight vector $\boldsymbol{w}_t$, and the data sets $\boldsymbol{X}^{\mathcal{D}}$ and $\boldsymbol{O}^{\mathcal{D}}$, we obtain a mini-batch $\boldsymbol{X}_s^{\mathcal{D}}$ containing N samples from $\boldsymbol{X}^{\mathcal{D}}$ by uniform sampling, and a mini-batch of N weak learners $\boldsymbol{O}_s^{\mathcal{D}}$ from $\boldsymbol{O}^{\mathcal{D}}$ by sampling according to $\boldsymbol{w}_t$ with replacement. We then feed the respective mini-batches into the neural network $\phi_{\mathrm{base}}$ to create a latent feature (in our experiments, we always use the feature of the penultimate layer of a ResNet). Our proposed approach then uses two heads:

1. A linear classification head that maps the latent feature to the class logits for $\mathcal{L}_{\mathrm{CE}}$.
2. A 2-layer MLP $\phi_{\mathrm{proj}}$ maps the features from the penultimate layer to the output for $\mathcal{L}_{\mathrm{OOD}}$.

Hopfield Boosting computes $\mathcal{L}_{\mathrm{OOD}}$ on the representations it obtains from $\phi = \phi_{\mathrm{proj}} \circ \phi_{\mathrm{base}}$ as follows:

$$\mathcal{L}_{\mathrm{OOD}} = \frac{1}{2N} \sum_{\boldsymbol{\xi}} \mathrm{E}_b(\boldsymbol{\xi}; \boldsymbol{X}_s, \boldsymbol{O}_s), \tag{12}$$

where the memories $\boldsymbol{X}_s$ and $\boldsymbol{O}_s$ contain the encodings of the sampled data instances: $\boldsymbol{X}_s = \phi(\boldsymbol{X}_s^{\mathcal{D}})$ and $\boldsymbol{O}_s = \phi(\boldsymbol{O}_s^{\mathcal{D}})$. The sum is taken over the observations $\boldsymbol{\xi}$, which are drawn from $(\boldsymbol{X}_s \parallel \boldsymbol{O}_s)$. Hopfield Boosting computes $\mathcal{L}_{\mathrm{OOD}}$ for each mini-batch by first calculating the pairwise similarity matrix between the patterns in the mini-batch, followed by determining the $\mathrm{E}_b$ values of the individual observations $\boldsymbol{\xi}$, and, finally a mean reduction. To the best of our knowledge, Hopfield Boosting is the first method that uses Hopfield networks in this way to train a deep neural network. We note that there is a relation between Hopfield Boosting and SVMs with an RBF kernel (see Appendix H.3). However, the optimization procedure of SVMs is in general not differentiable. In contrast, our novel energy function is fully differentiable. This allows us to use it to train neural networks.

**Summary.**   Algorithm 1 provides an outline of Hopfield Boosting. Each iteration $t$ consists of three main steps: 1. weight, 2. evaluate, and 3. update. First, Hopfield Boosting samples a mini-batch from the ID data and **weights** the AUX data by sampling a mini-batch according to $\boldsymbol{w}_t$. Second, Hopfield Boosting **evaluates** the composite loss on the sampled mini-batch. Third, Hopfield Boosting **updates** the model parameters and, every $N$-th step, also the sampling weights for the AUX data set $\boldsymbol{w}_{t+1}$.

**Inference.**   At inference time, the OOD score $s(\boldsymbol{\xi})$ is

$$s(\boldsymbol{\xi}) = \mathrm{lse}(\beta, \boldsymbol{X}^T \boldsymbol{\xi}) - \mathrm{lse}(\beta, \boldsymbol{O}^T \boldsymbol{\xi}). \tag{13}$$

---
**Algorithm 1** Hopfield Boosting
---
**Require:** $T, N, \boldsymbol{X}, \boldsymbol{O}, \boldsymbol{Y}, \mathcal{L}_{\text{CE}}, \text{E}_b, \beta$
  Set all weights $w_1$ to $1/|\boldsymbol{O}|$
  **for** $t = 1$ to $T$ **do**
     1. **Weight**. Get hypothesis $\boldsymbol{X}_s \parallel \boldsymbol{O}_s \rightarrow \{\text{ID, AUX}\}$:
       1.a. Mini-batch sampling $\boldsymbol{X}_s$ from $\boldsymbol{X}$, and
       1.b. Sub-sampling of weak learners $\boldsymbol{O}_s$ from $\boldsymbol{O}$ according to the weighting $\boldsymbol{w}_t$.
     2. **Evaluate**. Compute loss from Equation (11) on $\boldsymbol{X}_s$ and $\boldsymbol{O}_s$.
     3. **Update**. Update model for the next iteration:
       3.a. At every step, update the full model (backbone, classification head, and MHE).
       3.b. At every $t * N$ step calculate new weights for $\boldsymbol{O}$ with $\boldsymbol{w}_{t+1} = \text{softmax}(\beta \text{E}_b(\boldsymbol{O}; \boldsymbol{X}, \boldsymbol{O}))$.
  **end for**
---

For computing $s(\boldsymbol{\xi})$, Hopfield Boosting uses the 50,000 random samples from the ID and AUX data sets, respectively. As we show in Appendix I.8, this step entails only a very moderate computational overhead in relation to a complete forward pass (e.g., an overhead of 7.5% for ResNet-18 on an NVIDIA Titan V GPU with 50,000 patterns stored in each of the memories $\boldsymbol{X}$ and $\boldsymbol{O}$). We additionally experimented with using only $\text{lse}(\beta, \boldsymbol{X}^T \boldsymbol{\xi})$ as a score, which also gives reasonable results. However, the approach in Equation (13) has turned out to be superior. Equation (13) uses information from both ID and AUX samples. This can, for example, be beneficial for handling query patterns $\boldsymbol{\xi}$ that are dissimilar from both the memory patterns in $\boldsymbol{X}$ as well as from the memory patterns in $\boldsymbol{O}$.

### 3.4 Comparison of Hopfield Boosting to HE and SHE

Zhang et al. (2023a) propose two post-hoc methods for OOD detection with MHE:"Hopfield Energy" (HE) and "Simplified Hopfield Energy" (SHE). In contrast to Hopfield Boosting, HE and SHE do not use AUX data to get a better boundary between ID and OOD data. Rather, their methods evaluate the MHE on ID patterns only to determine whether a sample is ID or OOD. Additional differences include the selection of patterns stored in the memory or the normalization of the patterns. The OE process of Hopfield Boosting drastically improves the OOD detection performance compared to HE and SHE. We verify that the unique contributions of Hopfield Boosting (like the energy-based loss and the boosting process) are responsible for the superior performance with two extensions of HE that include AUX data (the comparison can be found in Appendix I.9). For further information on the differences to HE and SHE, we refer to Appendix H.4.

## 4 Experiments

### 4.1 Toy Example

This section presents a toy example illustrating the main intuitions behind Hopfield Boosting. For the sake of clarity, the toy example does not consider the inlier classification task that would induce secondary processes, which would obscure the explanations. Formally, we do not consider the first term on the right-hand side of Equation (11). For further toy examples, we refer to Appendix F.

Figure 2 demonstrates how the weighting in Hopfield Boosting allows good estimations of the decision boundary, even if Hopfield Boosting only samples a small number of weak learners. This is advantageous because the AUX data set contains a large number of data instances that are uninformative for the OOD detection task. For small, low dimensional data, one can always use all the data to compute $\text{E}_b$ (Figure 2, a). For large problems (like in Ming et al., 2022), this strategy is difficult, and the naive solution of uniformly sampling N data points would also not work. This will yield many uninformative points (Figure 2, b). When using Hopfield Boosting and sampling N weak learners according to $\boldsymbol{w}_t$, the result better approximates the decision boundary of the full data (Figure 2, c).

Table 1: OOD detection performance on CIFAR-10. We compare results from Hopfield Boosting, DOS (Jiang et al., 2024), DOE (Wang et al., 2023b), DivOE (Zhu et al., 2023), DAL (Wang et al., 2023a), MixOE (Zhang et al., 2023b), POEM (Ming et al., 2022), EBO-OE (Liu et al., 2020), and MSP-OE (Hendrycks et al., 2019b) on ResNet-18. ↓ indicates "lower is better" and ↑ "higher is better". All values in %. Standard deviations are estimated across five training runs.

| | Metric | HB (ours) | DOS | DOE | DivOE | DAL | MixOE | POEM | EBO-OE | MSP-OE |
|---|---|---|---|---|---|---|---|---|---|---|
| SVHN | FPR95 ↓ | $\mathbf{0.23^{\pm 0.08}}$ | $3.09^{\pm 0.75}$ | $1.97^{\pm 0.58}$ | $6.21^{\pm 0.84}$ | $1.25^{\pm 0.62}$ | $27.54^{\pm 2.46}$ | $1.48^{\pm 0.68}$ | $2.66^{\pm 0.91}$ | $4.31^{\pm 1.10}$ |
| | AUROC ↑ | $\mathbf{99.57^{\pm 0.06}}$ | $99.15^{\pm 0.22}$ | $\mathbf{99.60^{\pm 0.13}}$ | $98.53^{\pm 0.08}$ | $\mathbf{99.61^{\pm 0.15}}$ | $95.37^{\pm 0.44}$ | $99.33^{\pm 0.15}$ | $99.15^{\pm 0.23}$ | $99.20^{\pm 0.15}$ |
| LSUN-Crop | FPR95 ↓ | $0.82^{\pm 0.17}$ | $3.66^{\pm 0.98}$ | $3.22^{\pm 0.45}$ | $1.88^{\pm 0.25}$ | $4.17^{\pm 0.27}$ | $\mathbf{0.14^{\pm 0.07}}$ | $4.02^{\pm 0.91}$ | $6.82^{\pm 0.74}$ | $7.02^{\pm 1.14}$ |
| | AUROC ↑ | $99.40^{\pm 0.04}$ | $99.04^{\pm 0.20}$ | $99.30^{\pm 0.12}$ | $99.50^{\pm 0.02}$ | $99.13^{\pm 0.02}$ | $\mathbf{99.61^{\pm 0.11}}$ | $98.89^{\pm 0.15}$ | $98.43^{\pm 0.10}$ | $98.83^{\pm 0.15}$ |
| LSUN-Resize | FPR95 ↓ | $\mathbf{0.00^{\pm 0.00}}$ | $0.00^{\pm 0.00}$ | $0.00^{\pm 0.00}$ | $0.00^{\pm 0.00}$ | $0.00^{\pm 0.00}$ | $0.16^{\pm 0.17}$ | $0.00^{\pm 0.00}$ | $0.00^{\pm 0.00}$ | $0.00^{\pm 0.00}$ |
| | AUROC ↑ | $99.98^{\pm 0.02}$ | $99.99^{\pm 0.01}$ | $\mathbf{100.00^{\pm 0.00}}$ | $99.89^{\pm 0.05}$ | $99.92^{\pm 0.05}$ | $99.89^{\pm 0.06}$ | $99.88^{\pm 0.12}$ | $99.98^{\pm 0.02}$ | $99.96^{\pm 0.00}$ |
| Textures | FPR95 ↓ | $\mathbf{0.16^{\pm 0.02}}$ | $1.28^{\pm 0.20}$ | $2.75^{\pm 0.57}$ | $1.20^{\pm 0.11}$ | $0.95^{\pm 0.13}$ | $4.68^{\pm 0.22}$ | $0.49^{\pm 0.04}$ | $1.11^{\pm 0.17}$ | $2.29^{\pm 0.16}$ |
| | AUROC ↑ | $\mathbf{99.84^{\pm 0.01}}$ | $99.63^{\pm 0.04}$ | $99.35^{\pm 0.12}$ | $99.59^{\pm 0.05}$ | $99.74^{\pm 0.01}$ | $98.91^{\pm 0.07}$ | $99.72^{\pm 0.05}$ | $99.61^{\pm 0.02}$ | $99.57^{\pm 0.01}$ |
| iSUN | FPR95 ↓ | $\mathbf{0.00^{\pm 0.00}}$ | $0.00^{\pm 0.00}$ | $0.00^{\pm 0.00}$ | $0.00^{\pm 0.00}$ | $0.00^{\pm 0.00}$ | $0.17^{\pm 0.12}$ | $0.00^{\pm 0.00}$ | $0.00^{\pm 0.00}$ | $0.00^{\pm 0.00}$ |
| | AUROC ↑ | $99.97^{\pm 0.02}$ | $99.99^{\pm 0.01}$ | $\mathbf{100.00^{\pm 0.00}}$ | $99.88^{\pm 0.05}$ | $99.93^{\pm 0.04}$ | $99.87^{\pm 0.05}$ | $99.87^{\pm 0.12}$ | $99.98^{\pm 0.01}$ | $99.96^{\pm 0.00}$ |
| Places 365 | FPR95 ↓ | $\mathbf{4.28^{\pm 0.23}}$ | $12.26^{\pm 0.97}$ | $19.72^{\pm 2.39}$ | $13.70^{\pm 0.50}$ | $14.22^{\pm 0.51}$ | $16.30^{\pm 1.09}$ | $7.70^{\pm 0.68}$ | $11.77^{\pm 0.68}$ | $21.42^{\pm 0.88}$ |
| | AUROC ↑ | $\mathbf{98.51^{\pm 0.10}}$ | $96.63^{\pm 0.43}$ | $95.06^{\pm 0.72}$ | $96.95^{\pm 0.09}$ | $96.77^{\pm 0.07}$ | $96.92^{\pm 0.22}$ | $97.56^{\pm 0.26}$ | $96.39^{\pm 0.30}$ | $95.91^{\pm 0.17}$ |
| Mean | FPR95 ↓ | $\mathbf{0.92}$ | $3.38$ | $4.61$ | $3.83$ | $3.43$ | $8.17$ | $2.28$ | $3.73$ | $5.84$ |
| | AUROC ↑ | $\mathbf{99.55}$ | $99.07$ | $98.88$ | $99.06$ | $99.18$ | $98.43$ | $99.21$ | $98.92$ | $98.90$ |
| ID | Accuracy ↑ | $94.02^{\pm 0.09}$ | $94.74^{\pm 0.13}$ | $94.93^{\pm 0.12}$ | $94.72^{\pm 0.17}$ | $\mathbf{95.11^{\pm 0.05}}$ | $\mathbf{96.60^{\pm 1.50}}$ | $89.20^{\pm 1.30}$ | $91.32^{\pm 0.35}$ | $94.83^{\pm 0.23}$ |

Table 2: OOD detection performance on ImageNet-1K. We compare results from Hopfield Boosting, DOS (Jiang et al., 2024), DOE (Wang et al., 2023b), DivOE (Zhu et al., 2023), DAL (Wang et al., 2023a), MixOE (Zhang et al., 2023b), POEM (Ming et al., 2022), EBO-OE (Liu et al., 2020), and MSP-OE (Hendrycks et al., 2019b) on ResNet-50. ↓ indicates "lower is better" and ↑ "higher is better". All values in %. Standard deviations are estimated across five training runs.

| | Metric | HB (ours) | DOS | DOE | DivOE | DAL | MixOE | POEM | EBO-OE | MSP-OE |
|---|---|---|---|---|---|---|---|---|---|---|
| Textures | FPR95 ↓ | $44.59^{\pm 1.05}$ | $40.29^{\pm 0.93}$ | $83.83^{\pm 7.19}$ | $42.80^{\pm 0.74}$ | $43.88^{\pm 0.66}$ | $41.05^{\pm 1.29}$ | $31.26^{\pm 0.67}$ | $\mathbf{29.67^{\pm 1.26}}$ | $48.38^{\pm 0.87}$ |
| | AUROC ↑ | $88.01^{\pm 0.57}$ | $89.88^{\pm 0.18}$ | $64.22^{\pm 9.25}$ | $88.18^{\pm 0.06}$ | $87.39^{\pm 0.15}$ | $88.51^{\pm 1.29}$ | $\mathbf{92.22^{\pm 0.14}}$ | $\mathbf{92.40^{\pm 0.23}}$ | $86.25^{\pm 0.25}$ |
| SUN | FPR95 ↓ | $\mathbf{37.37^{\pm 1.84}}$ | $59.29^{\pm 0.96}$ | $83.73^{\pm 8.78}$ | $61.00^{\pm 0.57}$ | $65.31^{\pm 0.61}$ | $65.14^{\pm 2.53}$ | $57.46^{\pm 0.90}$ | $57.69^{\pm 1.61}$ | $66.01^{\pm 0.26}$ |
| | AUROC ↑ | $\mathbf{91.24^{\pm 0.52}}$ | $84.30^{\pm 0.21}$ | $72.95^{\pm 7.94}$ | $83.64^{\pm 0.30}$ | $81.47^{\pm 0.22}$ | $82.20^{\pm 0.72}$ | $85.38^{\pm 0.35}$ | $85.83^{\pm 0.60}$ | $81.45^{\pm 0.20}$ |
| Places 365 | FPR95 ↓ | $\mathbf{53.31^{\pm 2.05}}$ | $69.72^{\pm 1.01}$ | $86.30^{\pm 6.69}$ | $71.09^{\pm 0.60}$ | $74.46^{\pm 0.75}$ | $71.34^{\pm 1.49}$ | $68.87^{\pm 1.05}$ | $70.03^{\pm 1.83}$ | $74.58^{\pm 0.44}$ |
| | AUROC ↑ | $\mathbf{87.10^{\pm 0.52}}$ | $81.62^{\pm 0.22}$ | $70.37^{\pm 7.17}$ | $80.35^{\pm 0.33}$ | $78.72^{\pm 0.28}$ | $80.31^{\pm 0.42}$ | $81.79^{\pm 0.40}$ | $81.35^{\pm 0.63}$ | $78.89^{\pm 0.19}$ |
| iNaturalist | FPR95 ↓ | $\mathbf{11.11^{\pm 0.66}}$ | $49.55^{\pm 1.41}$ | $70.82^{\pm 13.89}$ | $30.51^{\pm 0.42}$ | $51.92^{\pm 0.74}$ | $47.28^{\pm 1.55}$ | $45.37^{\pm 1.79}$ | $49.02^{\pm 4.40}$ | $51.73^{\pm 1.35}$ |
| | AUROC ↑ | $\mathbf{97.65^{\pm 0.20}}$ | $90.49^{\pm 0.38}$ | $83.82^{\pm 5.75}$ | $93.81^{\pm 0.10}$ | $88.33^{\pm 0.21}$ | $90.19^{\pm 0.35}$ | $92.01^{\pm 0.33}$ | $91.44^{\pm 0.79}$ | $88.51^{\pm 0.30}$ |
| Mean | FPR95 ↓ | $\mathbf{36.60}$ | $54.71$ | $81.17$ | $51.35$ | $58.90$ | $56.20$ | $50.74$ | $51.60$ | $60.17$ |
| | AUROC ↑ | $\mathbf{91.00}$ | $86.57$ | $72.84$ | $86.49$ | $83.98$ | $85.30$ | $87.85$ | $87.75$ | $83.78$ |
| ID | Accuracy ↑ | $\mathbf{76.30^{\pm 0.04}}$ | $76.04^{\pm 0.02}$ | $64.36^{\pm 6.94}$ | $74.86^{\pm 0.03}$ | $75.46^{\pm 0.04}$ | $75.47^{\pm 0.20}$ | $75.66^{\pm 0.05}$ | $75.64^{\pm 0.09}$ | $75.71^{\pm 0.03}$ |

## 4.2 Data & Setup

**CIFAR-10 & CIFAR-100.** Our training and evaluation proceeds as follows: We train Hopfield Boosting with ResNet-18 (He et al., 2016) on the CIFAR-10 and CIFAR-100 data sets (Krizhevsky, 2009), respectively. In these settings, we use ImageNet-RC (Chrabaszcz et al., 2017) (a low-resolution version of ImageNet) as the AUX data set. For testing the OOD detection performance, we use the data sets SVHN (Street View House Numbers) (Netzer et al., 2011), Textures (Cimpoi et al., 2014), iSUN (Xu et al., 2015), Places 365 (López-Cifuentes et al., 2020), and two versions of the LSUN data set (Yu et al., 2015) — one where the images are cropped, and one where they are resized to match the resolution of the CIFAR data sets (32x32 pixels). We refer to the two LSUN data sets as LSUN-Crop and LSUN-Resize, respectively. We compute the scores $s(\boldsymbol{\xi})$ as described in Equation (13) and then evaluate the discriminative power of $s(\boldsymbol{\xi})$ between CIFAR and the respective OOD data set using the FPR95 and the AUROC. We use a validation process with different OOD data for model selection. Specifically, we validate the model on MNIST (LeCun et al., 1998), and ImageNet-RC with different pre-processing than in training (resize to 32x32 pixels instead of crop to 32x32 pixels), as well as Gaussian and uniform noise.

**ImageNet-1K.** We evaluate Hopfield Boosting on the large-scale benchmark: We use ImageNet-1K (Russakovsky et al., 2015) as ID data set and ImageNet-21K (Ridnik et al., 2021) as AUX data set. The OOD test data sets are Textures (Cimpoi et al., 2014), SUN (Xu et al., 2015), Places 365 (López-Cifuentes et al., 2020), and iNaturalist (Van Horn et al., 2018). In this setting, all images are scaled to a resolution of 224x224. To keep our method comparable to other OE methods, we closely follow the training and evaluation protocol of (Zhu et al., 2023). This implies htat we fine-tune a ResNet-50 that was pre-trained on the ImageNet-1K ID classification task (as provided by TorchVision, 2016).

Table 3: Ablated training procedures on CIFAR-10. We compare the result of Hopfield Boosting to the results of our method when not using weighted sampling, the projection head, or the OOD loss. ↓ indicates "lower is better" and ↑ "higher is better". All values in %. Standard deviations are estimated across five training runs.

| | | Weighted Sampling ✓
Projection Head ✓
$\mathcal{L}_{\text{OOD}}$ ✓ | ✗
✓
✓ | ✗
✗
✓ | ✗
✗
✗ |
|---|---|---|---|---|---|
| SVHN | FPR95 ↓ | $\mathbf{0.23^{\pm 0.06}}$ | $0.70^{\pm 0.13}$ | $1.01^{\pm 0.27}$ | $45.65^{\pm 13.51}$ |
| | AUROC ↑ | $\mathbf{99.57^{\pm 0.06}}$ | $\mathbf{99.55^{\pm 0.08}}$ | $99.21^{\pm 0.21}$ | $90.99^{\pm 2.68}$ |
| LSUN-Crop | FPR95 ↓ | $\mathbf{0.28^{\pm 0.05}}$ | $1.58^{\pm 0.31}$ | $2.22^{\pm 0.36}$ | $28.59^{\pm 4.33}$ |
| | AUROC ↑ | $\mathbf{99.40^{\pm 0.05}}$ | $99.24^{\pm 0.10}$ | $98.28^{\pm 0.25}$ | $94.08^{\pm 0.86}$ |
| LSUN-Resize | FPR95 ↓ | $\mathbf{0.00^{\pm 0.02}}$ | $\mathbf{0.00^{\pm 0.00}}$ | $\mathbf{0.00^{\pm 0.00}}$ | $50.30^{\pm 7.16}$ |
| | AUROC ↑ | $\mathbf{99.98^{\pm 0.02}}$ | $\mathbf{99.98^{\pm 0.01}}$ | $\mathbf{99.98^{\pm 0.01}}$ | $89.15^{\pm 1.89}$ |
| Textures | FPR95 ↓ | $\mathbf{0.16^{\pm 0.01}}$ | $0.26^{\pm 0.06}$ | $0.38^{\pm 0.10}$ | $49.36^{\pm 1.63}$ |
| | AUROC ↑ | $\mathbf{99.85^{\pm 0.01}}$ | $99.81^{\pm 0.02}$ | $99.70^{\pm 0.06}$ | $89.58^{\pm 0.52}$ |
| iSUN | FPR95 ↓ | $0.00^{\pm 0.02}$ | $\mathbf{0.00^{\pm 0.00}}$ | $\mathbf{0.00^{\pm 0.00}}$ | $51.08^{\pm 6.38}$ |
| | AUROC ↑ | $99.97^{\pm 0.02}$ | $\mathbf{99.99^{\pm 0.00}}$ | $99.98^{\pm 0.01}$ | $88.89^{\pm 1.74}$ |
| Places 365 | FPR95 ↓ | $\mathbf{4.28^{\pm 0.11}}$ | $6.20^{\pm 0.21}$ | $8.73^{\pm 1.06}$ | $77.44^{\pm 0.81}$ |
| | AUROC ↑ | $\mathbf{98.51^{\pm 0.11}}$ | $97.68^{\pm 0.21}$ | $94.77^{\pm 0.81}$ | $78.30^{\pm 0.83}$ |
| Mean | FPR95 ↓ | $\mathbf{0.92}$ | $1.46$ | $2.06$ | $50.40$ |
| | AUROC ↑ | $\mathbf{99.55}$ | $99.38$ | $98.65$ | $88.50$ |

**Baselines.** As mentioned earlier, previous works offer vast experimental evidence that OE methods offer superior OOD detection compared to methods without OE (see e.g., Ming et al., 2022; Wang et al., 2023a). Our experiments in Appendix I.14 confirm this. Thus, we focus on a comprehensive comparison of Hopfield Boosting to eight OE methods: MSP-OE (Hendrycks et al., 2019b), EBO-OE (Liu et al., 2020), POEM (Ming et al., 2022), MixOE (Zhang et al., 2023b), DAL (Wang et al., 2023a), DivOE (Zhu et al., 2023), DOE (Wang et al., 2023b) and DOS (Jiang et al., 2024).

**Training setup.** The network trains for 100 epochs (CIFAR-10/100) or 4 epochs (ImageNet-1K), respectively. In each epoch, the model processes the entire ID data set and a selection of AUX samples (sampled according to $\boldsymbol{w}_t$). We sample mini-batches of size 128 per data set, resulting in a combined batch size of 256. We evaluate the composite loss from Equation (11) for each resulting mini-batch and update the model accordingly. After an epoch, we update the sample weights, yielding $\boldsymbol{w}_{t+1}$. For efficiency reasons, we only compute the weights for 500,000 AUX data instances (∼40% of ImageNet), which we denote as $\boldsymbol{\Xi}$. The weights of the remaining samples are set to 0. During the sample weight update, Hopfield Boosting does not compute gradients or update model parameters. The update of the sample weights $\boldsymbol{w}_{t+1}$ proceeds as follows: First, we fill the memories $\boldsymbol{X}$ and $\boldsymbol{O}$ with 50,000 ID samples and 50,000 AUX samples, respectively. Second, we use the obtained $\boldsymbol{X}$ and $\boldsymbol{O}$ to get the energy $\mathrm{E}_b(\boldsymbol{\Xi}; \boldsymbol{X}, \boldsymbol{O})$ for each of the 500,000 AUX samples in $\boldsymbol{\Xi}$ and compute $\boldsymbol{w}_{t+1}$ according to Equation (10). In the following epoch, Hopfield Boosting samples the mini-batches $\boldsymbol{O}_s^{\mathcal{D}}$ according to $\boldsymbol{w}_{t+1}$ with replacement. To allow the storage of even more patterns in the Hopfield memory during the weight update process, one could incorporate a vector similarity engine (e.g., Douze et al., 2024) into the process. This would potentially allow a less noisy estimate of the sample weights. For the sake of simplicity, we did not opt to do this in our implementation of Hopfield Boosting. As we show in section 4.3, Hopfield Boosting achieves state-of-the-art OOD detection results and can scale to large datasets (ImageNet-1K) even without access to a similarity engine.

**Hyperparameters & Model Selection.** Like Yang et al. (2022), we use SGD with an initial learning rate of $0.1$ and a weight decay of $5 \cdot 10^{-4}$. We decrease the learning rate during the training process with a cosine schedule (Loshchilov & Hutter, 2016). Appendix I.2 describes the image transformations and pre-processing. We apply optimizer, weight decay, learning rate, scheduler, and transformations consistently to all OOD detection methods of the comparison. For training Hopfield Boosting, we use a single value for $\beta$ throughout the training and evaluation process and for all OOD data sets. For model selection, we use a grid search with $\lambda$, chosen from the set $\{0.1, 0.25, 0.5, 1.0\}$, and $\beta$, chosen from the set $\{2, 4, 8, 16, 32\}$. From these hyperparameter configurations, we select the model with the lowest mean FPR95 metric (where the mean is taken over the validation OOD data sets) and do not consider the ID classification accuracy for model selection. In our experiments, $\beta = 4$ and $\lambda = 0.5$ yields the best results for CIFAR-10 and CIFAR-100. For ImageNet-1K, we set $\beta = 32$ and $\lambda = 0.25$.

### 4.3 Results & Discussion

Table 1 summarizes the results for CIFAR-10. Hopfield Boosting achieves equal or better performance compared to the other methods regarding the FPR95 metric for all OOD data sets. It surpasses POEM (the previously best OOD detection approach with OE in our comparison), improving the mean FPR95 metric from 2.28 to 0.92. On CIFAR-100 (Appendix I.1), Hopfield Boosting improves the mean FPR95 metric from 11.76 to 7.94. It is notable that all methods achieve perfect FPR95 results on the LSUN-Resize and iSUN data sets. This is somewhat problematic since there exists evidence that the LSUN-Resize data set can give misleading results due to image artifacts resulting from the resizing procedure (Tack et al., 2020; Yang et al., 2022). We hypothesize that a similar issue exists with the iSUN data set, as in our experiments, LSUN-Resize and iSUN behave very similarly.

On ImageNet-1K (Table 2), Hopfield Boosting surpasses all methods in our comparison in terms of both mean FPR95 and mean AUROC. Compared to POEM (the previously best method) Hopfield Boosting improves the mean FPR95 from 50.74 to 36.60. This demonstrates that Hopfield Boosting scales very favourably to large-scale settings.

We observe that all methods tested perform worst on the Places 365 data set. To gain more insights regarding this behavior, we look at the data instances from the Places 365 data set that Hopfield Boosting trained on CIFAR-10 most confidently classifies as in-distribution (i.e., which receive the highest scores $s(\boldsymbol{\xi})$). Visual inspection shows that among those images, a large portion contains objects from semantic classes included in CIFAR-10 (e.g., airplanes, horses, automobiles). We refer to Appendix I.6 for more details.

We evaluate the performance of the following 3 ablated training procedures on the CIFAR-10 benchmark to gauge the importance of the individual contributions of Hopfield Boosting: (a) Random sampling instead of weighted sampling, (b) Random sampling instead of weighted sampling and no projection head, (c) No application of $\mathcal{L}_{\text{OOD}}$. The results (Table 3) show that all contributions (i.e, weighted sampling, the projection head, and $\mathcal{L}_{\text{OOD}}$) are important factors for Hopfield Boosting 's performance. For additional ablations, we refer to Appendix I.

When subjecting Hopfield Boosting to data sets that were designed to show the weakness of OOD detection approaches (Appendix I.7), we identify instances where a substantial number of outliers are wrongly classified as inliers. Testing with EBO-OE yields comparable outcomes, indicating that this phenomenon extends beyond Hopfield Boosting.

## 5 Limitations

Lastly, we would like to discuss two limitations that we found: First, we see an opportunity to improve the evaluation procedure for OOD detection. Specifically, it remains unclear how reliably the performance on specific data sets can indicate the general ability to detect OOD inputs. Our results from iSUN and LSUN-Resize (Section 4.3) indicate that issues like image artifacts in data sets greatly influence model evaluation. Second, although OE-based approaches improve the OOD detection capability, their reliance on AUX data can limit their applicability. For one, the selection of the AUX data is crucial (since it determines the characteristics of the decision boundary surrounding the inlier data). Furthermore, the use of AUX data can be prohibitive in domains where only a few or no outliers at all are available for training the model.

## 6 Conclusions

We introduce Hopfield Boosting: an approach for OOD detection with OE. Hopfield Boosting uses an energy term to *boost* a classifier between inlier and outlier data by sampling weak learners that are close to the decision boundary. We illustrate how Hopfield Boosting shapes the energy surface to form a decision boundary. Additionally, we demonstrate how the boosting mechanism creates a sharper decision boundary than with random sampling. We compare Hopfield Boosting to eight modern OOD detection approaches using OE. Overall, Hopfield Boosting shows the best results.

## Acknowledgements

We thank Christian Huber for helpful feedback and fruitful discussions.

The ELLIS Unit Linz, the LIT AI Lab, the Institute for Machine Learning, are supported by the Federal State Upper Austria. We thank the projects AI-MOTION (LIT-2018-6-YOU-212), DeepFlood (LIT-2019-8-YOU-213), Medical Cognitive Computing Center (MC3), INCONTROL-RL (FFG-881064), PRIMAL (FFG-873979), S3AI (FFG-872172), DL for GranularFlow (FFG-871302), EPILEPSIA (FFG-892171), AIRI FG 9-N (FWF-36284, FWF-36235), AI4GreenHeatingGrids(FFG- 899943), INTEGRATE (FFG-892418), ELISE (H2020-ICT-2019-3 ID: 951847), Stars4Waters (HORIZON-CL6-2021-CLIMATE-01-01). We thank Audi.JKU Deep Learning Center, TGW LOGISTICS GROUP GMBH, Silicon Austria Labs (SAL), University SAL Labs initiative, FILL Gesellschaft mbH, Anyline GmbH, Google, ZF Friedrichshafen AG, Robert Bosch GmbH, UCB Biopharma SRL, Merck Healthcare KGaA, Verbund AG, GLS (Univ. Waterloo) Software Competence Center Hagenberg GmbH, TÜV Austria, Frauscher Sensonic, Borealis AG, TRUMPF and the NVIDIA Corporation. This work has been supported by the "University SAL Labs" initiative of Silicon Austria Labs (SAL) and its Austrian partner universities for applied fundamental research for electronic-based systems. Daniel Klotz acknowledges funding from the Helmholtz Initiative and Networking Fund (Young Investigator Group COMPOUNDX, grant agreement no. VH-NG-1537) We acknowledge EuroHPC Joint Undertaking for awarding us access to Karolina at IT4Innovations, Czech Republic and Leonardo at CINECA, Italy.

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

# A  Details on Continuous Modern Hopfield Networks

The following arguments are adopted from Fürst et al. (2022) and Ramsauer et al. (2021). Associative memory networks have been designed to store and retrieve samples. Hopfield networks are energy-based, binary associative memories, which were popularized as artificial neural network architectures in the 1980s (Hopfield, 1982, 1984). Their storage capacity can be considerably increased by polynomial terms in the energy function (Chen et al., 1986; Psaltis & Park, 1986; Baldi & Venkatesh, 1987; Gardner, 1987; Abbott & Arian, 1987; Horn & Usher, 1988; Caputo & Niemann, 2002; Krotov & Hopfield, 2016). In contrast to these binary memory networks, we use continuous associative memory networks with far higher storage capacity. These networks are continuous and differentiable, retrieve with a single update, and have exponential storage capacity (and are therefore scalable, i.e., able to tackle large problems; Ramsauer et al., 2021).

Formally, we denote a set of patterns $\{\boldsymbol{x}_1, \ldots, \boldsymbol{x}_N\} \subset \mathbb{R}^d$ that are stacked as columns to the matrix $\boldsymbol{X} = (\boldsymbol{x}_1, \ldots, \boldsymbol{x}_N)$ and a state pattern (query) $\boldsymbol{\xi} \in \mathbb{R}^d$ that represents the current state. The largest norm of a stored pattern is $M = \max_i \|\boldsymbol{x}_i\|$. Then, the energy E of continuous Modern Hopfield Networks with state $\boldsymbol{\xi}$ is defined as (Ramsauer et al., 2021)

$$\mathrm{E} \; = \; -\,\beta^{-1}\,\log\left(\sum_{i=1}^N \exp(\beta \boldsymbol{x}_i^T \boldsymbol{\xi})\right) \; + \; \frac{1}{2}\,\boldsymbol{\xi}^T\boldsymbol{\xi} \; + \; \mathrm{C}, \tag{14}$$

where $\mathrm{C} = \beta^{-1}\log N \; + \; \frac{1}{2}\,M^2$. For energy E and state $\boldsymbol{\xi}$, Ramsauer et al. (2021) proved that the update rule

$$\boldsymbol{\xi}^{\mathrm{new}} \; = \; \boldsymbol{X}\,\mathrm{softmax}(\beta \boldsymbol{X}^T \boldsymbol{\xi}) \tag{15}$$

converges globally to stationary points of the energy E and coincides with the attention mechanisms of Transformers (Vaswani et al., 2017; Ramsauer et al., 2021).

The *separation* $\Delta_i$ of a pattern $\boldsymbol{x}_i$ is its minimal dot product difference to any of the other patterns:

$$\Delta_i = \min_{j, j \neq i}\left(\boldsymbol{x}_i^T\boldsymbol{x}_i - \boldsymbol{x}_i^T\boldsymbol{x}_j\right). \tag{16}$$

A pattern is *well-separated* from the data if $\Delta_i$ is above a given threshold (specified in Ramsauer et al., 2021). If the patterns $\boldsymbol{x}_i$ are well-separated, the update rule Equation 15 converges to a fixed point close to a stored pattern. If some patterns are similar to one another and, therefore, not well-separated, the update rule converges to a fixed point close to the mean of the similar patterns.

The update rule of a Hopfield network thus identifies sample–sample relations between stored patterns. This enables similarity-based learning methods like nearest neighbor search (see Schäfl et al., 2022), which Hopfield Boosting leverages to detect samples outside the distribution of the training data.

Hopfield networks have recently been used for OOD detection (Zhang et al., 2023a). Hu et al. (2024) introduces Hopfield layers for outlier-efficient memory update.

# B  Notes on Langevin Sampling

Another method that is appropriate for earlier acquired models is to sample the posterior via the Stochastic Gradient Langevin Dynamics (SGLD) (Welling & Teh, 2011). This method is efficient since it iteratively learns from small mini-batches Welling & Teh (2011); Ahn et al. (2012). See basic work on Langevin dynamics Welling & Teh (2011); Ahn et al. (2012); Teh et al. (2016); Xu et al. (2018). A cyclical stepsize schedule for SGLD was very promising for uncertainty quantification Zhang et al. (2020). Larger steps discover new modes, while smaller steps characterize each mode and perform the posterior sampling.

# C  Related work

## C.1  Details on further OE approaches

This section gives details about related works from the area of OE in OOD detection. With OE, we refer to the usage of AUX for training an OOD detector in general.

**MSP-OE.** Hendrycks et al. (2019b) were the first to introduce the term OE in the context of OOD detection. Specifically, they improve an MSP-based OOD detection (Hendrycks & Gimpel, 2017): They train a classifier on the ID data set and maximize the entropy of the predictive distribution of the classifier for the AUX data. The combined loss they employ is

$$\mathcal{L} = \mathcal{L}_{\text{CE}} + \lambda\mathcal{L}_{\text{OOD}} \tag{17}$$

$$\mathcal{L}_{\text{OOD}} = \mathbb{E}_{\boldsymbol{o}^{\mathcal{D}} \sim p_{\text{AUX}}} [H(\mathcal{U}, p_{\boldsymbol{\theta}}(\boldsymbol{o}))] \tag{18}$$

where $H$ denotes the cross-entropy, $\mathcal{U}$ denotes the uniform distribution over K classes, and $p_{\boldsymbol{\theta}}$ is the model mapping the features to the predictive distribution over the K classes.

**EBO-OE.** Liu et al. (2020) propose a post-hoc and an OE approach. Their post-hoc approach (EBO) is to use the classifier's energy to perform OOD detection:

$$\text{E}(\boldsymbol{\xi}^{\mathcal{D}}) = -\beta^{-1}\text{lse}(\beta, f_{\boldsymbol{\theta}}(\boldsymbol{\xi}^{\mathcal{D}})) \tag{19}$$

$$s(\boldsymbol{\xi}^{\mathcal{D}}) = -\text{E}(\boldsymbol{\xi}^{\mathcal{D}}, f_{\boldsymbol{\theta}}) \tag{20}$$

where $f_{\boldsymbol{\theta}}$ outputs the model's logits as a vector. Their OE approach (EBO-OE) promotes a low energy on ID samples and a high energy on AUX samples:

$$\mathcal{L}_{\text{OOD}} = \mathbb{E}_{\boldsymbol{x}^{\mathcal{D}} \sim p_{\text{ID}}} [(\max(0, \text{E}(\boldsymbol{x}^{\mathcal{D}}) - m_{\text{ID}}))^2] + \mathbb{E}_{\boldsymbol{o}^{\mathcal{D}} \sim p_{\text{AUX}}} [(\max(0, m_{\text{AUX}} - \text{E}(\boldsymbol{o}^{\mathcal{D}})))^2] \tag{21}$$

where $m_{\text{ID}}$ and $m_{\text{AUX}}$ are margin hyperparameters.

**POEM.** Ming et al. (2022) propose to incorporate Thompson sampling into the OE process. More specifically, they sample a linear decision boundary in embedding space between the ID and AUX data using Bayesian linear regression and then select those samples from the AUX data set that are closest to the sampled decision boundary. In the following epoch, they sample the AUX data uniformly from the selected data instances without replacement and optimize the model with the EBO-OE loss (Equation (21)).

**MixOE.** Zhang et al. (2023b) employ mixup (Zhang et al., 2018) between the ID and AUX samples to augment the OE task. Formally, this results in the following:

$$\tilde{\boldsymbol{x}} = \lambda\boldsymbol{x}^{\mathcal{D}} + (1 - \lambda)\boldsymbol{o}^{\mathcal{D}} \tag{22}$$

$$\tilde{y} = \lambda y + (1 - \lambda)\mathcal{U} \tag{23}$$

$$\mathcal{L}_{\text{OOD}} = \mathbb{E}_{\substack{(\boldsymbol{x}^{\mathcal{D}}, y) \sim p_{\text{ID}} \\ \boldsymbol{o}^{\mathcal{D}} \sim p_{\text{AUX}}}} [H(\tilde{y}, \tilde{\boldsymbol{x}})] \tag{24}$$

Alternatively, they also propose to employ CutMix (Yun et al., 2019) instead of mixup (which would change the mixing operation in Equation (22)).

**DAL.** Wang et al. (2023a) augment the AUX data by defining a Wasserstein-1 ball around the AUX data and performing OE using this Wasserstein ball. DAL is motivated by the concept of distribution discrepancy: The distribution of the real OOD data will in general be different from the distribution of the AUX data. The authors argue that their approach can make OOD detection more reliable if the distribution discrepancy is large.

**DivOE.** Zhu et al. (2023) pose the question of how to utilize the given outliers from the AUX data set if the auxiliary outliers are not informative enough to represent the unseen OOD distribution. They suggest solving this problem by diversifying the AUX data using extrapolation, which should

result in better coverage of the OOD space of the resultant extrapolated distribution. Formally, they employ a loss using a synthesized distribution with a manipulation $\Delta$:

$$\mathcal{L}_{\text{OOD}} = \mathop{\mathbb{E}}_{\boldsymbol{o}^{\mathcal{D}} \sim p_{\text{AUX}}} [(1 - \gamma)H(\mathcal{U}, p_{\boldsymbol{\theta}}(\boldsymbol{o}^{\mathcal{D}})) + \gamma \max_{\Delta}[H(\mathcal{U}, p_{\boldsymbol{\theta}}(\boldsymbol{o}^{\mathcal{D}} + \Delta)) - H(\mathcal{U}, p_{\boldsymbol{\theta}}(\boldsymbol{o}^{\mathcal{D}}))]] \tag{25}$$

**DOE.** Wang et al. (2023b) implicitly synthesize auxiliary outlier data using a transformation of the model weights. They argue that perturbing the model parameters has the same effect as transforming the data.

**DOS.** Jiang et al. (2024) apply K-means clustering to the features of the AUX data set. They then employ a balanced sampling from the K obtained clusters by selecting the same number of samples from each cluster for training. More specifically, they select those n samples from each cluster which are closest to the decision boundary between the ID and OOD regions.

# D    Future Work

## D.1    Smooth and Sharp Decision Boundaries

Our work treats samples close to the decision boundary as weak learners. However, the wider ramifications of this behavior remain unclear. One can also view the sampling of data instances close to the decision boundary as a form of adversarial training in that we search for something like "natural adversarial examples": Loosely speaking, the usual adversarial example case starts with a given ID sample and corrupts it in a specific way to get the classifier to output the wrong class probabilities. In our case, we start with a large set of potential adversarial instances (the AUX data) and search for the ones that could be either ID or OOD samples. That is, the sampling process will more likely select data instances that are hard to discriminate for the model — for example, if the model is uncertain whether a leaf in an auxiliary outlier sample is a frog or not.

This process can be viewed as "sharpening" the decision boundary: The boosting process frequently samples data instances with high uncertainty. $\mathcal{L}_{\text{OOD}}$ encourages the model to assign less uncertainty to the sampled data instances. After training, few training data instances will have high uncertainty. Nevertheless, a closer systematic evaluation of the sharpened decision boundary of Hopfield Boosting is important to fully understand the potential implications w.r.t. adversarial examples. We view such an investigation as out-of-scope for this work. However, we consider it an interesting avenue for future work.

Anderson & Sojoudi (2022) show that a smooth decision boundary helps with "classical" adversarial examples. In this framing, our approach would produce different adversarial examples that are not based on noise but are more akin to "natural adversarial examples". For example, it is perfectly fine for us that an OOD sample close to the boundary does not correspond to any of the ID classes. Furthermore, the noise based smoothing leads to adversarial robustness at the (potential) cost of degrading classification performance. Similarly, our sharpening of the boundaries leads to better discrimination between ID and OOD region at the (potential) cost of degrading ID classification performance.

# E    Societal Impact

This section discusses the potential positive and negative societal impacts of our work. As our work aims improves the state-of-the-art in OOD detection, we focus on potential societal impact of OOD detection in general.

- **Postive Impacts**
    - **Improved model reliability**: OOD detection aims to detect unfamiliar inputs that have little support in the model's training distribution. When these samples are detected, one can, for example, notify the user that no prediction is possible, or trigger a manual intervention. This can lead to an increase in a model's reliability.

- **Abstain from doing uncertain predictions**: When a model with appropriate OOD detection recognizes that a query sample has limited support in the training distribution, it can abstain from performing a prediction. This can, for example, increase trust in ML models, as they will rather tell the user they are uncertain than report a confidently wrong prediction.

- **Negative Impacts**
  - **Wrong sense of safety**: Having OOD detection in place could cause users to wrongly assume that all OOD inputs will be detected. However, like most systems, also OOD detection methods can make errors. It is important to consider that certain OOD examples could remain undetected.
  - **Potential for misinterpretation**: As with many other ML systems, the outcomes of OOD detection methods are prone to misinterpretation. It is important to acquaint oneself with the respective method before applying it in practice.

# F   Toy Examples

## F.1   3D Visualizations of $\mathrm{E}_b$ on a hypersphere

(a) Out-of-Distribution energy function

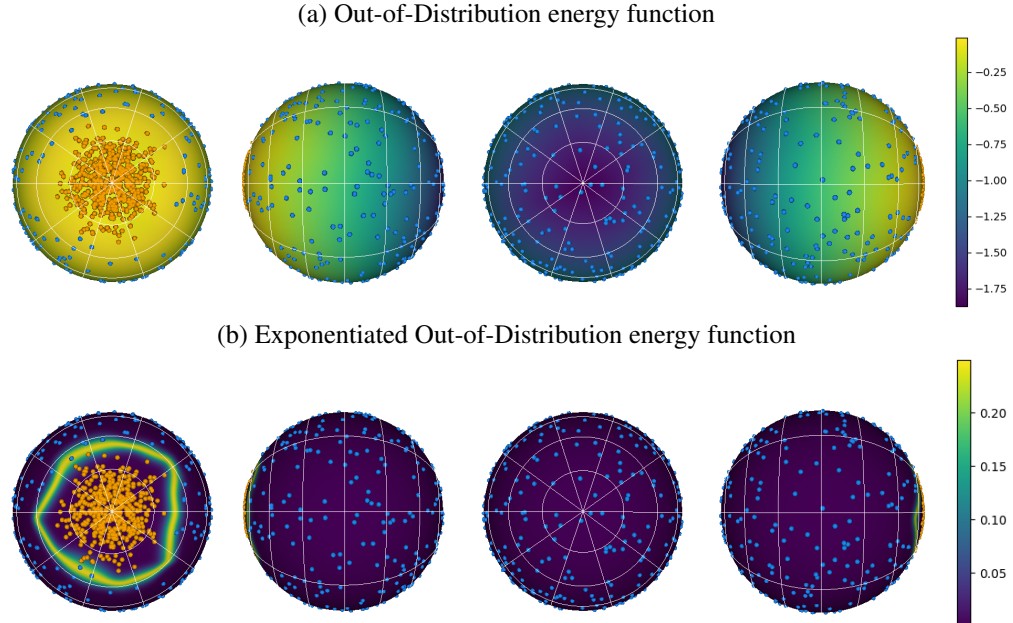

(b) Exponentiated Out-of-Distribution energy function

Figure 3: Depiction of the energy function $\mathrm{E}_b(\boldsymbol{\xi}; \boldsymbol{X}, \boldsymbol{O})$ on a hypersphere. (a) shows $\mathrm{E}_b(\boldsymbol{\xi}, \boldsymbol{X}, \boldsymbol{O})$ with exemplary inlier (orange) and outlier (blue) points; and (b) shows $\exp(\beta \mathrm{E}_b(\boldsymbol{\xi}, \boldsymbol{X}, \boldsymbol{O}))$. $\beta$ was set to 128. Both, (a) and (b), rotate the sphere by 0, 90, 180, and 270 degrees around the vertical axis.

This example depicts how inliers and outliers shape the energy surface (Figure 3). We generated patterns so that $\boldsymbol{X}$ clusters around a pole and the outliers populate the remaining perimeter of the sphere. This is analogous to the idea that one has access to a large AUX data set, where some data points are more and some less informative for OOD detection (e.g., as conceptualized in Ming et al., 2022).

## F.2   Dynamics of $\mathcal{L}_{\mathbf{OOD}}$ on Patterns in Euclidean Space

In this example, we applied our out-of-distribution loss $\mathcal{L}_{\mathrm{OOD}}$ on a simple binary classification problem. As we are working in Euclidean space and not on a sphere, we use a modified version of MHE, which uses the negative squared Euclidean distance instead of the dot-product-similarity. For the formal relation between Equation (26) and MHE, we refer to Appendix H.1:

$$\mathrm{E}(\boldsymbol{\xi}; \boldsymbol{X}) = -\beta^{-1} \log \left( \sum_{i=1}^{N} \exp(-\frac{\beta}{2} ||\boldsymbol{\xi} - \boldsymbol{x}_i||_2^2) \right) \qquad (26)$$

Figure 4a shows the initial state of the patterns and the decision boundary $\exp(\beta E_b(\boldsymbol{\xi}; \boldsymbol{X}, \boldsymbol{O}))$. We store the samples of the two classes as stored patterns in $\boldsymbol{X}$ and $\boldsymbol{O}$, respectively, and compute $\mathcal{L}_{\mathrm{OOD}}$ for all samples. We then set the learning rate to 0.1 and perform gradient descent with $\mathcal{L}_{\mathrm{OOD}}$ on the data points. Figure 4b shows that after 25 steps, the distance between the data points and the decision boundary has increased, especially for samples that had previously been close to the decision boundary. After 100 steps, as shown in Figure 4d, the variability orthogonal to the decision boundary has almost completely vanished, while the variability parallel to the decision boundary is maintained.

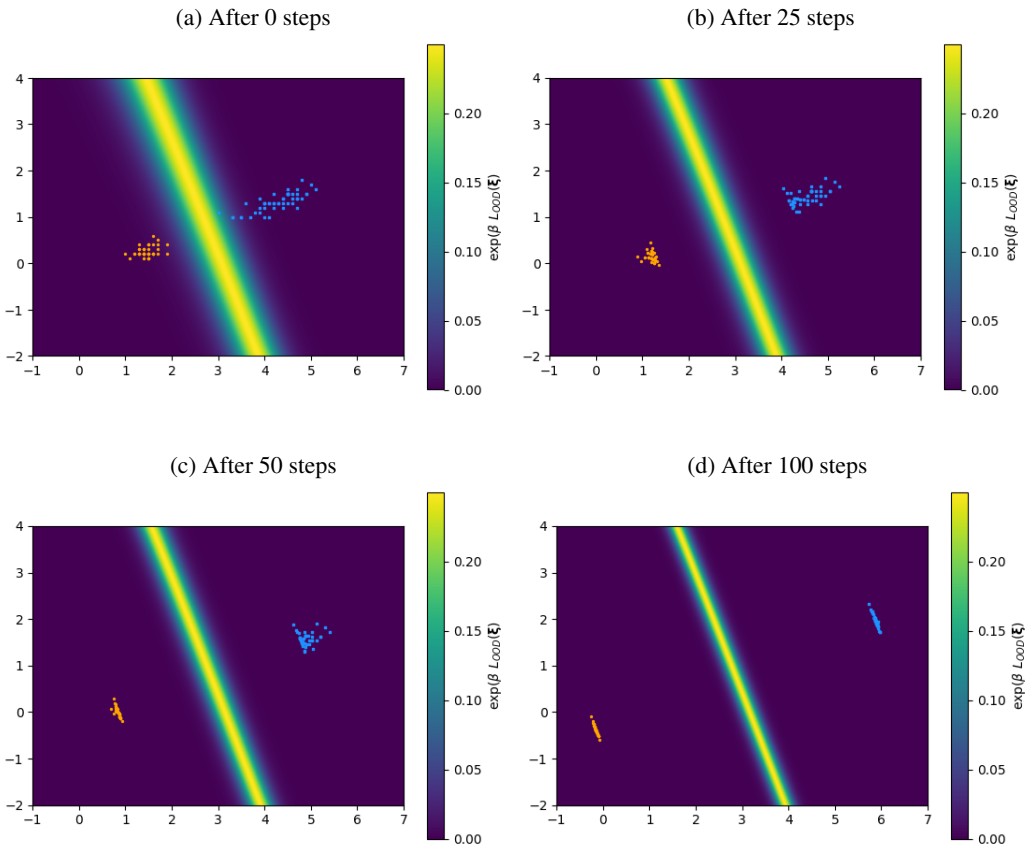

Figure 4: $\mathcal{L}_{\text{OOD}}$ applied to exemplary data points on euclidean space. Gradient updates are applied to the data points directly. We observe that the variance orthogonal to the decision boundary shrinks while the variance parallel to the decision boundary does not change to this extent. $\beta$ is set to 2.

## F.3 Dynamics of $\mathcal{L}_{\text{OOD}}$ on Patterns on the Sphere

(a) After 0 steps

(b) After 500 steps

(c) After 2500 steps

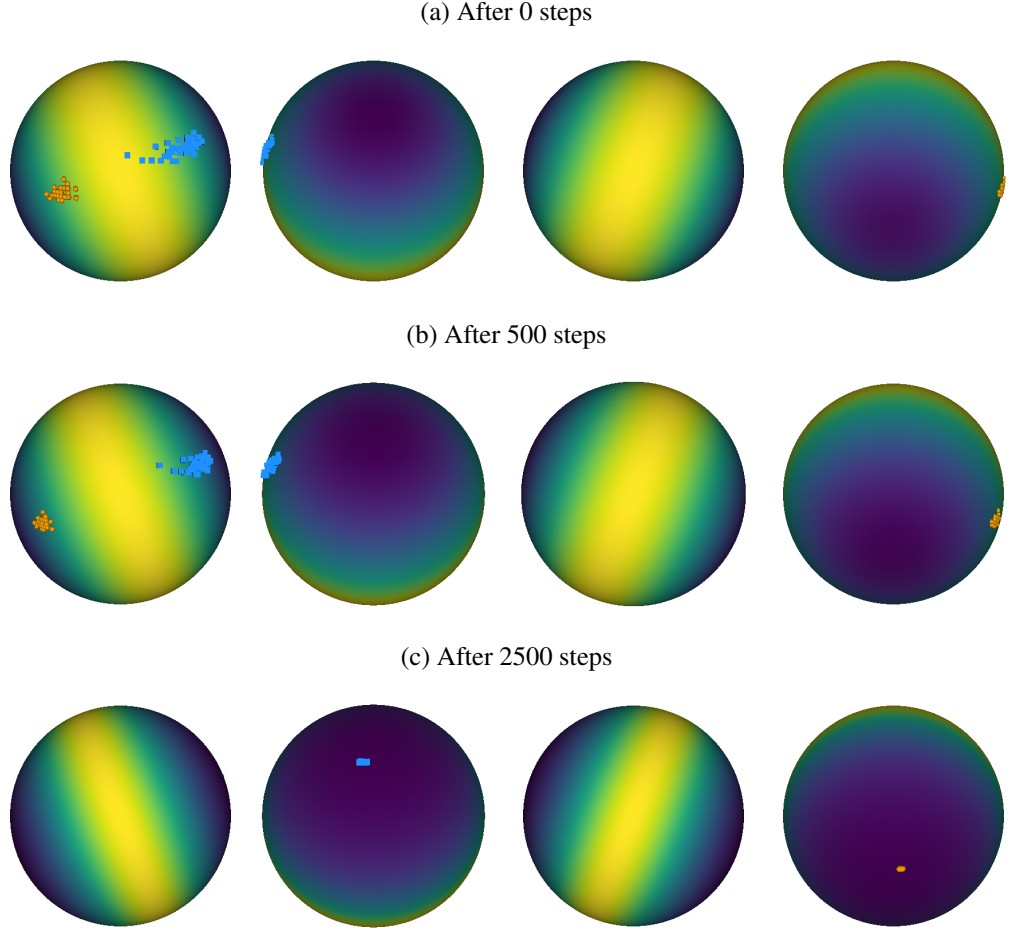

Figure 5: $\mathcal{L}_{\text{OOD}}$ applied to exemplary data points on a sphere. Gradients are applied to the data points directly. We observe that the geometry of the space forces the patterns to opposing poles of the sphere.

## F.4 Learning Dynamics of Hopfield Boosting on Patterns on a Sphere - Video

The example video[1] demonstrates the learning dynamics of Hopfield Boosting on a 3-dimensional sphere. We randomly generate ID patterns $X$ clustering around one of the sphere's poles and AUX patterns $O$ on the remaining surface of the sphere. We then apply Hopfield Boosting on this data set. First, we sample the weak learners close to the decision boundary for both classes, $X$ and $O$. Then, we perform 2000 steps of gradient descent with $\mathcal{L}_{\text{OOD}}$ on the sampled weak learners. We apply the gradient updates to the patterns directly and do not propagate any gradients to an encoder. Every 50 gradient steps, we re-sample the weak learners. For this example, the initial learning rate is set to 0.02 and increased after every gradient step by 0.1%.

---

[1] https://youtu.be/H5tGdL-0fok

## F.5  Location of Weak Learners near the Decision Boundary

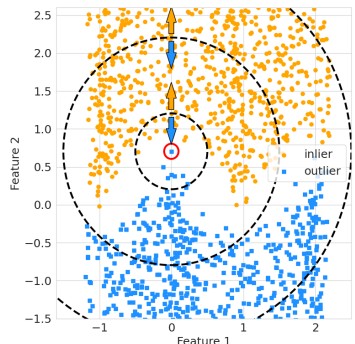

Figure 6: A prototypical classifier (red circle) that is constructed with a sample close to the decision boundary. Classifiers like this one will only perform slightly better than random guessing (as indicated by the radial decision boundaries) and are, therefore, well-suited for weak learners.

## F.6  Interaction between ID and OOD losses

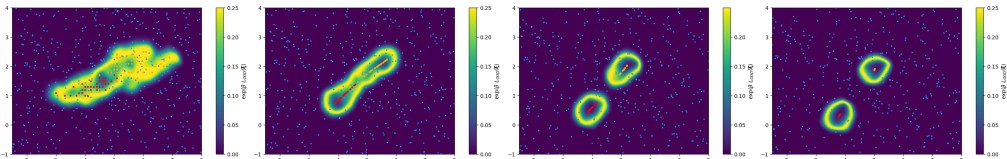

Figure 7: Synthetic example of Hopfield Boosting training dynamics. The ID data is split in two classes (shown in red and orange); the AUX data (blue) is sampled uniformly. We minimize the loss $\mathcal{L} = \mathcal{L}_{\mathrm{CE}} + \mathcal{L}_{\mathrm{OOD}}$, and apply the gradient updates on the patterns directly. The left Figure shows the initial pattern positions, the rightmost Figure shows the positions after 1000 gradient updates: The classes are well-separated; $\mathrm{E}_b$ forms a tight decision boundary around the ID data.

To demonstrate how Hopfield Boosting can interact in a classification task we created a toy example that resembles the ID classification setting (Figure 7): The example shows the decision boundary and the inlier samples organized in two classes (shown in red and orange). We sample uniformly distributed auxiliary outliers. Then, we minimize the compound objective (applying the gradient updates on the patterns directly). This shows that Hopfield Boosting is able to separate the two classes well and that still forms a tight decision boundary around the ID data.

# G  Notes on $\mathrm{E}_b$

## G.1  Probabilistic Interpretation of $\mathrm{E}_b$

We model the class-conditional densities of the in-distribution data and auxiliary data as mixtures of Gaussians with the patterns as the component means and tied, diagonal covariance matrices with $\beta^{-1}$ in the main diagonal.

$$p(\,\boldsymbol{\xi}\,|\,\mathrm{ID}\,) \;=\; \frac{1}{N}\sum_{i=1}^{N}\mathcal{N}\left(\boldsymbol{\xi};\boldsymbol{x}_i,\beta^{-1}\boldsymbol{I}\right) \tag{27}$$

$$p(\,\boldsymbol{\xi}\,|\,\mathrm{AUX}\,) \;=\; \frac{1}{M}\sum_{i=1}^{M}\mathcal{N}\left(\boldsymbol{\xi};\boldsymbol{o}_i,\beta^{-1}\boldsymbol{I}\right) \tag{28}$$

Further, we assume the distribution $p(\boldsymbol{\xi})$ as a mixture of $p(\,\boldsymbol{\xi}\,|\,\mathrm{ID}\,)$ and $p(\,\boldsymbol{\xi}\,|\,\mathrm{AUX}\,)$ with equal prior probabilities (mixture weights):

$$p(\boldsymbol{\xi}) \;=\; p(\mathrm{ID})\,p(\,\boldsymbol{\xi}\,|\,\mathrm{ID}\,) + p(\mathrm{AUX})\,p(\,\boldsymbol{\xi}\,|\,\mathrm{AUX}\,) \tag{29}$$

$$=\; \frac{1}{2}\,p(\,\boldsymbol{\xi}\,|\,\mathrm{ID}\,) \;+\; \frac{1}{2}\,p(\,\boldsymbol{\xi}\,|\,\mathrm{AUX}\,) \tag{30}$$

The probability of an unknown sample $\boldsymbol{\xi}$ being an AUX sample is given by

$$p(\,\mathrm{AUX}\,|\,\boldsymbol{\xi}\,) \;=\; \frac{p(\,\boldsymbol{\xi}\,|\,\mathrm{AUX}\,)\,p(\mathrm{AUX})}{p(\boldsymbol{\xi})} \tag{31}$$

$$=\; \frac{p(\,\boldsymbol{\xi}\,|\,\mathrm{AUX}\,)}{2\,p(\boldsymbol{\xi})} \tag{32}$$

$$=\; \frac{p(\,\boldsymbol{\xi}\,|\,\mathrm{AUX}\,)}{p(\,\boldsymbol{\xi}\,|\,\mathrm{AUX}\,) \;+\; p(\,\boldsymbol{\xi}\,|\,\mathrm{ID}\,)} \tag{33}$$

$$=\; \frac{1}{1 \;+\; \frac{p(\,\boldsymbol{\xi}\,|\,\mathrm{ID}\,)}{p(\,\boldsymbol{\xi}\,|\,\mathrm{AUX}\,)}} \tag{34}$$

$$=\; \frac{1}{1 + \exp(\log(p(\,\boldsymbol{\xi}\,|\,\mathrm{ID}\,)) - \log(p(\,\boldsymbol{\xi}\,|\,\mathrm{AUX}\,)))} \tag{35}$$

where in line (34) we have used that $p(\,\boldsymbol{\xi}\,|\,\mathrm{AUX}\,) > 0$ for all $\boldsymbol{\xi} \in \mathbb{R}^d$. The probability of $\boldsymbol{\xi}$ being an ID sample is given by

$$p(\,\mathrm{ID}\,|\,\boldsymbol{\xi}\,) = \frac{p(\,\boldsymbol{\xi}\,|\,\mathrm{ID}\,)}{2\,p(\boldsymbol{\xi})} \tag{36}$$

$$=\; \frac{1}{1 + \exp(\log(p(\,\boldsymbol{\xi}\,|\,\mathrm{AUX}\,)) - \log(p(\,\boldsymbol{\xi}\,|\,\mathrm{ID}\,)))} \tag{37}$$

$$=\; 1 - p(\,\mathrm{AUX}\,|\,\boldsymbol{\xi}\,) \tag{38}$$

Consider the function

$$f_b(\boldsymbol{\xi}) \;=\; p(\,\mathrm{AUX}\,|\,\boldsymbol{\xi}\,) \cdot p(\,\mathrm{ID}\,|\,\boldsymbol{\xi}\,) \tag{39}$$

$$=\; \frac{p(\,\boldsymbol{\xi}\,|\,\mathrm{AUX}\,) \cdot p(\,\boldsymbol{\xi}\,|\,\mathrm{ID}\,)}{4p(\boldsymbol{\xi})^2} \tag{40}$$

By taking the $\log$ of Equation (40) we obtain the following. We use $\overset{C}{=}$ to denote equality up to an additive constant that does not depend on $\boldsymbol{\xi}$.

$$\beta^{-1} \, \log\left(f_b(\boldsymbol{\xi})\right) \overset{C}{=} -\, 2\, \beta^{-1} \, \log\left(p(\boldsymbol{\xi})\right) \,+\, \beta^{-1} \, \log\left(p(\,\boldsymbol{\xi}\mid \text{ID}\,)\right) \,+\, \beta^{-1} \, \log\left(p(\,\boldsymbol{\xi}\mid \text{AUX}\,)\right) \tag{41}$$

Multiplication by $\beta^{-1}$ is equivalent to a change of base of the log. The term $-\,\beta^{-1}\,\log(p(\boldsymbol{\xi}))$ is equivalent to the MHE (Ramsauer et al., 2021) (up to an additive constant) when assuming normalized patterns, i.e. $||\boldsymbol{x}_i||_2 = 1$ and $||\boldsymbol{o}_i||_2 = 1$, and an equal number of patterns $M = N$ in the two Gaussian mixtures $p(\,\boldsymbol{\xi}\mid \text{ID}\,)$ and $p(\,\boldsymbol{\xi}\mid \text{AUX}\,)$:

$$-\,\beta^{-1}\log(p(\boldsymbol{\xi})) = -\,\beta^{-1}\log\left(\frac{1}{2}p(\,\boldsymbol{\xi}\mid \text{ID}\,) \,+\, \frac{1}{2}p(\,\boldsymbol{\xi}\mid \text{AUX}\,)\right) \tag{42}$$

$$\overset{C}{=} -\,\beta^{-1}\log\left(p(\,\boldsymbol{\xi}\mid \text{ID}\,) \,+\, p(\,\boldsymbol{\xi}\mid \text{AUX}\,)\right) \tag{43}$$

$$= -\,\beta^{-1}\log\left(\frac{1}{N}\sum_{i=1}^{N}\mathcal{N}\left(\boldsymbol{\xi};\boldsymbol{x}_i,\beta^{-1}\boldsymbol{I}\right) \,+\, \frac{1}{N}\sum_{i=1}^{N}\mathcal{N}\left(\boldsymbol{\xi};\boldsymbol{o}_i,\beta^{-1}\boldsymbol{I}\right)\right) \tag{44}$$

$$\overset{C}{=} -\,\beta^{-1}\log\left(\sum_{i=1}^{N}\mathcal{N}\left(\boldsymbol{\xi};\boldsymbol{x}_i,\beta^{-1}\boldsymbol{I}\right) \,+\, \sum_{i=1}^{N}\mathcal{N}\left(\boldsymbol{\xi};\boldsymbol{o}_i,\beta^{-1}\boldsymbol{I}\right)\right) \tag{45}$$

$$\overset{C}{=} -\,\beta^{-1}\log\left(\sum_{i=1}^{N}\exp(-\frac{\beta}{2}||\boldsymbol{\xi}-\boldsymbol{x}_i||_2^2) \,+\, \sum_{i=1}^{N}\exp(-\frac{\beta}{2}||\boldsymbol{\xi}-\boldsymbol{o}_i||_2^2)\right) \tag{46}$$

$$\overset{C}{=} -\,\beta^{-1}\log\left(\sum_{i=1}^{N}\exp(\beta\boldsymbol{x}_i^T\boldsymbol{\xi} - \frac{\beta}{2}\boldsymbol{\xi}^T\boldsymbol{\xi}) \,+\, \sum_{i=1}^{N}\exp(\beta\boldsymbol{o}_i^T\boldsymbol{\xi} - \frac{\beta}{2}\boldsymbol{\xi}^T\boldsymbol{\xi})\right) \tag{47}$$

$$= -\,\beta^{-1}\log\left(\sum_{i=1}^{N}\exp(\beta\boldsymbol{x}_i^T\boldsymbol{\xi}) \,+\, \sum_{i=1}^{N}\exp(\beta\boldsymbol{o}_i^T\boldsymbol{\xi})\right) \,+\, \frac{1}{2}\,\boldsymbol{\xi}^T\boldsymbol{\xi} \tag{48}$$

$$\overset{C}{=} -\,\text{lse}(\beta,(\boldsymbol{X}\mathbin\Vert\boldsymbol{O})^T\boldsymbol{\xi}) \,+\, \frac{1}{2}\,\boldsymbol{\xi}^T\boldsymbol{\xi} \,+\, \beta^{-1}\log N + \frac{1}{2}M^2 \tag{49}$$

Analogously, $\beta^{-1}\,\log(p(\,\boldsymbol{\xi}\mid \text{ID}\,))$ and $\beta^{-1}\,\log(p(\,\boldsymbol{\xi}\mid \text{AUX}\,))$ also yield MHE terms. Therefore, $\text{E}_b$ is equivalent to $\beta^{-1}\log(f_b(\boldsymbol{\xi}))$ under the assumption that $||\boldsymbol{x}_i||_2 = 1$ and $||\boldsymbol{o}_i||_2 = 1$ and $M = N$. The $\frac{1}{2}\boldsymbol{\xi}^T\boldsymbol{\xi}$ terms that are contained in the three MHEs cancel out.

$$\beta^{-1}\,\log\left(f_b(\boldsymbol{\xi})\right) \overset{C}{=} -\,2\,\text{lse}(\beta,(\boldsymbol{X}\mathbin\Vert\boldsymbol{O})^T\boldsymbol{\xi}) \,+\, \text{lse}(\beta,\boldsymbol{X}^T\boldsymbol{\xi}) \,+\, \text{lse}(\beta,\boldsymbol{O}^T\boldsymbol{\xi}) = \text{E}_b(\boldsymbol{\xi};\boldsymbol{X},\boldsymbol{O}) \tag{50}$$

$f_b(\boldsymbol{\xi})$ can also be interpreted as the variance of a Bernoulli distribution with outcomes ID and AUX:

$$f_b(\boldsymbol{\xi}) = \, p(\,\text{AUX}\mid \boldsymbol{\xi}\,)\,p(\,\text{ID}\mid \boldsymbol{\xi}\,) = p(\,\text{ID}\mid \boldsymbol{\xi}\,)(1 - p(\,\text{ID}\mid \boldsymbol{\xi}\,)) \, = \, p(\,\text{AUX}\mid \boldsymbol{\xi}\,)(1 - p(\,\text{AUX}\mid \boldsymbol{\xi}\,)) \tag{51}$$

In other words, minimizing $\text{E}_b$ means to drive a Bernoulli-distributed random variable with the outcomes ID and AUX towards minimum variance, i.e., $p(\,\text{ID}\mid \boldsymbol{\xi}\,)$ is driven towards 1 if $p(\,\text{ID}\mid \boldsymbol{\xi}\,) > 0.5$ and towards 0 if $p(\,\text{ID}\mid \boldsymbol{\xi}\,) < 0.5$. Conversely, the same is true for $p(\,\text{AUX}\mid \boldsymbol{\xi}\,)$.

From Equation (35), under the assumptions that $||\boldsymbol{x}_i||_2 = 1$ and $||\boldsymbol{o}_i||_2 = 1$ and $M = N$, the conditional probability $p(\,\text{AUX}\mid \boldsymbol{\xi}\,)$ can be computed as follows:

$$p(\,\text{AUX}\mid \boldsymbol{\xi}\,) \, = \, \sigma(\log(p(\,\boldsymbol{\xi}\mid \text{AUX}\,)) - \log(p(\,\boldsymbol{\xi}\mid \text{ID}\,))) \tag{52}$$

$$= \, \sigma(\beta\,(\text{lse}(\beta,\boldsymbol{O}^T\boldsymbol{\xi}) - \text{lse}(\beta,\boldsymbol{X}^T\boldsymbol{\xi}))) \tag{53}$$

where $\sigma$ denotes the logistic sigmoid function. Similarly, $p(\text{ ID }|\boldsymbol{\xi})$ can be computed using

$$p(\text{ ID }|\boldsymbol{\xi}) = \sigma(\beta(\text{lse}(\beta, \boldsymbol{X}^T\boldsymbol{\xi}) - \text{lse}(\beta, \boldsymbol{O}^T\boldsymbol{\xi}))) \tag{54}$$
$$= 1 - p(\text{ AUX }|\boldsymbol{\xi}) \tag{55}$$

## G.2  Alternative Formulations of $\text{E}_b$ and $f_b$

$\text{E}_b$ can be rewritten as follows.

$$\text{E}_b(\boldsymbol{\xi}; \boldsymbol{X}, \boldsymbol{O}) = -2\,\text{lse}(\beta, (\boldsymbol{X} \parallel \boldsymbol{O})^T\boldsymbol{\xi}) + \text{lse}(\beta, \boldsymbol{X}^T\boldsymbol{\xi}) + \text{lse}(\beta, \boldsymbol{O}^T\boldsymbol{\xi}) \tag{56}$$
$$= -2\beta^{-1}\,\log\cosh\left(\frac{\beta}{2}\,(\text{lse}(\beta, \boldsymbol{X}^T\boldsymbol{\xi}) - \text{lse}(\beta, \boldsymbol{O}^T\boldsymbol{\xi}))\right) - 2\beta^{-1}\,\log(2) \tag{57}$$

To prove this, we first show the following:

$$-\beta^{-1}\,\log\left(\exp(\beta\,\text{lse}(\beta, \boldsymbol{X}^T\boldsymbol{\xi})) + \exp(\beta\,\text{lse}(\beta, \boldsymbol{O}^T\boldsymbol{\xi}))\right) \tag{58}$$
$$= -\beta^{-1}\,\log\left(\exp\left(\beta\,\beta^{-1}\log\left(\sum_{i=1}^{N}\exp(\beta\boldsymbol{x}_i^T\boldsymbol{\xi})\right)\right) + \exp\left(\beta\,\beta^{-1}\log\left(\sum_{i=1}^{N}\exp(\beta\boldsymbol{o}_i^T\boldsymbol{\xi})\right)\right)\right) \tag{59}$$
$$= -\beta^{-1}\,\log\left(\sum_{i=1}^{N}\exp(\beta\boldsymbol{x}_i^T\boldsymbol{\xi}) + \sum_{i=1}^{N}\exp(\beta\boldsymbol{o}_i^T\boldsymbol{\xi})\right) \tag{60}$$
$$= -\text{lse}(\beta, (\boldsymbol{X} \parallel \boldsymbol{O})^T\boldsymbol{\xi}) \tag{61}$$

Let $\text{E}_{\boldsymbol{X}} = -\text{lse}(\beta, \boldsymbol{X}^T\boldsymbol{\xi})$ and $\text{E}_{\boldsymbol{O}} = -\text{lse}(\beta, \boldsymbol{O}^T\boldsymbol{\xi})$.

$$\text{E}_b(\boldsymbol{\xi}; \boldsymbol{X}, \boldsymbol{O}) = -2\,\text{lse}(\beta, (\boldsymbol{X} \parallel \boldsymbol{O})^T\boldsymbol{\xi}) + \text{lse}(\beta, \boldsymbol{X}^T\boldsymbol{\xi}) + \text{lse}(\beta, \boldsymbol{O}^T\boldsymbol{\xi}) \tag{62}$$
$$= -2\beta^{-1}\,\log\left(\exp(-\beta\,\text{E}_{\boldsymbol{X}}) + \exp(-\beta\,\text{E}_{\boldsymbol{O}})\right) - \text{E}_{\boldsymbol{X}} - \text{E}_{\boldsymbol{O}} \tag{63}$$
$$= -2\beta^{-1}\,\log\left(\exp(-\frac{\beta}{2}\,\text{E}_{\boldsymbol{X}}) + \exp(-\beta\,\text{E}_{\boldsymbol{O}} + \frac{\beta}{2}\text{E}_{\boldsymbol{X}})\right) - \text{E}_{\boldsymbol{O}} \tag{64}$$
$$= -2\beta^{-1}\,\log\left(\exp(-\frac{\beta}{2}\,\text{E}_{\boldsymbol{X}} + \frac{\beta}{2}\,\text{E}_{\boldsymbol{O}}) + \exp(-\frac{\beta}{2}\,\text{E}_{\boldsymbol{O}} + \frac{\beta}{2}\,\text{E}_{\boldsymbol{X}})\right) \tag{65}$$
$$= -2\beta^{-1}\,\log\cosh\left(\frac{\beta}{2}(-\text{E}_{\boldsymbol{X}} + \text{E}_{\boldsymbol{O}})\right) - 2\beta^{-1}\,\log(2) \tag{66}$$
$$= -2\beta^{-1}\,\log\cosh\left(\frac{\beta}{2}\,(\text{lse}(\beta, \boldsymbol{X}^T\boldsymbol{\xi}) - \text{lse}(\beta, \boldsymbol{O}^T\boldsymbol{\xi}))\right) - 2\beta^{-1}\,\log(2) \tag{67}$$
$$= -2\beta^{-1}\,\log\cosh\left(\frac{\beta}{2}\,(\text{lse}(\beta, \boldsymbol{O}^T\boldsymbol{\xi}) - \text{lse}(\beta, \boldsymbol{X}^T\boldsymbol{\xi}))\right) - 2\beta^{-1}\,\log(2) \tag{68}$$

By exponentiation of the above result we obtain

$$f_b(\boldsymbol{\xi}) \propto \exp(\beta\text{E}_b(\boldsymbol{\xi}; \boldsymbol{X}, \boldsymbol{O})) = \frac{1}{4\cosh^2\left(\frac{\beta}{2}\,(\text{lse}(\beta, \boldsymbol{X}^T\boldsymbol{\xi}) - \text{lse}(\beta, \boldsymbol{O}^T\boldsymbol{\xi}))\right)} \tag{69}$$

The function $\log\cosh(x)$ is related to the negative log-likelihood of the hyperbolic secant distribution (see e.g. Saleh & Saleh, 2022). For values of $x$ close to 0, $\log\cosh$ can be approximated by $\frac{x^2}{2}$, and for values far from 0, the function behaves as $|x| - \log(2)$.

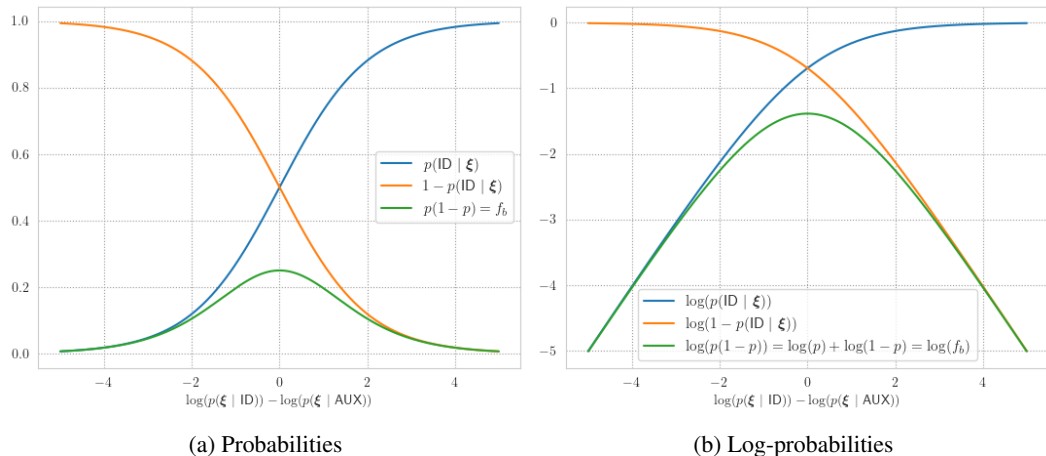

(a) Probabilities                    (b) Log-probabilities

Figure 8: The product of two logistic sigmoids yields $f_b$ (a); the sum of two log-sigmoids yields $\log(f_b) = \mathrm{E}_b$ (b).

### G.3 Derivatives of $\mathrm{E}_b$

In this section, we investigate the derivatives of the energy function $\mathrm{E}_b$. The derivative of the lse is:

$$\nabla_{\boldsymbol{z}} \, \mathrm{lse}(\beta, \boldsymbol{z}) \;=\; \nabla_{\boldsymbol{z}} \, \beta^{-1} \, \log\left(\sum_{i=1}^{N} \exp(\beta z_i)\right) \;=\; \mathrm{softmax}(\beta \, \boldsymbol{z}) \tag{70}$$

Thus, the derivative of the MHE $\mathrm{E}(\boldsymbol{\xi}; \boldsymbol{X})$ w.r.t. $\boldsymbol{\xi}$ is:

$$\nabla_{\boldsymbol{\xi}} \, \mathrm{E}(\boldsymbol{\xi}; \boldsymbol{X}) \;=\; \nabla_{\boldsymbol{\xi}} \left(-\mathrm{lse}(\beta, \boldsymbol{X}^T \boldsymbol{\xi}) + \frac{1}{2}\boldsymbol{\xi}^T\boldsymbol{\xi} + C\right) \;=\; -\boldsymbol{X}\mathrm{softmax}(\beta\boldsymbol{X}^T\boldsymbol{\xi}) + \boldsymbol{\xi} \tag{71}$$

The update rule of the MHN

$$\boldsymbol{\xi}^{t+1} \;=\; \boldsymbol{X}\mathrm{softmax}(\beta\boldsymbol{X}^T\boldsymbol{\xi}^t) \tag{72}$$

is derived via the concave-convex procedure. It coincides with the attention mechanisms of Transformers and has been proven to converge globally to stationary points of the energy $\mathrm{E}(\boldsymbol{\xi}; \boldsymbol{X})$ (Ramsauer et al., 2021). It can also be shown that the update rule emerges when performing gradient descent on $\mathrm{E}(\boldsymbol{\xi}; \boldsymbol{X})$ with step size $\eta = 1$ Park et al. (2023):

$$\boldsymbol{\xi}^{t+1} \;=\; \boldsymbol{\xi}^t \;-\; \eta \, \nabla_{\boldsymbol{\xi}}\mathrm{E}(\boldsymbol{\xi}^t; \boldsymbol{X}) \tag{73}$$
$$\boldsymbol{\xi}^{t+1} \;=\; \boldsymbol{X}\mathrm{softmax}(\beta\boldsymbol{X}^T\boldsymbol{\xi}^t) \tag{74}$$

From Equation (71), we can see that the gradient of $\mathrm{E}_b(\boldsymbol{\xi}; \boldsymbol{X}, \boldsymbol{O})$ w.r.t. $\boldsymbol{\xi}$ is:

$$\nabla_{\boldsymbol{\xi}}\mathrm{E}_b(\boldsymbol{\xi}; \boldsymbol{X}, \boldsymbol{O}) \;=\; \nabla_{\boldsymbol{\xi}} \left(-\,2\,\mathrm{lse}(\beta, (\boldsymbol{X} \,\|\, \boldsymbol{O})^T\boldsymbol{\xi}) + \mathrm{lse}(\beta, \boldsymbol{X}^T\boldsymbol{\xi}) + \mathrm{lse}(\beta, \boldsymbol{O}^T\boldsymbol{\xi})\right) \tag{75}$$
$$=\; -\,2\,(\boldsymbol{X} \,\|\, \boldsymbol{O})\,\mathrm{softmax}(\beta(\boldsymbol{X} \,\|\, \boldsymbol{O})^T\boldsymbol{\xi}) \;+\; \boldsymbol{X}\mathrm{softmax}(\beta\boldsymbol{X}^T\boldsymbol{\xi}) \;+\; \boldsymbol{O}\mathrm{softmax}(\beta\boldsymbol{O}^T\boldsymbol{\xi}) \tag{76}$$

When $\boldsymbol{X}\mathrm{softmax}(\beta\boldsymbol{X}^T\boldsymbol{\xi})$, $\boldsymbol{O}\mathrm{softmax}(\beta\boldsymbol{O}^T\boldsymbol{\xi})$, $\mathrm{lse}(\beta, \boldsymbol{X}^T\boldsymbol{\xi})$ and $\mathrm{lse}(\beta, \boldsymbol{O}^T\boldsymbol{\xi})$ are available, one can efficiently compute $(\boldsymbol{X} \,\|\, \boldsymbol{O})\,\mathrm{softmax}(\beta(\boldsymbol{X} \,\|\, \boldsymbol{O})^T\boldsymbol{\xi})$ as follows:

$$(\boldsymbol{X} \parallel \boldsymbol{O}) \operatorname{softmax}(\beta(\boldsymbol{X} \parallel \boldsymbol{O})^T \boldsymbol{\xi}) \tag{77}$$

$$= \nabla_{\boldsymbol{\xi}} \operatorname{lse}(\beta, (\boldsymbol{X} \parallel \boldsymbol{O})^T \boldsymbol{\xi}) \tag{78}$$

$$= \nabla_{\boldsymbol{\xi}} \, \beta^{-1} \log\left(\exp(\beta \operatorname{lse}(\beta, \boldsymbol{X}^T \boldsymbol{\xi})) \, + \, \exp(\beta \operatorname{lse}(\beta, \boldsymbol{O}^T \boldsymbol{\xi}))\right) \tag{79}$$

$$= \begin{pmatrix} \boldsymbol{X} \operatorname{softmax}(\beta \boldsymbol{X}^T \boldsymbol{\xi}) & \boldsymbol{O} \operatorname{softmax}(\beta \boldsymbol{O}^T \boldsymbol{\xi}) \end{pmatrix} \operatorname{softmax}\left(\beta \begin{pmatrix} \operatorname{lse}(\beta, \boldsymbol{X}^T \boldsymbol{\xi}) \\ \operatorname{lse}(\beta, \boldsymbol{O}^T \boldsymbol{\xi}) \end{pmatrix}\right) \tag{80}$$

We can also compute the gradient of $\mathrm{E}_b(\boldsymbol{\xi}; \boldsymbol{X}, \boldsymbol{O})$ w.r.t. $\boldsymbol{\xi}$ via the $\log \cosh$-representation of $\mathrm{E}_b$ (see Equation (68)). The derivative of the $\log \cosh$ function is

$$\frac{\mathrm{d}}{\mathrm{d}x} \beta^{-1} \log \cosh(\beta x) = \tanh(\beta x) \tag{81}$$

Therefore, we can compute the gradient of $\mathrm{E}_b(\boldsymbol{\xi}; \boldsymbol{X}, \boldsymbol{O})$ as

$$\nabla_{\boldsymbol{\xi}} \, \mathrm{E}_b(\boldsymbol{\xi}; \boldsymbol{X}, \boldsymbol{O}) \tag{82}$$

$$= \nabla_{\boldsymbol{\xi}} \, - \, 2\beta^{-1} \, \log \cosh\left(\frac{\beta}{2} \left(\operatorname{lse}(\beta, \boldsymbol{O}^T \boldsymbol{\xi}) \, - \, \operatorname{lse}(\beta, \boldsymbol{X}^T \boldsymbol{\xi})\right)\right) \tag{83}$$

$$= -\tanh\left(\frac{\beta}{2}(\operatorname{lse}(\beta, \boldsymbol{O}^T \boldsymbol{\xi}) \, - \, \operatorname{lse}(\beta, \boldsymbol{X}^T \boldsymbol{\xi}))\right) \left(\boldsymbol{O} \operatorname{softmax}(\beta \boldsymbol{O}^T \boldsymbol{\xi}) - \boldsymbol{X} \operatorname{softmax}(\beta \boldsymbol{X}^T \boldsymbol{\xi})\right) \tag{84}$$

$$= -\tanh\left(\frac{\beta}{2}(\operatorname{lse}(\beta, \boldsymbol{X}^T \boldsymbol{\xi}) \, - \, \operatorname{lse}(\beta, \boldsymbol{O}^T \boldsymbol{\xi}))\right) \left(\boldsymbol{X} \operatorname{softmax}(\beta \boldsymbol{X}^T \boldsymbol{\xi}) - \boldsymbol{O} \operatorname{softmax}(\beta \boldsymbol{O}^T \boldsymbol{\xi})\right) \tag{85}$$

Next, we would like to compute the gradient of $\mathrm{E}_b(\boldsymbol{\xi}; \boldsymbol{X}, \boldsymbol{O})$ w.r.t. the memory matrices $\boldsymbol{X}$ and $\boldsymbol{O}$. For this, let us first look at the gradient of the MHE $\mathrm{E}(\boldsymbol{\xi}; \boldsymbol{X})$ w.r.t. a single stored pattern $\boldsymbol{x}_i$ (where $\boldsymbol{X}$ is the matrix of concatenated stored patterns $(\boldsymbol{x}_1, \boldsymbol{x}_2, \ldots, \boldsymbol{x}_N)$):

$$\nabla_{\boldsymbol{x}_i} \mathrm{E}(\boldsymbol{\xi}; \boldsymbol{X}) = - \boldsymbol{\xi} \operatorname{softmax}(\beta \boldsymbol{X}^T \boldsymbol{\xi})_i \tag{86}$$

Thus, the gradient w.r.t. the full memory matrix $\boldsymbol{X}$ is

$$\nabla_{\boldsymbol{X}} \mathrm{E}(\boldsymbol{\xi}; \boldsymbol{X}) = -\boldsymbol{\xi} \operatorname{softmax}(\beta \boldsymbol{X}^T \boldsymbol{\xi})^T \tag{87}$$

We can now also use the $\log \cosh$ formulation of $\mathrm{E}_b(\boldsymbol{\xi}; \boldsymbol{X}, \boldsymbol{O})$ to compute the gradient of $\mathrm{E}_b(\boldsymbol{\xi}; \boldsymbol{X}, \boldsymbol{O})$, w.r.t $\boldsymbol{X}$ and $\boldsymbol{O}$:

$$\nabla_{\boldsymbol{X}} \mathrm{E}_b(\boldsymbol{\xi}; \boldsymbol{X}, \boldsymbol{O}) = \nabla_{\boldsymbol{X}} - 2\beta^{-1} \, \log \cosh\left(\frac{\beta}{2} \left(\operatorname{lse}(\beta, \boldsymbol{X}^T \boldsymbol{\xi}) \, - \operatorname{lse}(\beta, \boldsymbol{O}^T \boldsymbol{\xi})\right)\right) \tag{88}$$

$$= - \tanh\left(\frac{\beta}{2}(\operatorname{lse}(\beta, \boldsymbol{X}^T \boldsymbol{\xi}) - \operatorname{lse}(\beta, \boldsymbol{O}^T \boldsymbol{\xi}))\right) \boldsymbol{\xi} \operatorname{softmax}(\beta \boldsymbol{X}^T \boldsymbol{\xi})^T \tag{89}$$

$$\tag{90}$$

Analogously, the gradient w.r.t $\boldsymbol{O}$ is

$$\nabla_{\boldsymbol{O}} \mathrm{E}_b(\boldsymbol{\xi}; \boldsymbol{X}, \boldsymbol{O}) = - \tanh\left(\frac{\beta}{2}(\operatorname{lse}(\beta, \boldsymbol{O}^T \boldsymbol{\xi}) - \operatorname{lse}(\beta, \boldsymbol{X}^T \boldsymbol{\xi}))\right) \boldsymbol{\xi} \operatorname{softmax}(\beta \boldsymbol{O}^T \boldsymbol{\xi})^T \tag{91}$$

# H  Notes on the Relationship between Hopfield Boosting and other methods

## H.1  Relation to Radial Basis Function Networks

This section shows the relation between radial basis function networks (RBF networks; Moody & Darken, 1989) and modern Hopfield energy (following Schäfl et al., 2022). Consider an RBF network with normalized linear weights:

$$\varphi(\boldsymbol{\xi}) = \sum_{i=1}^{N} \omega_i \exp(-\frac{\beta}{2}||\boldsymbol{\xi} - \boldsymbol{\mu}_i||_2^2) \tag{92}$$

where $\beta$ denotes the inverse tied variance $\beta = \frac{1}{\sigma^2}$, and the $\omega_i$ are normalized using the softmax function:

$$\omega_i = \mathrm{softmax}(\beta\boldsymbol{a})_i = \frac{\exp(\beta a_i)}{\sum_{j=1}^{N} \exp(\beta a_j)} \tag{93}$$

An energy can be obtained by taking the negative log of $\varphi(\boldsymbol{\xi})$:

$$\mathrm{E}(\boldsymbol{\xi}) = -\beta^{-1}\log\left(\varphi(\boldsymbol{\xi})\right) \tag{94}$$

$$= -\beta^{-1}\log\left(\sum_{i=1}^{N}\omega_i\exp(-\frac{\beta}{2}||\boldsymbol{\xi}-\boldsymbol{\mu}_i||_2^2))\right) \tag{95}$$

$$= -\beta^{-1}\log\left(\sum_{i=1}^{N}\exp(\beta(-\frac{1}{2}||\boldsymbol{\xi}-\boldsymbol{\mu}_i||_2^2 + \beta^{-1}\log\mathrm{softmax}(\beta\boldsymbol{a})_i))\right) \tag{96}$$

$$= -\beta^{-1}\log\left(\sum_{i=1}^{N}\exp(\beta(-\frac{1}{2}||\boldsymbol{\xi}-\boldsymbol{\mu}_i||_2^2 + a_i - \mathrm{lse}(\beta,\boldsymbol{a})))\right) \tag{97}$$

$$= -\beta^{-1}\log\left(\sum_{i=1}^{N}\exp(\beta(-\frac{1}{2}\boldsymbol{\xi}^T\boldsymbol{\xi} + \boldsymbol{\mu}_i^T\boldsymbol{\xi} - \frac{1}{2}\boldsymbol{\mu}_i^T\boldsymbol{\mu}_i + a_i))\right) + \mathrm{lse}(\beta,\boldsymbol{a}) \tag{98}$$

Next, we define $a_i = \frac{1}{2}\boldsymbol{\mu}_i^T\boldsymbol{\mu}_i$

$$\mathrm{E}(\boldsymbol{\xi}) = -\beta^{-1}\log\left(\sum_{i=1}^{N}\exp(\beta\boldsymbol{\mu}_i^T\boldsymbol{\xi})\right) + \frac{1}{2}\boldsymbol{\xi}^T\boldsymbol{\xi} + \mathrm{lse}(\beta,\boldsymbol{a}) \tag{99}$$

Finally, we use the fact that $\mathrm{lse}(\beta,\boldsymbol{a}) \leq \max_i a_i + \beta^{-1}\log N$

$$\mathrm{E}(\boldsymbol{\xi}) = -\beta^{-1}\log\left(\sum_{i=1}^{N}\exp(\beta\boldsymbol{\mu}_i^T\boldsymbol{\xi})\right) + \frac{1}{2}\boldsymbol{\xi}^T\boldsymbol{\xi} + \beta^{-1}\log N + \frac{1}{2}M^2 \tag{100}$$

where $M = \max_i ||\boldsymbol{\mu}_i||_2$

## H.2  Contrastive Representation Learning

A commonly used loss function in contrastive representation learning (e.g., Chen et al., 2020; He et al., 2020) is the InfoNCE loss (Oord et al., 2018):

$$\mathcal{L}_{\mathrm{NCE}} = \mathop{\mathbb{E}}_{\substack{(x,y)\sim p_{\mathrm{pos}} \\ \{x_i^-\}_{i=1}^{M}\sim p_{\mathrm{data}}}}\left[-\log\frac{e^{f(x)^T f(y)/\tau}}{e^{f(x)^T f(y)/\tau} + \sum_i e^{f(x_i^-)^T f(y)/\tau}}\right] \tag{101}$$

Wang & Isola (2020) show that $\mathcal{L}_{\text{NCE}}$ optimizes two objectives:

$$\mathcal{L}_{\text{NCE}} = \underbrace{\mathbb{E}_{(x,y)\sim p_{pos}}\left[-f(x)^T f(y)/\tau\right]}_{\text{Alignment}} + \underbrace{\mathbb{E}_{\substack{(x,y)\sim p_{pos}\\ \{x_i^-\}_{i=1}^M \sim p_{data}}}\left[\log\left(e^{f(x)^T f(y)/\tau} + \sum_i e^{f(x_i^-)^T f(x)/\tau}\right)\right]}_{\text{Uniformity}}$$

(102)

Alignment enforces that features from positive pairs are similar, while uniformity encourages a uniform distribution of the samples over the hypersphere.

In comparison, our proposed loss, $\mathcal{L}_{\text{OOD}}$, does not visibly enforce alignment between samples within the same class. Instead, we can observe that it promotes uniformity to the instances of the *foreign* class. Due to the constraints that are imposed by the geometry of the space the optimization is performed on, that is, $||f(x)|| = 1$ when the samples move on a hypersphere, the loss encourages the patterns in the ID data have maximum distance to the samples of the AUX data, i.e., they concentrate on opposing poles of the hypersphere. A demonstration of this mechanism can be found in Appendix F.2 and F.3

### H.3 Support Vector Machines

In the following, we will show the relation of Hopfield Boosting to support vector machines (SVMs; Cortes & Vapnik, 1995) with RBF kernel. We adopt and expand the arguments of Schäfl et al. (2022).

Assume we apply an SVM with RBF kernel to model the decision boundary between ID and AUX data. We train on the features $\boldsymbol{Z} = (\boldsymbol{x}_1, \ldots, \boldsymbol{x}_N, \boldsymbol{o}_1, \ldots, \boldsymbol{o}_M)$ and assume that the patterns are normalized, i.e., $||\boldsymbol{x}_i||_2 = 1$ and $||\boldsymbol{o}_i||_2 = 1$. We define the targets $(y_1, \ldots, y_{(N+M)})$ as $1$ for ID and $-1$ for AUX data. The decision rule of the SVM equates to

$$\hat{B}(\boldsymbol{\xi}) = \begin{cases} \text{ID} & \text{if } s(\boldsymbol{\xi}) \geq 0 \\ \text{OOD} & \text{if } s(\boldsymbol{\xi}) < 0 \end{cases}$$

(103)

where

$$s(\boldsymbol{\xi}) = \sum_{i=1}^{N+M} \alpha_i y_i k(\boldsymbol{z}_i, \boldsymbol{\xi})$$

(104)

$$k(\boldsymbol{z}_i, \boldsymbol{\xi}) = \exp\left(-\frac{\beta}{2}||\boldsymbol{\xi} - \boldsymbol{z}_i||_2^2\right)$$

(105)

We assume that there is at least one support vector for both ID and AUX data, i.e., there exists at least one index $i$ s.t. $\alpha_i y_i > 0$ and at least one index $j$ s.t. $\alpha_j y_j < 0$. We now split the samples $\boldsymbol{z}_i$ in $s(\boldsymbol{\xi})$ according to their label:

$$s(\boldsymbol{\xi}) = \sum_{i=1}^{N} \alpha_i k(\boldsymbol{x}_i, \boldsymbol{\xi}) - \sum_{i=1}^{M} \alpha_{N+i} k(\boldsymbol{o}_i, \boldsymbol{\xi})$$

(106)

We define an alternative score:

$$s_{\text{frac}}(\boldsymbol{\xi}) = \frac{\sum_{i=1}^{N} \alpha_i k(\boldsymbol{x}_i, \boldsymbol{\xi})}{\sum_{i=1}^{M} \alpha_{N+i} k(\boldsymbol{o}_i, \boldsymbol{\xi})}$$

(107)

(108)

Because we assumed there is at least one support vector for both ID and AUX data and as the $\alpha_i$ are constrained to be non-negative and because $k(\cdot, \cdot) > 0$, the numerator and denominator are strictly positive. We can, therefore, specify a new decision rule $\hat{B}_{\mathrm{frac}}(\boldsymbol{\xi})$.

$$\hat{B}_{\mathrm{frac}}(\boldsymbol{\xi}) = \begin{cases} \mathrm{ID} & \text{if } s_{\mathrm{frac}}(\boldsymbol{\xi}) \geq 1 \\ \mathrm{OOD} & \text{if } s_{\mathrm{frac}}(\boldsymbol{\xi}) < 1 \end{cases} \tag{109}$$

Although the functions $s(\boldsymbol{\xi})$ and $s_{\mathrm{frac}}(\boldsymbol{\xi})$ are different, the decision rules $\hat{B}(\boldsymbol{\xi})$ and $\hat{B}_{\mathrm{frac}}(\boldsymbol{\xi})$ are equivalent. Another possible pair of score and decision rule is the following:

$$s_{\log}(\boldsymbol{\xi}) = \beta^{-1} \log(s_{\mathrm{frac}}(\boldsymbol{\xi})) = \beta^{-1} \log\left(\sum_{i=1}^{N} \alpha_i k(\boldsymbol{x}_i, \boldsymbol{\xi})\right) - \beta^{-1} \log\left(\sum_{i=1}^{M} \alpha_{N+i} k(\boldsymbol{o}_i, \boldsymbol{\xi})\right) \tag{110}$$

$$\hat{B}_{\log}(\boldsymbol{\xi}) = \begin{cases} \mathrm{ID} & \text{if } s_{\log}(\boldsymbol{\xi}) \geq 0 \\ \mathrm{OOD} & \text{if } s_{\log}(\boldsymbol{\xi}) < 0 \end{cases} \tag{111}$$

Let us more closely examine the term $\beta^{-1} \log\left(\sum_{i=1}^{N} \alpha_i k(\boldsymbol{x}_i, \boldsymbol{\xi})\right)$. We define $a_i = \beta^{-1} \log(\alpha_i)$.

$$\beta^{-1} \log\left(\sum_{i=1}^{N} \alpha_i k(\boldsymbol{x}_i, \boldsymbol{\xi})\right) = \beta^{-1} \log\left(\sum_{i=1}^{N} \exp(\beta a_i) \exp\left(-\frac{\beta}{2}\|\boldsymbol{\xi} - \boldsymbol{x}_i\|_2^2\right)\right) \tag{112}$$

$$= \beta^{-1} \log\left(\sum_{i=1}^{N} \exp(\beta a_i) \exp\left(-\frac{\beta}{2}\boldsymbol{\xi}^T\boldsymbol{\xi} + \beta\boldsymbol{x}_i^T\boldsymbol{\xi} - \frac{\beta}{2}\boldsymbol{x}_i^T\boldsymbol{x}_i\right)\right) \tag{113}$$

$$= \beta^{-1} \log\left(\sum_{i=1}^{N} \exp\left(-\frac{\beta}{2}\boldsymbol{\xi}^T\boldsymbol{\xi} + \beta\boldsymbol{x}_i^T\boldsymbol{\xi} - \frac{\beta}{2}\boldsymbol{x}_i^T\boldsymbol{x}_i + \beta a_i\right)\right) \tag{114}$$

$$= \beta^{-1} \log\left(\sum_{i=1}^{N} \exp\left(\beta\boldsymbol{x}_i^T\boldsymbol{\xi} + \beta a_i\right)\right) - \frac{1}{2}\boldsymbol{\xi}^T\boldsymbol{\xi} - \frac{1}{2} \tag{115}$$

We now construct a memory $\boldsymbol{X}_H$ and query $\boldsymbol{\xi}_H$ such that we can compute (115) using the MHE (Equation (5)):

$$\boldsymbol{X}_H = \begin{pmatrix} \boldsymbol{x}_1 & \cdots & \boldsymbol{x}_N \\ a_1 & \cdots & a_N \end{pmatrix} \tag{116}$$

$$\boldsymbol{\xi}_H = \begin{pmatrix} \boldsymbol{\xi} \\ 1 \end{pmatrix} \tag{117}$$

We obtain

$$\mathrm{E}(\boldsymbol{\xi}_H; \boldsymbol{X}_H) = -\mathrm{lse}(\beta, \boldsymbol{X}_H^T\boldsymbol{\xi}_H) + \frac{1}{2}\boldsymbol{\xi}_H^T\boldsymbol{\xi}_H + C \tag{118}$$

$$= -\beta^{-1} \log\left(\sum_{i=1}^{N} \exp\left(\beta\boldsymbol{x}_i^T\boldsymbol{\xi} + 1\beta a_i\right)\right) + \frac{1}{2}\boldsymbol{\xi}^T\boldsymbol{\xi} + \frac{1}{2}\cdot 1^2 + C \tag{119}$$

$$= -\beta^{-1} \log\left(\sum_{i=1}^{N} \exp\left(\beta\boldsymbol{x}_i^T\boldsymbol{\xi} + \beta a_i\right)\right) + \frac{1}{2}\boldsymbol{\xi}^T\boldsymbol{\xi} + \frac{1}{2} + C \tag{120}$$

$$= -\beta^{-1} \log\left(\sum_{i=1}^{N} \alpha_i k(\boldsymbol{x}_i, \boldsymbol{\xi})\right) + C \tag{121}$$

We construct $\boldsymbol{O}_H$ analogously to Equation (116) and thus can compute

$$s_{\log}(\boldsymbol{\xi}) \;=\; \mathrm{E}(\boldsymbol{\xi}_H; \boldsymbol{O}_H) \;-\; \mathrm{E}(\boldsymbol{\xi}_H; \boldsymbol{X}_H) \;=\; \mathrm{lse}(\beta, \boldsymbol{X}_H^T \boldsymbol{\xi}_H) \;-\; \mathrm{lse}(\beta, \boldsymbol{O}_H^T \boldsymbol{\xi}_H) \qquad (122)$$

which is exactly the score Hopfield Boosting uses for determining whether a sample is OOD (Equation (13)). In contrast to SVMs, Hopfield Boosting uses a uniform weighting of the patterns in the memory when computing the score. However, Hopfield Boosting can emulate a weighting of the patterns by more frequently sampling patterns with high weights into the memory.

### H.4 HE and SHE

Zhang et al. (2023a) introduce two post-hoc methods for OOD detection using MHE, which are called "Hopfield Energy" (HE) and "Simplified Hopfield Energy" (SHE). Like Hopfield Boosting, HE and SHE both employ the MHE to determine whether a sample is ID or OOD. However, unlike Hopfield Boosting, HE and SHE offer no possibility to include AUX data in the training process to improve the OOD detection performance of their method. The rest of this section is structured as follows: First, we briefly introduce the methods HE and SHE, second, we formally analyze the two methods, and third, we relate them to Hopfield Boosting.

**Hopfield Energy (HE)** The method HE (Zhang et al., 2023a) computes the OOD score $s_{\mathrm{HE}}(\boldsymbol{\xi})$ as follows:

$$s_{\mathrm{HE}}(\boldsymbol{\xi}) \;=\; \mathrm{lse}(\beta, \boldsymbol{X}_c^T \boldsymbol{\xi}) \qquad (123)$$

where $\boldsymbol{X}_c \in \mathbb{R}^{d \times N_c}$ denotes the memory $(\boldsymbol{x}_{c1}, \dots, \boldsymbol{x}_{cN_c})$ containing $N_c$ encoded data instances of class $c$. HE uses the prediction of the ID classification head to determine which patterns to store in the Hopfield memory:

$$c \;=\; \underset{y}{\mathrm{argmax}}\; p(\, y \mid \boldsymbol{\xi}^{\mathcal{D}}\,) \qquad (124)$$

**Simplified Hopfield Energy (SHE)** The method SHE (Zhang et al., 2023a) employs a simplified score $s_{\mathrm{SHE}}(\boldsymbol{\xi})$:

$$s_{\mathrm{SHE}}(\boldsymbol{\xi}) \;=\; \boldsymbol{m}_c^T \boldsymbol{\xi} \qquad (125)$$

where $\boldsymbol{m}_c \in \mathbb{R}^d$ denotes the mean of the patterns in memory $\boldsymbol{X}_c$:

$$\boldsymbol{m}_c \;=\; \frac{1}{N_c} \sum_{i=1}^{N_c} \boldsymbol{x}_{ci} \qquad (126)$$

**Relation between HE and SHE** In the following, we show a simple yet enlightening relation between the scores $s_{\mathrm{HE}}$ and $s_{\mathrm{SHE}}$. For mathematical convenience, we first slightly modify the score $s_{\mathrm{HE}}$:

$$s_{\mathrm{HE}}(\boldsymbol{\xi}) \;=\; \mathrm{lse}(\beta, \boldsymbol{X}_c^T \boldsymbol{\xi}) \;-\; \beta^{-1} \log N_c \qquad (127)$$

All data sets which were employed in the experiments of Zhang et al. (2023a) (CIFAR-10 and CIFAR-100) are class-balanced. Therefore, the additional term $\beta^{-1} \log N_c$ does not change the result of the OOD detection on those data sets, as it only amounts to the same constant offset for all classes.

The function

$$\text{lse}(\beta, \boldsymbol{z}) - \beta^{-1} \log N \;=\; \beta^{-1} \log \left( \frac{1}{N} \sum_{i=1}^{N} \exp(\beta z_i) \right) \tag{128}$$

converges to the mean function as $\beta \to 0$:

$$\lim_{\beta \to 0} \left( \text{lse}(\beta, \boldsymbol{z}) - \beta^{-1} \log N \right) \;=\; \frac{1}{N} \sum_{i=1}^{N} z_i \tag{129}$$

We now investigate the behavior of $s_{\text{HE}}$ in this limit:

$$\lim_{\beta \to 0} \left( \text{lse}(\beta, \boldsymbol{X}_c^T \boldsymbol{\xi}) \;-\; \beta^{-1} \log N \right) = \frac{1}{N} \sum_{i=1}^{N} (\boldsymbol{x}_{ci}^T \boldsymbol{\xi}) \tag{130}$$

$$= \left( \frac{1}{N} \sum_{i=1}^{N} \boldsymbol{x}_{ci} \right)^T \boldsymbol{\xi} \tag{131}$$

$$= \boldsymbol{m}_c^T \boldsymbol{\xi} \tag{132}$$

where

$$\boldsymbol{m}_c \;=\; \frac{1}{N} \sum_{i=1}^{N} \boldsymbol{x}_{ci} \tag{133}$$

Therefore, we have shown that

$$\lim_{\beta \to 0} s_{\text{HE}}(\boldsymbol{\xi}) = s_{\text{SHE}}(\boldsymbol{\xi}) \tag{134}$$

**Relation of HE and SHE to Hopfield Boosting.** In contrast to HE and SHE, Hopfield Boosting uses an AUX data set to learn a decision boundary between the ID and OOD regions during the training process. To do this, our work introduces a novel MHE-based energy function, $\text{E}_b(\boldsymbol{\xi}; \boldsymbol{X}, \boldsymbol{O})$, to determine how close a sample is to the learnt decision boundary. Hopfield Boosting uses this energy function to frequently sample weak learners into the Hopfield memory and for computing a novel Hopfield-based OOD loss $\mathcal{L}_{\text{OOD}}$. To the best our knowledge, we are the first to use MHE in this way to train a neural network.

The OOD detection score of Hopfield Boosting is

$$s(\boldsymbol{\xi}) \;=\; \text{lse}(\beta, \boldsymbol{X}^T \boldsymbol{\xi}) \;-\; \text{lse}(\beta, \boldsymbol{O}^T \boldsymbol{\xi}). \tag{135}$$

where $\boldsymbol{X} \in \mathbb{R}^{d \times N}$ contains the full encoded training set $(\boldsymbol{x}_1, \ldots, \boldsymbol{x}_N)$ of all classes and $\boldsymbol{O} \in \mathbb{R}^{d \times M}$ contains AUX samples. While certainly similar to $s_{\text{HE}}$, the Hopfield Boosting score $s$ differs from $s_{\text{HE}}$ in three crucial aspects:

1. Hopfield Boosting uses AUX data samples in the OOD detection score in order to create a sharper decision boundary between the ID and OOD regions.

2. Hopfield Boosting normalizes the patterns in the memories $\boldsymbol{X}$ and $\boldsymbol{O}$ and the query $\boldsymbol{\xi}$ to unit length, while HE and SHE use unnormalized patterns to construct their memories $\boldsymbol{X}_c$ and their query pattern $\boldsymbol{\xi}$.

3. The score of Hopfield Boosting, $s(\boldsymbol{\xi})$, contains the full encoded training data set, while $s_{\text{HE}}$ only contains the patterns of a single class. Therefore Hopfield Boosting computes the similarities of a query sample $\boldsymbol{\xi}$ to the entire ID data set. In Appendix I.8, we show that this process only incurs a moderate overhead of $7.5\%$ compared to the forward pass of the ResNet-18.

The selection of the score function $s(\boldsymbol{\xi})$ is only a small aspect of Hopfield Boosting. Hopfield Boosting additionally samples informative AUX data close to the decision boundary, optimizes an MHE-based loss function, and thereby learns a sharp decision boundary between ID and OOD regions. Those three aspects are novel contributions of Hopfield Boosting. In contrast, the work of Zhang et al. (2023a) solely focuses on the selection of a suitable Hopfield-based OOD detection score for post-hoc OOD detection.

Table 4: OOD detection performance on CIFAR-100. We compare results from Hopfield Boosting, DOS (Jiang et al., 2024), DOE (Wang et al., 2023b), DivOE (Zhu et al., 2023), DAL (Wang et al., 2023a), MixOE (Zhang et al., 2023b), POEM (Ming et al., 2022), EBO-OE (Liu et al., 2020), and MSP-OE (Hendrycks et al., 2019b) on ResNet-18. ↓ indicates "lower is better" and ↑ "higher is better". All values in %. Standard deviations are estimated across five training runs.

| | Metric | HB (ours) | DOS | DOE | DivOE | DAL | MixOE | POEM | EBO-OE | MSP-OE |
|---|---|---|---|---|---|---|---|---|---|---|
| SVHN | FPR95 ↓ | $13.27^{\pm 5.46}$ | $\mathbf{9.84}^{\pm \mathbf{2.75}}$ | $19.38^{\pm 4.60}$ | $28.77^{\pm 5.42}$ | $19.95^{\pm 2.34}$ | $41.54^{\pm 13.16}$ | $33.59^{\pm 4.12}$ | $36.33^{\pm 2.95}$ | $19.86^{\pm 6.90}$ |
| | AUROC ↑ | $97.07^{\pm 0.81}$ | $\mathbf{97.64}^{\pm \mathbf{0.39}}$ | $95.72^{\pm 1.12}$ | $94.25^{\pm 0.98}$ | $95.69^{\pm 0.66}$ | $92.27^{\pm 2.71}$ | $94.06^{\pm 0.51}$ | $92.93^{\pm 0.72}$ | $95.74^{\pm 1.60}$ |
| LSUN-Crop | FPR95 ↓ | $\mathbf{12.68}^{\pm \mathbf{2.38}}$ | $19.40^{\pm 2.45}$ | $28.23^{\pm 2.69}$ | $35.10^{\pm 4.23}$ | $24.24^{\pm 2.12}$ | $23.10^{\pm 7.39}$ | $15.72^{\pm 3.46}$ | $21.06^{\pm 3.12}$ | $32.88^{\pm 1.28}$ |
| | AUROC ↑ | $\mathbf{96.54}^{\pm \mathbf{0.65}}$ | $96.42^{\pm 0.35}$ | $93.79^{\pm 0.88}$ | $92.45^{\pm 0.94}$ | $95.04^{\pm 0.43}$ | $96.11^{\pm 1.09}$ | $\mathbf{96.85}^{\pm \mathbf{0.60}}$ | $95.79^{\pm 0.62}$ | $92.85^{\pm 0.33}$ |
| LSUN-Resize | FPR95 ↓ | $\mathbf{0.00}^{\pm \mathbf{0.00}}$ | $0.01^{\pm 0.00}$ | $0.05^{\pm 0.04}$ | $0.01^{\pm 0.00}$ | $\mathbf{0.00}^{\pm \mathbf{0.00}}$ | $10.27^{\pm 10.72}$ | $\mathbf{0.00}^{\pm \mathbf{0.00}}$ | $\mathbf{0.00}^{\pm \mathbf{0.00}}$ | $0.03^{\pm 0.01}$ |
| | AUROC ↑ | $99.98^{\pm 0.01}$ | $99.96^{\pm 0.02}$ | $99.99^{\pm 0.01}$ | $\mathbf{99.99}^{\pm \mathbf{0.00}}$ | $99.94^{\pm 0.02}$ | $97.99^{\pm 1.92}$ | $99.57^{\pm 0.09}$ | $99.57^{\pm 0.03}$ | $99.97^{\pm 0.00}$ |
| Textures | FPR95 ↓ | $\mathbf{2.35}^{\pm \mathbf{0.13}}$ | $6.02^{\pm 0.52}$ | $19.42^{\pm 1.58}$ | $11.52^{\pm 0.49}$ | $5.22^{\pm 0.39}$ | $28.99^{\pm 6.79}$ | $2.89^{\pm 0.32}$ | $5.07^{\pm 0.54}$ | $10.34^{\pm 0.40}$ |
| | AUROC ↑ | $\mathbf{99.22}^{\pm \mathbf{0.02}}$ | $98.33^{\pm 0.11}$ | $94.93^{\pm 0.48}$ | $97.02^{\pm 0.08}$ | $98.50^{\pm 0.16}$ | $94.24^{\pm 1.21}$ | $98.97^{\pm 0.08}$ | $98.15^{\pm 0.16}$ | $97.42^{\pm 0.08}$ |
| iSUN | FPR95 ↓ | $\mathbf{0.00}^{\pm \mathbf{0.00}}$ | $0.03^{\pm 0.01}$ | $0.01^{\pm 0.02}$ | $0.06^{\pm 0.01}$ | $0.01^{\pm 0.02}$ | $14.40^{\pm 13.48}$ | $\mathbf{0.00}^{\pm \mathbf{0.00}}$ | $\mathbf{0.00}^{\pm \mathbf{0.00}}$ | $0.08^{\pm 0.02}$ |
| | AUROC ↑ | $99.98^{\pm 0.01}$ | $99.95^{\pm 0.02}$ | $\mathbf{99.99}^{\pm \mathbf{0.00}}$ | $99.97^{\pm 0.00}$ | $99.93^{\pm 0.02}$ | $97.23^{\pm 2.59}$ | $99.59^{\pm 0.09}$ | $99.57^{\pm 0.03}$ | $99.96^{\pm 0.01}$ |
| Places 365 | FPR95 ↓ | $19.36^{\pm 1.02}$ | $32.13^{\pm 1.55}$ | $58.68^{\pm 4.15}$ | $44.20^{\pm 0.95}$ | $33.43^{\pm 1.11}$ | $47.01^{\pm 6.41}$ | $\mathbf{18.39}^{\pm \mathbf{0.68}}$ | $26.68^{\pm 2.18}$ | $45.96^{\pm 0.85}$ |
| | AUROC ↑ | $\mathbf{95.85}^{\pm \mathbf{0.37}}$ | $91.73^{\pm 0.39}$ | $83.47^{\pm 1.55}$ | $88.28^{\pm 0.26}$ | $91.10^{\pm 0.29}$ | $89.20^{\pm 1.86}$ | $95.03^{\pm 0.71}$ | $91.35^{\pm 0.70}$ | $87.77^{\pm 0.15}$ |
| Mean | FPR95 ↓ | $\mathbf{7.94}$ | 11.24 | 20.96 | 19.94 | 13.81 | 27.55 | 11.76 | 14.86 | 18.19 |
| | AUROC ↑ | $\mathbf{98.11}$ | 97.34 | 94.65 | 95.33 | 96.70 | 94.51 | 97.34 | 96.23 | 95.62 |
| ID | Accuracy ↑ | $75.08^{\pm 0.46}$ | $75.72^{\pm 0.26}$ | $\mathbf{76.96}^{\pm \mathbf{0.33}}$ | $\mathbf{76.91}^{\pm \mathbf{0.30}}$ | $\mathbf{77.29}^{\pm \mathbf{0.14}}$ | $\mathbf{79.20}^{\pm \mathbf{2.99}}$ | $66.38^{\pm 0.85}$ | $69.07^{\pm 0.32}$ | $\mathbf{76.87}^{\pm \mathbf{0.39}}$ |

# I  Additional Experiments & Experimental Details

## I.1  Results on CIFAR-100

When applying Hopfield Boosting on CIFAR-100 (Table 4), Hopfield Boosting surpasses POEM (the previously best method), improving the mean FPR95 from 11.76 to 7.95. On the SVHN data set, Hopfield Boosting improves the FPR95 metric the most, decreasing it from 33.59 to 13.27. For the LSUN-Resize and iSUN data sets, we observe a similar behavior as we saw in our CIFAR-10 evaluation — almost all methods achieve a perfect result with regard to the FPR95 metric.

## I.2  Pre-Processing and Transformations

For evaluating OOD detection methods, consistent pre-processing and image transformation is crucial. An inconsistent application of image transformations will skew results when comparing different OOD detection methods. We, therefore, apply the same pre-processing steps and transformations to all OOD detection methods we compare.

**CIFAR-10 & CIFAR-100.**    For CIFAR-10 and CIFAR-100, we apply the following transformations:

1. RandomCrop (32x32, padding 4)
2. RandomHorizontalFlip

**ImageNet-RC.**    For ImageNet-RC (used as AUX data set for the ID data sets CIFAR-10 and CIFAR-100), we apply the following transformations:

1. RandomCrop (32x32)
2. RandomCrop (32x32, padding 4)
3. RandomHorizontalFlip

**ImageNet-1K.**    For ImageNet-1K, we apply the following transformations. We closely follow the transformations used in the experiments of Zhu et al. (2023):

1. Resize (224x224)
2. RandomCrop (224x224, padding 4)
3. RandomHorizontalFlip

Table 5: Comparison between HE, SHE and our version. ↓ indicates "lower is better" and ↑ indicates "higher is better".

| | Ours | | HE | | SHE | |
|---|---|---|---|---|---|---|
| OOD Dataset | FPR95 ↓ | AUROC ↑ | FPR95 ↓ | AUROC ↑ | FPR95 ↓ | AUROC ↑ |
| SVHN | 36.79 | **93.18** | 35.81 | 92.35 | **35.07** | 92.81 |
| LSUN-Crop | **13.10** | **97.25** | 17.74 | 95.96 | 18.19 | 96.10 |
| LSUN-Resize | **16.65** | **96.84** | 20.69 | 95.87 | 21.66 | 95.85 |
| Textures | **44.54** | **89.38** | 46.29 | 86.67 | 46.19 | 87.44 |
| iSUN | **19.20** | **96.08** | 22.52 | 95.08 | 23.25 | 95.06 |
| Places 365 | **39.02** | **90.63** | 41.56 | 88.41 | 42.57 | 88.38 |
| **Mean** | **28.21** | **93.89** | 30.77 | 92.39 | 31.66 | 92.60 |

**ImageNet-21K.** For ImageNet-21K (used as AUX data set for the ID data sets ImageNet-1K), we apply the following transformations. We closely follow the transformations used in the experiments of Zhu et al. (2023):

1. RandAugment (Cubuk et al., 2020)
2. Resize (224x224)
3. RandomCrop (224x224, padding 4)
4. RandomHorizontalFlip

## I.3   Comparison HE/SHE

Since Hopfield Boosting shares similarities with the MHE-based methods HE and SHE (Zhang et al., 2023a), we also looked at the approach as used for their methods. We use the same ResNet-18 as a backbone network as we used in the experiments for Hopfield Boosting, but train it on CIFAR-10 without OE. We modify the approach of Zhang et al. (2023a) to not only use the penultimate layer, but perform a search over all layer activation combinations of the backbone for the best-performing combination. We also do not use the classifier to separate by class. From the search, we see that the concatenated activations of layers 3 and 5 give the best performance on average, so we use this setting. We experience a quite noticeable drop in performance compared to their results (Table 5). Since the computation of the MHE is the same, we assume the reason for the performance drop is the different training of the ResNet-18 backbone network, where (Zhang et al., 2023a) used strong augmentations.

## I.4   Ablations

We investigate the impact of different encoder backbone architectures on OOD detection performance with Hopfield Boosting. The baseline uses a ResNet-18 as the encoder architecture. For the ablation, the following architectures are used as a comparison: ResNet-34, ResNet-50, and Densenet-100. It can be observed, that the larger architectures lead to a slight increase in OOD performance (Table 6). We also see that a change in architecture from ResNet to Densenet leads to a different OOD behavior: The result on the Places365 data set is greatly improved, while the performance on SVHN is noticeably worse than on the ResNet architectures. The FPR95 of Densenet on SVHN also shows a high variance, which is due to one of the five independent training runs performing very badly at detecting SVHN samples as OOD: The worst run scores an FPR95 5.59, while the best run achieves an FPR95 of 0.24.

## I.5   Effect on Learned Representation

In order to analyze the impact of Hopfield Boosting on learned representations, we utilize the output of our model's embedding layer (see 4.2) as the input for a manifold learning-based visualization. Uniform Manifold Approximation and Projection (UMAP) McInnes et al. (2018) is a non-linear

Table 6: Comparison of OOD detection performance on CIFAR-10 of Hopfield Boosting on different encoders. ↓ indicates "lower is better" and ↑ indicates "higher is better". Standard deviations are estimated across five independent training runs.

| OOD Dataset | ResNet-18 | | ResNet-34 | | ResNet-50 | | Densenet-100 | |
|---|---|---|---|---|---|---|---|---|
| | FPR95 ↓ | AUROC ↑ | FPR95 ↓ | AUROC ↑ | FPR95 ↓ | AUROC ↑ | FPR95 ↓ | AUROC ↑ |
| SVHN | $0.23^{\pm0.08}$ | $99.57^{\pm0.06}$ | $0.33^{\pm0.25}$ | $99.63^{\pm0.07}$ | $\mathbf{0.19^{\pm0.09}}$ | $\mathbf{99.64^{\pm0.11}}$ | $2.11^{\pm2.76}$ | $99.31^{\pm0.35}$ |
| LSUN-Crop | $0.82^{\pm0.20}$ | $99.40^{\pm0.05}$ | $0.65^{\pm0.14}$ | $99.54^{\pm0.07}$ | $0.69^{\pm0.15}$ | $99.47^{\pm0.09}$ | $\mathbf{0.40^{\pm0.23}}$ | $\mathbf{99.52^{\pm0.09}}$ |
| LSUN-Resize | $\mathbf{0.00^{\pm0.00}}$ | $99.98^{\pm0.02}$ | $\mathbf{0.00^{\pm0.00}}$ | $99.89^{\pm0.04}$ | $\mathbf{0.00^{\pm0.00}}$ | $99.93^{\pm0.10}$ | $\mathbf{0.00^{\pm0.00}}$ | $\mathbf{100.0^{\pm0.00}}$ |
| Textures | $0.16^{\pm0.02}$ | $99.85^{\pm0.01}$ | $0.15^{\pm0.07}$ | $99.89^{\pm0.04}$ | $0.16^{\pm0.07}$ | $99.83^{\pm0.01}$ | $\mathbf{0.08^{\pm0.03}}$ | $\mathbf{99.88^{\pm0.01}}$ |
| iSUN | $\mathbf{0.00^{\pm0.00}}$ | $99.97^{\pm0.02}$ | $\mathbf{0.00^{\pm0.00}}$ | $99.98^{\pm0.02}$ | $\mathbf{0.00^{\pm0.00}}$ | $99.98^{\pm0.02}$ | $\mathbf{0.00^{\pm0.00}}$ | $\mathbf{99.99^{\pm0.01}}$ |
| Places 365 | $4.28^{\pm0.26}$ | $98.51^{\pm0.11}$ | $4.13^{\pm0.54}$ | $98.46^{\pm0.22}$ | $4.75^{\pm0.45}$ | $98.71^{\pm0.05}$ | $\mathbf{2.56^{\pm0.20}}$ | $\mathbf{99.26^{\pm0.03}}$ |
| **Mean** | 0.92 | 99.55 | 0.88 | 99.57 | 0.97 | 99.59 | **0.86** | **99.66** |

|    (a) without Hopfield Boosting    |    (b) with Hopfield Boosting    |

Figure 9: UMAP embeddings of ID (CIFAR-10) and OOD (AUX and SVHN) data based on our model trained without (a) and with Hopfield Boosting (b). Clearly, without Hopfield Boosting, the embedded OOD data points tend to overlap with the ID data points, making it impossible to distinguish between ID and OOD. On the other hand, Hopfield Boosting shows a clear separation of ID and OOD data in the embedding.

dimensionality reduction technique known for its ability to preserve both global and local structure in high-dimensional data.

First, we train two models – with and without Hopfield Boosting– and extract the embeddings of both ID and OOD data sets from them. This results in a 512-dimensional vector representation for each data point, which we further reduce to two dimensions with UMAP. The training data for UMAP always corresponds to the training data of the respective method. That is, the model trained without Hopfield Boosting is solely trained on CIFAR-10 data, and the model trained with Hopfield Boosting is presented with CIFAR-10 and AUX data during training, respectively. We then compare the learned representations concerning ID and OOD data.

Figure 9 shows the UMAP embeddings of ID (CIFAR-10) and OOD (AUX and SVHN) data based on our model trained without (a) and with Hopfield Boosting (b). Without Hopfield Boosting, OOD data points typically overlap with ID data points, with just a few exceptions, making it difficult to differentiate between them. Conversely, Hopfield Boosting allows to distinctly separate ID and OOD data in the embedding.

## I.6 OOD Examples from the Places 365 Data Set with High Semantic Similarity to CIFAR-10

We observe that Hopfield Boosting and all competing methods struggle with correctly classifying the samples from the Places 365 data set as OOD the most. Table 1 shows that for Hopfield Boosting, the FPR95 for the Places 365 data set with CIFAR-10 as the ID data set is at 4.28. The second worst FPR95 for Hopfield Boosting was measured on the LSUN-Crop data set at 0.82.

We inspect the 100 images from Places 365 that perform worst (i.e., that achieve the highest score $s(\boldsymbol{\xi})$) on a model trained with Hopfield Boosting on the CIFAR-10 data set as the in-distribution data set. Figure 10 shows that within those 100 images, the Places 365 data set contains a non-negligible amount of data instances that show objects from semantic classes contained in CIFAR-10 (e.g., horses, automobiles, dogs, trucks, and airplanes). We argue that data instances that clearly show objects of semantic classes contained in CIFAR-10 should be considered as in-distribution, which Hopfield Boosting correctly recognizes. Therefore, a certain amount of error can be anticipated on the Places 365 data set for all OOD detection methods. We leave a closer evaluation of the amount of the anticipated error up to future work.

For comparison, Figure 11 shows the 100 images from Places 365 with the lowest score $s(\boldsymbol{\xi})$, as evaluated by a model trained with Hopfield Boosting on CIFAR-10. There are no objects visible that have clear semantic overlap with the CIFAR-10 classes.

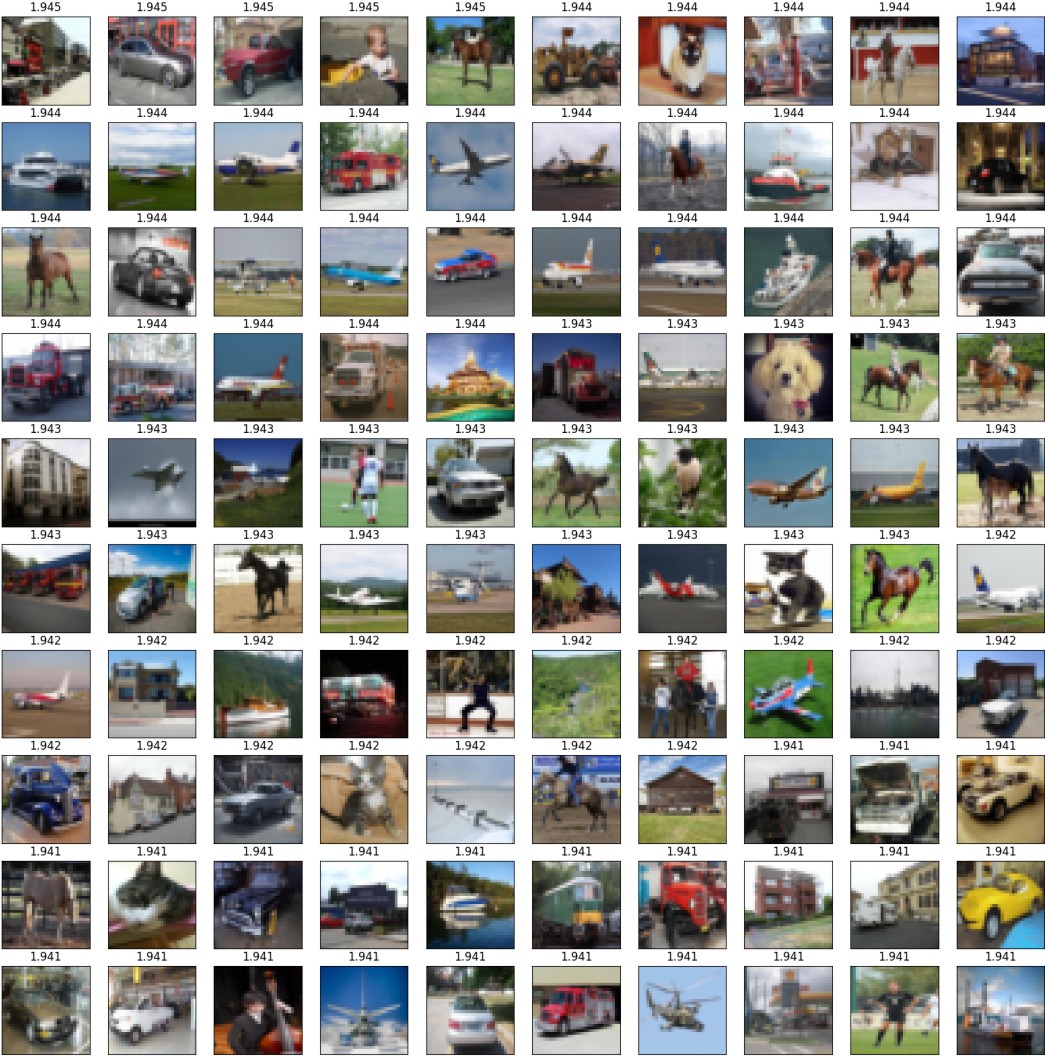

Figure 10: The set of top-100 images from the Places 365 data set which Hopfield Boosting recognized as in-distribution. The image captions show $s(\boldsymbol{\xi})$ of the respective image below the caption.

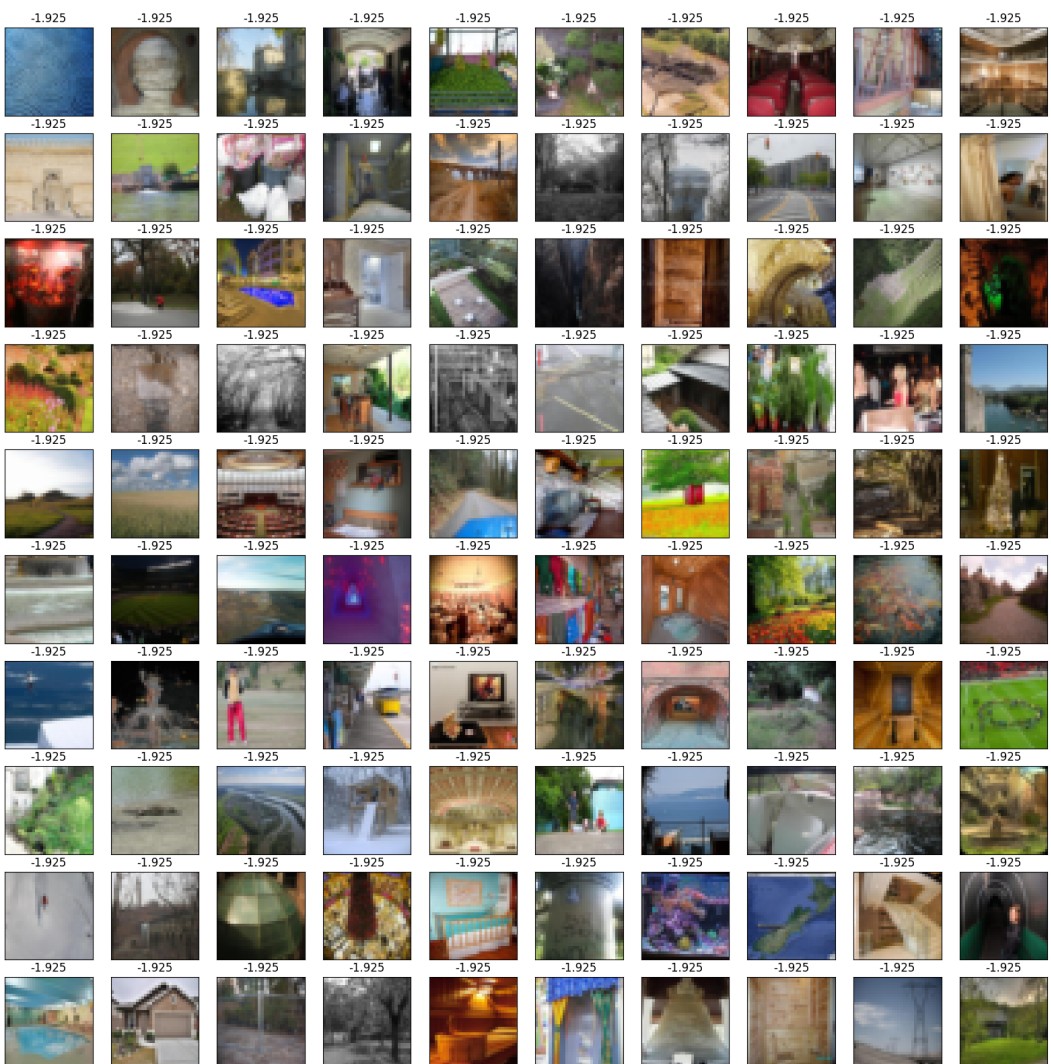

Figure 11: The set of top-100 images from the Places 365 data set which Hopfield Boosting recognized as out-of-distribution. The image captions show $s(\boldsymbol{\xi})$ of the respective image below the caption.

## I.7 Results on Noticeably Different Data Sets

The choice of additional data sets should not be driven by a desire to showcase good performance; rather, we suggest opting for data that highlights weaknesses, as it holds the potential to drive investigations and uncover novel insights. Simple toy data is preferable due to its typically clearer and more intuitive characteristics compared to complex natural image data. In alignment with these considerations, the following data sets captivated our interest: iCartoonFace (Zheng et al., 2020), Four Shapes (smeschke, 2018), and Retail Product Checkout (RPC) (Wei et al., 2022b). In Figure 12, we show random samples from these data sets to demonstrate the noticeable differences compared to CIFAR-10.

Table 7: Comparison between EBO-OE (Liu et al., 2020) and our version. ↓ indicates "lower is better" and ↑ indicates "higher is better".

| OOD Dataset | Hopfield Boosting | | EBO-OE | |
|---|---|---|---|---|
| | FPR95 ↓ | AUROC ↑ | FPR95 ↓ | AUROC ↑ |
| iCartoonFace | **0.60** | **99.57** | 4.01 | 98.94 |
| Four Shapes | **40.81** | **90.53** | 62.55 | 75.34 |
| RPC | **4.07** | **98.65** | 18.51 | 96.10 |

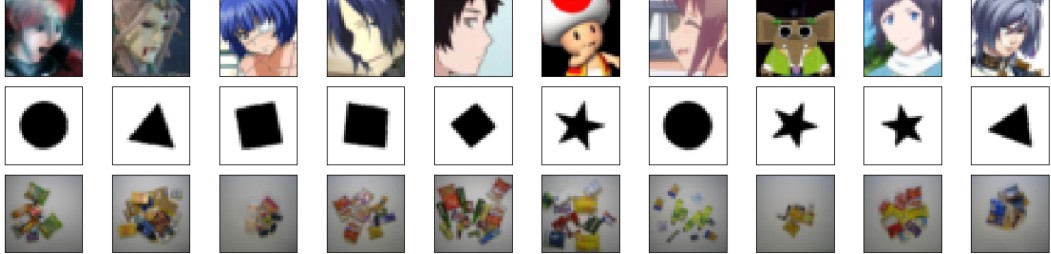

Figure 12: Random samples from three data sets, each noticeably different from CIFAR-10. First row: iCartoonFace; Second row: Four shapes; Third row: RPC.

In Table 7, we present some preliminary results using models trained with the respective method on CIFAR-10 as ID data set (as in Table 1). Results for comparison are presented for EBO-OE only, as time constraints prevented experimenting with additional baseline methods. Although one would expect near-perfect results due to the evident disparities with CIFAR-10, Four Shapes (smeschke, 2018) and RPC (Wei et al., 2022b) seem to defy that expectation. Their results indicate a weakness in the capability to identify outliers robustly since many samples are classified as inliers. Only iCartoonFace (Zheng et al., 2020) is correctly detected as OOD, at least to a large degree. Interestingly, the weakness uncovered by this data is present in both methods, although more pronounced in EBO-OE. Therefore, we suspect that this specific behavior may be a general weakness when training OOD detectors using OE, an aspect we plan to investigate further in our future work.

### I.8 Runtime Considerations for Inference

When using Hopfield Boosting in inference, an additional inference step is needed to check whether a given sample is ID or OOD. Namely, to obtain the score (Equation (13)) of a query sample $\boldsymbol{\xi}^{\mathcal{D}}$, Hopfield Boosting computes the dot product similarity of the embedding obtained from $\boldsymbol{\xi} = \phi(\boldsymbol{\xi}^{\mathcal{D}})$ to all samples in the Hopfield memories $\boldsymbol{X}$ and $\boldsymbol{O}$. In our experiments, $\boldsymbol{X}$ contains the full in-distribution data set (50,000 samples) and $\boldsymbol{O}$ contains a subset of the AUX data set of equal size. We investigate the computational overhead of computing the dot-product similarity to 100,000 samples in relation to the computational load of the encoder. For this, we feed 100 batches of size 1024 to an encoder (1) without using the score and (2) with using the score, measure the runtimes per batch, and compute the mean and standard deviation. We conduct this experiment with four different encoders on an NVIDIA Titan V GPU. The results are shown in Figure 13 and Table 8. One can see that, especially for larger models, the computational overhead of determining the score is very moderate in comparison.

Table 9: OOD detection performance on CIFAR-10. We compare results from Hopfield Boosting with two extensions of HE (Zhang et al., 2023a) on ResNet-18: HE+AUX includes AUX data in the OOD score. HE+OE applies OE (Hendrycks et al., 2019b) during the training process. ↓ indicates "lower is better" and ↑ "higher is better". All values in %.

| OOD Dataset | Metric | HB (ours) | HE+AUX | HE+OE |
|---|---|---|---|---|
| SVHN | FPR95 ↓ | **0.23** | 25.02 | 2.38 |
| | AUROC ↑ | **99.57** | 94.90 | 99.30 |
| LSUN-Crop | FPR95 ↓ | **0.82** | 7.35 | 2.39 |
| | AUROC ↑ | **99.40** | 98.67 | 99.22 |
| LSUN-Resize | FPR95 ↓ | **0.00** | 13.69 | **0.00** |
| | AUROC ↑ | **99.98** | 97.68 | 99.95 |
| Textures | FPR95 ↓ | **0.16** | 17.42 | 0.70 |
| | AUROC ↑ | **99.84** | 97.08 | 99.72 |
| iSUN | FPR95 ↓ | **0.00** | 14.76 | **0.00** |
| | AUROC ↑ | **99.97** | 97.68 | 99.95 |
| Places 365 | FPR95 ↓ | **4.28** | 41.24 | 10.84 |
| | AUROC ↑ | **98.51** | 91.16 | 96.83 |
| Mean | FPR95 ↓ | **0.92** | 19.91 | 2.72 |
| | AUROC ↑ | **99.55** | 96.15 | 99.16 |

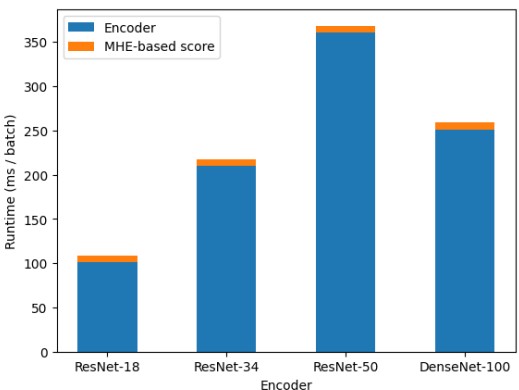

Figure 13: Mean inference runtimes for Hopfield Boosting on four different encoders on an NVIDIA Titan V GPU. We plot the contributions to the total runtime of the encoder and the MHE-based score (Equation (13)) separately. The evaluation shows that the score computation adds a negligible amount of computational overhead to the total runtime.

Table 8: Inference runtimes for Hopfield Boosting with four different encoders on an NVIDIA Titan V GPU. We compare the runtime of the encoder only and the runtime of the encoder with the MHE-based score computation (Equation (13)) combined.

| Encoder | Time encoder (ms / batch) | Time encoder + score (ms / batch) | Rel. overhead (%) |
|---|---|---|---|
| ResNet-18 | $100.93^{\pm 0.24}$ | $108.50^{\pm 0.19}$ | 7.50 |
| ResNet-34 | $209.80^{\pm 0.40}$ | $217.33^{\pm 0.51}$ | 3.59 |
| ResNet-50 | $360.93^{\pm 1.51}$ | $368.17^{\pm 0.62}$ | 2.01 |
| Densenet-100 | $251.24^{\pm 1.36}$ | $258.82^{\pm 0.84}$ | 3.02 |

## I.9   HE and SHE Extensions with AUX Data

To show that the unique contributions of Hopfield Boosting (like the energy-based loss $\mathcal{L}_{OOD}$ and the boosting process) are responsible for the superior performance of Hopfield Boosting, we devise two extensions of HE that include AUX data and compare them to Hopfield Boosting.

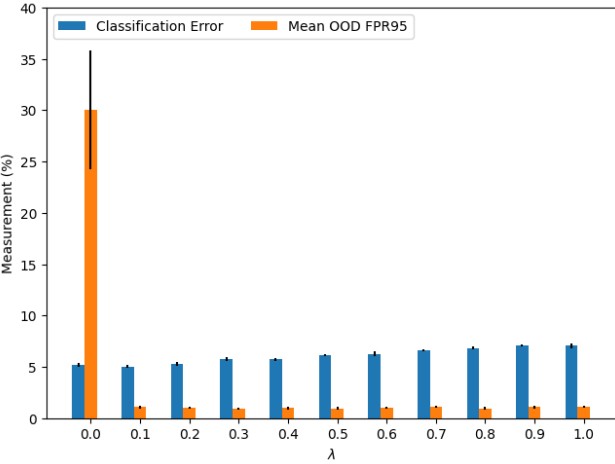

Figure 14: Tradeoff between classification error and Mean OOD FPR95 for different values of $\lambda$. Decreasing the value of $\lambda$ to $0.1$ improves the classification error while maintaining low OOD FPR95. $\lambda = 0$ (i.e., training only the ID classifier) achieves low classification error but dramatically increases the OOD FPR95.

The first extension (HE+AUX) uses a model trained only on the ID data and adapts HE to include an MHE term that measures the energy of $\boldsymbol{\xi}$ on the AUX data $\boldsymbol{O}$:

$$s_{\mathrm{mod}}(\boldsymbol{\xi}) \;=\; s_{\mathrm{HE}}(\boldsymbol{\xi}) \;-\; \mathrm{lse}(\beta, \boldsymbol{O}^T\boldsymbol{\xi}) \tag{136}$$

The second extension (HE+OE) applies OE (Hendrycks et al., 2019b) while training the model. It then uses $s_{\mathrm{HE}}$ to estimate whether a sample is ID or OOD. For both extensions, we select $\beta$ by minimizing the mean FPR95 on the OOD test data sets to obtain an upper bound of the possible performance of these extensions. The $\beta$ we selected for HE+AUX is $0.001$, and for HE+OE is $0.01$.

Our results (Table 9) show that Hopfield Boosting is superior to both extensions. HE+AUX results in a mean FPR95 of 19.91, HE+OE achieves a mean FPR95 of 2.72. Hopfield Boosting improves on both extensions, achieving a mean FPR95 of 0.92.

### I.10 Ablation on the Hyperparameter $\lambda$

There is usually an inherent tradeoff between ID accuracy and OOD detection performance when employing OE methods. In practice one can always improve the tradeoff by using models with more capacity — in the extreme case practitioners can even train a separate ID network. Hence, the model selection process we employed only considered the OOD detection performance and did not take the ID accuracy into account. To investigate if and how this tradeoff can be controlled by changing the hyperparameters of Hopfield Boosting, we conduct the following experiment:

We (1) ablate the hyperparameter (the weight of the out-of-distribution loss) and run Hopfield Boosting on the CIFAR-10 benchmark; (2) select $\lambda$ from the range $[0, 1]$ with a step size of $0.1$; and (3) record the OOD detection performance (the mean FPR95 where the mean is taken over the OOD test data sets) and the ID classification error for the individual settings of $\lambda$.

The results indicate that decreasing the hyperparameter $\lambda$ improves the ID classification accuracy of Hopfield Boosting (Figure 14). At the same time, the mean OOD AUROC is only moderately influenced: When setting, the hyperparameter setting reported in the original manuscript, the mean ID classification error is $5.98\%$, and the mean FPR95 is $0.92\%$. When decreasing $\lambda$ to $0.1$, the mean ID classification error improves to $5.02\%$. Similarly, the FPR95 only slightly increases to $1.08\%$ (which is still substantially better than the second-best outlier exposed method, POEM, which achieves a mean FPR95 of $2.28\%$). Hence, practitioners can control the tradeoff between ID classification accuracy and OOD detection performance.

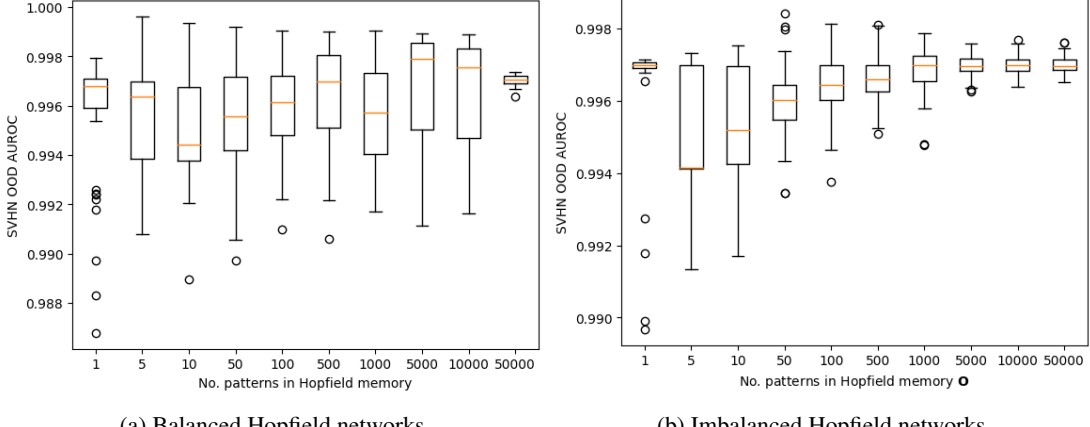

| (a) Balanced Hopfield networks | (b) Imbalanced Hopfield networks |

Figure 15: Ablating the number of patterns stored in the Hopfield memories during inference. AUROC on SVHN based on the number of patterns in the Hopfield memory. In (a), $X$ and $O$ contain the same number of patterns; in (b) $X$ contains $50,000$ patterns, and we vary the number of patterns in $O$. The variability of the AUROC is reduced when $X$ and $O$ contain 50,000 patterns, respectively.

## I.11 Ablation on the Number of Patterns Stored in the Memories during Inference

In our implementation of Hopfield Boosting, we fill the memories $X$ and $O$ with $N = 50,000$ patterns to compute the score $s(\boldsymbol{\xi})$, respectively. To investigate the robustness of Hopfield Boosting when changing the number of patterns $N$, we conduct the following experiments:

1. We train Hopfield Boosting on CIFAR-10 (ID data) and ImageNet (AUX data). During the weight update process, we store 50,000 patterns in the memories $X$ and $O$, and then ablate the number of patterns stored in the memories for computing the score $s(\boldsymbol{\xi})$ at inference time. We evaluate the discriminative power of $s(\boldsymbol{\xi})$ on SVHN with 1, 5, 10, 50, 100, 500, 1,000, 5,000, 10,000, and 50,000 patterns stored in the memories $X$ and $O$. To investigate the influence of the stochastic process of sampling $N$ patterns from the ID and AUX data sets, we conduct 50 runs for all of the and create boxplots of the runs. The results (Figure 15a) show that sampling $50,000$ patterns has the lowest variability of the individual trials. We argue that the reason for this is that by this time the entire ID data set is stored in the Hopfield memory — which effectively eliminates stochasticity from randomly selecting $N$ patterns from the ID data.

2. To verify that we can use $s(\boldsymbol{\xi})$ when the number of patterns in $X$ and $O$ is imbalanced, we fill $X$ with all 50,000 data instances of CIFAR-10 and fill $O$ with 1, 5, 10, 50, 100, 500, 1000, 5000, 10,000, and 50,000 data instances of the AUX data set. Then, we evaluate the discriminative power of $s(\boldsymbol{\xi})$ for the different instances. Our results (Figure 15b) show that Hopfield Boosting is robust to an imbalance in the number of samples in $X$ and $O$. The setting with 50,000 samples in both memories (which is the setting we use in the experiments in our original manuscript) incurs the least variability.

## I.12 Compute Ressources

Our experiments were conducted on an internal cluster equipped with a variety of different GPU types (ranging from the NVIDIA Titan V to the NVIDIA A100-SXM-80GB). For our experiments on ImageNet-1K, we additionally used resources of an external cluster that is equipped with NVIDIA A100-SXM-64GB GPUs.

For our experiments with Hopfield Boosting on CIFAR-10 and CIFAR-100, one run (100 epochs) of Hopfield Boosting trained for about 8.0 hours on a single NVIDIA RTX 2080 Ti GPU and required 4.3 GB of VRAM. Fnding the hyperparameters required 160h of compute for CIFAR-10 and CIFAR-100, respectively. These were divided across four RTX 2080 Ti. Estimating the standard deviation required 40 hours of compute on a single RTX 2080 Ti for CIFAR-10 and CIFAR-100 respectively.

For ImageNet-1K, one run (4 epochs) of Hopfield Boosting trained for about 4.4 hours on a single NVIDIA A-100-SXM64GB GPU and required 26.9 GB of VRAM. Finding the optimal hyperparameters required a total of 86h of compute, divided across 20 NVIDIA A-100-SXM64GB GPUs. Estimating the standard deviation required 22 hours of compute, divided across 5 NVIDIA A-100-SXM64GB GPUs.

The amount of resources reported above cover the compute for obtaining the results of Hopfield Boosting reported in the paper. The total amount of compute resources for the project is substantially higher. Notable additional compute expenses are preliminary training runs during the development of Hopfield Boosting, and the training runs for tuning the hyperparameters and evaluating the results of the methods we compare Hopfield Boosting to.

### I.13 Data Sets and Licenses

We provide a list of the data sets we used in our experiments and, where applicable, specify their licenses:

- CIFAR-10 (Krizhevsky, 2009): License unknown
- CIFAR-100 (Krizhevsky, 2009): License unknown
- ImageNet-RC (Chrabaszcz et al., 2017): Custom License[2]
- SVHN (Netzer et al., 2011): Creative Commons (CC)
- Textures (Cimpoi et al., 2014): Custom License[3]
- iSUN (Xu et al., 2015): License unknown
- Places 365 (López-Cifuentes et al., 2020): License unknown
- LSUN (Yu et al., 2015): License unknown
- ImageNet-1K (Russakovsky et al., 2015): Custom License[2]
- ImagetNet-21K (Ridnik et al., 2021): Custom License[2]
- SUN (Isola et al., 2011): License unknown
- iNaturalist (Van Horn et al., 2018): Custom License[4]

---

[2]https://image-net.org/download.php
[3]https://www.robots.ox.ac.uk/~vgg/data/dtd/index.html
[4]https://github.com/visipedia/inat_comp/tree/master/2017

Table 10: OOD detection performance on CIFAR-10. We compare results from Hopfield Boosting, PALM (Lu et al., 2024), NPOS (Tao et al., 2023), SSD+ (Sehwag et al., 2021), ASH (Djurisic et al., 2023), GEN (Liu et al., 2023), EBO (Liu et al., 2020), MaxLogit (Hendrycks et al., 2019a), and MSP (Hendrycks & Gimpel, 2017) on ResNet-18. ↓ indicates "lower is better" and ↑ "higher is better". All values in %. Standard deviations are estimated across five training runs.

| | | HB (ours) | PALM | NPOS | SSD+ | ASH | GEN | EBO | MaxLogit | MSP |
|---|---|---|---|---|---|---|---|---|---|---|
| SVHN | FPR95 ↓ | $\mathbf{0.23^{\pm 0.08}}$ | $1.24^{\pm 0.49}$ | $9.04^{\pm 1.13}$ | $3.05^{\pm 0.22}$ | $25.17^{\pm 9.55}$ | $33.26^{\pm 5.99}$ | $32.10^{\pm 6.41}$ | $33.27^{\pm 6.18}$ | $49.41^{\pm 3.77}$ |
| | AUROC ↑ | $99.57^{\pm 0.06}$ | $\mathbf{99.70^{\pm 0.12}}$ | $98.37^{\pm 0.23}$ | $99.41^{\pm 0.06}$ | $94.86^{\pm 2.09}$ | $93.53^{\pm 1.42}$ | $93.43^{\pm 1.60}$ | $93.29^{\pm 1.57}$ | $92.48^{\pm 0.93}$ |
| LSUN-Crop | FPR95 ↓ | $\mathbf{0.82^{\pm 0.17}}$ | $1.21^{\pm 0.27}$ | $5.52^{\pm 0.50}$ | $2.83^{\pm 1.10}$ | $13.13^{\pm 1.81}$ | $19.40^{\pm 2.22}$ | $17.25^{\pm 2.30}$ | $18.50^{\pm 2.24}$ | $38.32^{\pm 2.61}$ |
| | AUROC ↑ | $99.40^{\pm 0.04}$ | $\mathbf{99.65^{\pm 0.05}}$ | $98.97^{\pm 0.04}$ | $99.37^{\pm 0.16}$ | $97.33^{\pm 0.36}$ | $96.48^{\pm 0.46}$ | $96.73^{\pm 0.46}$ | $96.52^{\pm 0.47}$ | $94.37^{\pm 0.53}$ |
| LSUN-Resize | FPR95 ↓ | $\mathbf{0.00^{\pm 0.00}}$ | $27.01^{\pm 5.82}$ | $26.85^{\pm 3.14}$ | $34.30^{\pm 2.17}$ | $38.18^{\pm 5.78}$ | $31.50^{\pm 3.92}$ | $30.69^{\pm 4.03}$ | $31.64^{\pm 4.01}$ | $45.82^{\pm 3.48}$ |
| | AUROC ↑ | $\mathbf{99.98^{\pm 0.02}}$ | $95.41^{\pm 0.74}$ | $95.68^{\pm 0.36}$ | $94.78^{\pm 0.25}$ | $90.39^{\pm 2.00}$ | $94.04^{\pm 0.84}$ | $94.02^{\pm 0.86}$ | $93.90^{\pm 0.86}$ | $92.84^{\pm 0.80}$ |
| Textures | FPR95 ↓ | $\mathbf{0.16^{\pm 0.02}}$ | $17.32^{\pm 2.50}$ | $27.72^{\pm 2.55}$ | $21.20^{\pm 2.20}$ | $46.08^{\pm 6.22}$ | $44.62^{\pm 4.14}$ | $44.67^{\pm 4.46}$ | $44.97^{\pm 4.44}$ | $55.04^{\pm 2.86}$ |
| | AUROC ↑ | $\mathbf{99.84^{\pm 0.01}}$ | $96.82^{\pm 0.71}$ | $95.36^{\pm 0.35}$ | $96.46^{\pm 0.35}$ | $88.32^{\pm 2.08}$ | $90.12^{\pm 1.32}$ | $89.61^{\pm 1.50}$ | $89.56^{\pm 1.48}$ | $90.10^{\pm 0.92}$ |
| iSUN | FPR95 ↓ | $\mathbf{0.00^{\pm 0.00}}$ | $25.71^{\pm 4.83}$ | $26.90^{\pm 3.52}$ | $35.71^{\pm 2.27}$ | $42.41^{\pm 6.28}$ | $35.85^{\pm 4.05}$ | $34.99^{\pm 4.33}$ | $36.02^{\pm 4.18}$ | $49.10^{\pm 3.06}$ |
| | AUROC ↑ | $\mathbf{99.97^{\pm 0.02}}$ | $95.60^{\pm 0.65}$ | $95.74^{\pm 0.38}$ | $94.49^{\pm 0.25}$ | $89.06^{\pm 2.26}$ | $93.05^{\pm 0.84}$ | $92.99^{\pm 0.90}$ | $92.88^{\pm 0.90}$ | $91.99^{\pm 0.74}$ |
| Places 365 | FPR95 ↓ | $\mathbf{4.28^{\pm 0.23}}$ | $22.97^{\pm 2.17}$ | $32.62^{\pm 0.13}$ | $24.99^{\pm 1.21}$ | $48.03^{\pm 2.04}$ | $45.82^{\pm 1.07}$ | $44.87^{\pm 1.11}$ | $45.63^{\pm 1.26}$ | $57.58^{\pm 0.97}$ |
| | AUROC ↑ | $\mathbf{98.51^{\pm 0.10}}$ | $94.95^{\pm 0.53}$ | $93.76^{\pm 0.12}$ | $94.93^{\pm 0.22}$ | $85.65^{\pm 0.77}$ | $88.68^{\pm 0.28}$ | $88.53^{\pm 0.30}$ | $88.42^{\pm 0.29}$ | $88.06^{\pm 0.25}$ |
| Mean | FPR95 ↓ | **0.92** | 15.91 | 21.44 | 20.35 | 35.50 | 35.07 | 34.09 | 35.00 | 49.21 |
| | AUROC ↑ | **99.55** | 97.02 | 96.31 | 96.57 | 90.94 | 92.65 | 92.55 | 92.43 | 91.64 |
| Method type | | OE | Training | Training | Training | Post-hoc | Post-hoc | Post-hoc | Post-hoc | Post-hoc |
| Augmentations | | Weak | Strong | Strong | Strong | Weak | Weak | Weak | Weak | Weak |
| Auxiliary outlier data | | ✓ | ✗ | ✗ | ✗ | ✗ | ✗ | ✗ | ✗ | ✗ |

## I.14 Non-OE Baselines

To confirm the prevailing notion that OE methods can improve the OOD detection capability in general, we compare Hopfield Boosting to 3 training methods (Sehwag et al., 2021; Tao et al., 2023; Lu et al., 2024) and 5 post-hoc methods (Hendrycks & Gimpel, 2017; Hendrycks et al., 2019b; Liu et al., 2020, 2023; Djurisic et al., 2023). For all methods, we train a ResNet-18 on CIFAR-10. For Hopfield Boosting, we use the same training setup as described in section 4.2. For the post-hoc methods, we do not use the auxiliary outlier data. For the training methods, we use the training procedures described in the respective publications for 100 epochs. Notably, all training methods employ stronger augmentations than the OE or the post-hoc methods. The OE and post-hoc methods use the following augmentations (denoted as "Weak"):

1. RandomCrop (32x32), padding 4
2. RandomHorizontalFlip

The training methods use the following augmentations (denoted as "Strong"):

1. RandomResizedCrop (32x32), scale 0.2-1
2. RandomHorizontalFlip
3. ColorJitter applied with probability 0.8
4. RandomGrayscale applied with probability 0.2

Table 10 shows the results of the comparison of Hopfield Boosting to the post-hoc and training methods. Hopfield Boosting is better at OOD detection than all non-OE baselines on CIFAR-10 in terms of both mean AUROC and mean FPR95 by a large margin. Further, Hopfield Boosting achieves the best OOD detection on all OOD data sets in terms of FPR95 and AUROC, except for SVHN and LSUN-Crop, where PALM (Lu et al., 2024) shows better AUROC results. An interesting avenue for future work is to combine one of the non-OE based training methods with the OE method Hopfield Boosting.

# J Informativeness of Sampling with High Boundary Scores

This section adopts and expands the arguments of Ming et al. (2022) on sampling with high boundary scores.

We assume the extracted features of a trained deep neural network to approximately equal a Gaussian mixture model with equal class priors:

$$p(\boldsymbol{\xi}) = \frac{1}{2}\mathcal{N}(\boldsymbol{\xi}; \boldsymbol{\mu}, \sigma^2 \boldsymbol{I}) + \frac{1}{2}\mathcal{N}(\boldsymbol{\xi}; -\boldsymbol{\mu}, \sigma^2 \boldsymbol{I}) \tag{137}$$

$$p_{\text{ID}}(\boldsymbol{\xi}) = p(\boldsymbol{\xi}|\text{ID}) = \mathcal{N}(\boldsymbol{\xi}; \boldsymbol{\mu}, \sigma^2 \boldsymbol{I}) \tag{138}$$

$$p_{\text{AUX}}(\boldsymbol{\xi}) = p(\boldsymbol{\xi}|\text{AUX}) = \mathcal{N}(\boldsymbol{\xi}; -\boldsymbol{\mu}, \sigma^2 \boldsymbol{I}) \tag{139}$$

Using the MHE and sufficient data from those distributions, we can estimate the densities $p(\boldsymbol{\xi})$, $p(\boldsymbol{\xi}|\text{ID})$ and $p(\boldsymbol{\xi}|\text{AUX})$.

**Lemma J.1.** *(see Lemma E.1 in Ming et al. (2022)) Assume the M sampled data points $\boldsymbol{o}_i \sim p_{AUX}$ satisfy the following constraint on high boundary scores $\mathrm{E}_b(\boldsymbol{\xi})$*

$$\frac{-\sum_{i=1}^{M} \mathrm{E}_b(\boldsymbol{o}_i)}{M} \leq \epsilon \tag{140}$$

*Then they have*

$$\sum_{i=1}^{M} |2\boldsymbol{\mu}^T \boldsymbol{o}_i| \leq M\epsilon\sigma^2 \tag{141}$$

*Proof.* They first obtain the expression for $\mathrm{E}_b(\boldsymbol{\xi})$ under the Gaussian mixture model described above and can express $p(\text{AUX}|\boldsymbol{\xi})$ as

$$p(\text{AUX}|\boldsymbol{\xi}) = \frac{p(\boldsymbol{\xi}|\text{AUX})p(\text{AUX})}{p(\boldsymbol{\xi})} \tag{142}$$

$$= \frac{\frac{1}{2}p(\boldsymbol{\xi}|\text{AUX})}{\frac{1}{2}p(\boldsymbol{\xi}|\text{ID}) + \frac{1}{2}p(\boldsymbol{\xi}|\text{AUX})} \tag{143}$$

$$= \frac{(2\pi\sigma^2)^{-d/2}\exp(-\frac{1}{2\sigma^2}||\boldsymbol{\xi}-\boldsymbol{\mu}||_2^2)}{(2\pi\sigma^2)^{-d/2}\exp(-\frac{1}{2\sigma^2}||\boldsymbol{\xi}+\boldsymbol{\mu}||_2^2) + (2\pi\beta^{-1})^{-d/2}\exp(-\frac{1}{2\sigma^2}||\boldsymbol{\xi}-\boldsymbol{\mu}||_2^2)} \tag{144}$$

$$= \frac{1}{1 + \exp(-\frac{1}{2\sigma^2}(||\boldsymbol{\xi}-\boldsymbol{\mu}||_2^2 - ||\boldsymbol{\xi}+\boldsymbol{\mu}||_2^2))} \tag{145}$$

When defining $f_{\text{AUX}}(\boldsymbol{\xi}) = \frac{1}{2\sigma^2}(||\boldsymbol{\xi}-\boldsymbol{\mu}||_2^2 - ||\boldsymbol{\xi}+\boldsymbol{\mu}||_2^2)$ such that $p(\text{AUX}|\boldsymbol{\xi}) = \sigma(f_{\text{AUX}}(\boldsymbol{\xi})) = \frac{1}{1 + \exp(-f_{\text{AUX}}(\boldsymbol{\xi}))}$, they define $\mathrm{E}_b$ as follows:

$$\mathrm{E}_b(\boldsymbol{\xi}) = -|f_{\text{AUX}}(\boldsymbol{\xi})| \tag{146}$$

$$= -\frac{1}{2\sigma^2}|\,||\boldsymbol{\xi}-\boldsymbol{\mu}||_2^2 - ||\boldsymbol{\xi}+\boldsymbol{\mu}||_2^2\,| \tag{147}$$

$$= -\frac{1}{2\sigma^2}|\,\boldsymbol{\xi}^T\boldsymbol{\xi} - 2\boldsymbol{\mu}^T\boldsymbol{\xi} + \boldsymbol{\mu}^T\boldsymbol{\mu} - (\boldsymbol{\xi}^T\boldsymbol{\xi} + 2\boldsymbol{\mu}^T\boldsymbol{\xi} + \boldsymbol{\mu}^T\boldsymbol{\mu})| \tag{148}$$

$$= -\frac{|2\boldsymbol{\mu}^T\boldsymbol{\xi}|}{\sigma^2} \tag{149}$$

Therefore, the constraint in Equation (141) is translated to

$$\sum_{i=1}^{M} |2\boldsymbol{\mu}^T \boldsymbol{o}_i| \leq M\epsilon\sigma^2 \tag{150}$$

$\square$

As $\max_{i \in M} |\boldsymbol{\mu}^T \boldsymbol{o}_i| \leq \sum_{i=1}^{M} |\boldsymbol{\mu}^T \boldsymbol{o}_i|$ given a fixed $M$, the selected samples can be seen as generated from $p_{\text{AUX}}$ with the constraint that all samples lie within the two hyperplanes in Equation (150).

**Parameter estimation.** Now they show the benefit of such constraint in controlling the sample complexity. Assume the signal/noise ratio is large: $\frac{||\boldsymbol{\mu}||}{\sigma} = r \gg 1$, and $\epsilon \leq 1$ is some constant.

Assume the classifier is given by

$$\boldsymbol{\theta} = \frac{1}{N+M}\left(\sum_{i=1}^{M} \boldsymbol{x}_i - \sum_{i=1}^{N} \boldsymbol{o}_i\right) \tag{151}$$

where $\boldsymbol{o}_i \sim p_{\text{AUX}}$ and $\boldsymbol{x}_i \sim p_{\text{ID}}$. One can decompose $\boldsymbol{\theta}$. Assuming $M = N$:

$$\boldsymbol{\theta} = \boldsymbol{\mu} + \frac{1}{2}\boldsymbol{\eta} + \frac{1}{2}\boldsymbol{\omega} \tag{152}$$

$$\boldsymbol{\eta} = \frac{1}{N}\left(\sum_{i=1}^{N} \boldsymbol{x}_i\right) - \boldsymbol{\mu} \tag{153}$$

$$\boldsymbol{\omega} = \frac{1}{N}\left(\sum_{i=1}^{M} - \boldsymbol{o}_i\right) - \boldsymbol{\mu} \tag{154}$$

We would now like to determine the distributions of the random variables $||\boldsymbol{\eta}||_2^2$ and $\boldsymbol{\mu}^T \boldsymbol{\eta}$

$$||\boldsymbol{\eta}||_2^2 = \sum_{i=1}^{d} \eta_i^2 \tag{155}$$

$$\eta_i \sim \mathcal{N}(0, \frac{\sigma^2}{N}) \tag{156}$$

$$\frac{\sqrt{N}}{\sigma}\eta_i \sim \mathcal{N}(0,1) \tag{157}$$

$$(\frac{\sqrt{N}}{\sigma}\eta_i)^2 \sim \chi_1^2 \tag{158}$$

Therefore, for $||\boldsymbol{\eta}||_2^2$ we have

$$\frac{N}{\sigma^2}||\boldsymbol{\eta}||_2^2 = \sum_{i=1}^{d}(\frac{\sqrt{N}}{\sigma}\eta_i)^2 \sim \chi_d^2 \tag{159}$$

Now we would like to determine the distribution of $\boldsymbol{\mu}^T \boldsymbol{\eta}$:

$$\boldsymbol{\mu}^T \boldsymbol{\eta} = \sum_{i=1}^{d} \mu_i \, \eta_i \tag{160}$$

$$\mu_i \, \eta_i \sim \mathcal{N}(0, \frac{\sigma^2 \mu_i^2}{N}) \tag{161}$$

$$\sum_{i=1}^{d} \mu_i \, \eta_i \sim \mathcal{N}(0, \sum_{i=1}^{d} \frac{\sigma^2 \mu_i^2}{N}) \tag{162}$$

$$\sum_{i=1}^{d} \mu_i \, \eta_i \sim \mathcal{N}(0, \frac{\sigma^2}{N} \sum_{i=1}^{d} \mu_i^2) \tag{163}$$

$$\frac{\boldsymbol{\mu}^T \boldsymbol{\eta}}{||\boldsymbol{\mu}||} \sim \mathcal{N}(0, \frac{\sigma^2}{N}) \tag{164}$$

**Concentration bounds.** They now develop concentration bounds for $||\boldsymbol{\eta}||_2^2$ and $\boldsymbol{\mu}^T \boldsymbol{\eta}$. First, we look at $||\boldsymbol{\eta}||_2^2$. A concentration bound for $\chi_d^2$ is:

$$\mathbb{P}(X - d \geq 2\sqrt{dx} + 2x) \leq \exp(-x) \tag{165}$$

By assuming $x = \frac{d}{8\sigma^2}$ we obtain

$$\mathbb{P}(X - d \geq 2\sqrt{d\frac{d}{8\sigma^2}} + 2\frac{d}{8\sigma^2}) \leq \exp(-\frac{d}{8\sigma^2}) \tag{166}$$

$$\mathbb{P}(X \geq \frac{d}{\sqrt{2}\sigma} + \frac{d}{4\sigma^2} + d) \leq \exp(-\frac{d}{8\sigma^2}) \tag{167}$$

$$\mathbb{P}(\frac{N}{\sigma^2}||\boldsymbol{\eta}||_2^2 \geq \frac{d}{\sqrt{2}\sigma} + \frac{d}{4\sigma^2} + d) \leq \exp(-\frac{d}{8\sigma^2}) \tag{168}$$

$$\mathbb{P}(||\boldsymbol{\eta}||_2^2 \geq \frac{\sigma^2}{N}(\frac{d}{\sqrt{2}\sigma} + \frac{d}{4\sigma^2} + d)) \leq \exp(-\frac{d}{8\sigma^2}) \tag{169}$$

If $d \geq 2$ we have that[5]

$$\frac{d}{\sqrt{2}\sigma} + \frac{d}{4\sigma^2} > \frac{1}{\sigma} \tag{170}$$

and thus, the above bound can be simplified when assuming $d \geq 2$ as follows:

$$\mathbb{P}(||\boldsymbol{\eta}||_2^2 \geq \frac{\sigma^2}{N}(\frac{1}{\sigma} + d)) \leq \exp(-\frac{d}{8\sigma^2}) \tag{171}$$

For $||\boldsymbol{\omega}||_2^2$, since all $\boldsymbol{o}_i$ is drawn i.i.d. from $p_{\text{AUX}}$, under the constraint in Equation (150), the distribution of $\boldsymbol{\omega}$ can be seen as a truncated distribution of $\boldsymbol{\eta}$. Thus, with some finite positive constant $c$, we have

$$\mathbb{P}(||\boldsymbol{\omega}||_2^2 \geq \frac{\sigma^2}{N}(d + \frac{1}{\sigma})) \leq c\mathbb{P}(||\boldsymbol{\eta}||_2^2 \geq \frac{\sigma^2}{N}(d + \frac{1}{\sigma})) \leq c\exp(-\frac{d}{8\sigma^2}) \tag{172}$$

Now, we develop a bound for $\boldsymbol{\mu}^T \boldsymbol{\eta}$. A concentration bound for $\mathcal{N}(\mu, \sigma^2)$ is

---

[5]Strictly, the bound is valid for $d > \sqrt{2}$

$$\mathbb{P}(X - \mu \geq t) \leq \exp(\frac{-t^2}{2\sigma^2}) \tag{173}$$

By applying $\frac{\boldsymbol{\mu}^T \boldsymbol{\eta}}{||\boldsymbol{\mu}||} \sim \mathcal{N}(0, \frac{\sigma^2}{N})$ to the above bound we obtain

$$\mathbb{P}(\frac{\boldsymbol{\mu}^T \boldsymbol{\eta}}{||\boldsymbol{\mu}||} \geq t) \leq \exp(\frac{-t^2 N}{2\sigma^2}) \tag{174}$$

Assuming $t = (\sigma||\boldsymbol{\mu}||)^{1/2}$ we obtain

$$\mathbb{P}(\frac{\boldsymbol{\mu}^T \boldsymbol{\eta}}{||\boldsymbol{\mu}||} \geq (\sigma||\boldsymbol{\mu}||)^{1/2}) \leq \exp(\frac{-(\sigma||\boldsymbol{\mu}||)N}{2\sigma^2}) \tag{175}$$

$$\mathbb{P}(\frac{\boldsymbol{\mu}^T \boldsymbol{\eta}}{||\boldsymbol{\mu}||} \geq (\sigma||\boldsymbol{\mu}||)^{1/2}) \leq \exp(\frac{-||\boldsymbol{\mu}||N}{2\sigma}) \tag{176}$$

Due to symmetry, we have

$$\mathbb{P}(-\frac{\boldsymbol{\mu}^T \boldsymbol{\eta}}{||\boldsymbol{\mu}||} \leq -(\sigma||\boldsymbol{\mu}||)^{1/2}) \leq \exp(\frac{-||\boldsymbol{\mu}||N}{2\sigma}) \tag{177}$$

$$\mathbb{P}(-\frac{\boldsymbol{\mu}^T \boldsymbol{\eta}}{||\boldsymbol{\mu}||} \leq -(\sigma||\boldsymbol{\mu}||)^{1/2}) + \mathbb{P}(\frac{\boldsymbol{\mu}^T \boldsymbol{\eta}}{||\boldsymbol{\mu}||} \geq (\sigma||\boldsymbol{\mu}||)^{1/2}) \leq 2\exp(\frac{-||\boldsymbol{\mu}||N}{2\sigma}) \tag{178}$$

We can rewrite the above bound using the absolute value function.

$$\mathbb{P}(\frac{|\boldsymbol{\mu}^T \boldsymbol{\eta}|}{||\boldsymbol{\mu}||} \geq (\sigma||\boldsymbol{\mu}||)^{1/2}) \leq 2\exp(\frac{-||\boldsymbol{\mu}||N}{2\sigma}) \tag{179}$$

**Benefit of high boundary scores.** We will now show why sampling with high boundary scores is beneficial. Recall the results from Equations (150) and (154):

$$\sum_{i=1}^{M} |2\boldsymbol{\mu}^T \boldsymbol{o}_i| \leq M\epsilon\sigma^2 \tag{180}$$

$$\boldsymbol{\omega} = \frac{1}{M}(-\sum_{i=1}^{M} \boldsymbol{o}_i) - \boldsymbol{\mu} \tag{181}$$

The triangle inequality is

$$|a + b| \leq |a| + |b| \tag{182}$$

$$|a + (-b)| \leq |a| + |b| \tag{183}$$

Using the two facts above and the triangle inequality we can bound $|\boldsymbol{\mu}^T \boldsymbol{\omega}|$:

$$\frac{1}{M}|\sum_{i=1}^{M}\boldsymbol{\mu}^T\boldsymbol{o}_i| \leq \frac{\sigma^2\epsilon}{2} \tag{184}$$

$$\frac{1}{M}|-\sum_{i=1}^{M}\boldsymbol{\mu}^T\boldsymbol{o}_i| \leq \frac{\sigma^2\epsilon}{2} \tag{185}$$

$$\frac{1}{M}|-\sum_{i=1}^{M}\boldsymbol{\mu}^T\boldsymbol{o}_i| + ||\boldsymbol{\mu}||_2^2 \leq \frac{\sigma^2\epsilon}{2} + ||\boldsymbol{\mu}||_2^2 \tag{186}$$

$$\frac{1}{M}|-\sum_{i=1}^{M}\boldsymbol{\mu}^T\boldsymbol{o}_i - \boldsymbol{\mu}^T\boldsymbol{\mu}| \leq \frac{\sigma^2\epsilon}{2} + ||\boldsymbol{\mu}||_2^2 \tag{187}$$

$$|\boldsymbol{\mu}^T\boldsymbol{\omega}| \leq ||\boldsymbol{\mu}||_2^2 + \frac{\sigma^2\epsilon}{2} \tag{188}$$

**Developing a lower bound.**   Let

$$||\boldsymbol{\eta}||_2^2 \leq \frac{\sigma^2}{N}(d + \frac{1}{\sigma}) \tag{189}$$

$$||\boldsymbol{\omega}||_2^2 \leq \frac{\sigma^2}{N}(d + \frac{1}{\sigma}) \tag{190}$$

$$\frac{|\boldsymbol{\mu}^T\boldsymbol{\eta}|}{||\boldsymbol{\mu}||} \leq (\sigma||\boldsymbol{\mu}||)^{1/2} \tag{191}$$

hold simultaneously. The probability of this happening can be bounded as follows: We define $T$ and its complement $\bar{T}$:

$$T = \{||\boldsymbol{\eta}||_2^2 \leq \frac{\sigma^2}{N}(d + \frac{1}{\sigma})\} \cap \{||\boldsymbol{\omega}||_2^2 \leq \frac{\sigma^2}{N}(d + \frac{1}{\sigma})\} \cap \{\frac{|\boldsymbol{\mu}^T\boldsymbol{\eta}|}{||\boldsymbol{\mu}||} \leq (\sigma||\boldsymbol{\mu}||)^{1/2}\} \tag{192}$$

$$\bar{T} = \{||\boldsymbol{\eta}||_2^2 > \frac{\sigma^2}{N}(d + \frac{1}{\sigma})\} \cup \{||\boldsymbol{\omega}||_2^2 > \frac{\sigma^2}{N}(d + \frac{1}{\sigma})\} \cup \{\frac{|\boldsymbol{\mu}^T\boldsymbol{\eta}|}{||\boldsymbol{\mu}||} > (\sigma||\boldsymbol{\mu}||)^{1/2}\} \tag{193}$$

With $\mathbb{P}(T) + \mathbb{P}(\bar{T}) = 1$. The probability $\mathbb{P}(\bar{T})$ can be bounded using Boole's inequality and the results in Equations (171), (172) and (179):

$$\mathbb{P}(\bar{T}) \leq \exp(-d/8\sigma^2) + c\exp(-d/8\sigma^2) + 2\exp(\frac{-||\mu||N}{2\sigma}) \tag{194}$$

$$\mathbb{P}(\bar{T}) \leq (1+c)\exp(-d/8\sigma^2) + 2\exp(\frac{-||\mu||N}{2\sigma}) \tag{195}$$

Further, we can bound the probability $\mathbb{P}(T)$:

$$\mathbb{P}(\bar{T}) \leq (1+c)\exp(-d/8\sigma^2) + 2\exp(\frac{-||\mu||N}{2\sigma}) \tag{196}$$

$$1 - \mathbb{P}(T) \leq (1+c)\exp(-d/8\sigma^2) + 2\exp(\frac{-||\mu||N}{2\sigma}) \tag{197}$$

$$\mathbb{P}(T) \geq 1 - (1+c)\exp(-d/8\sigma^2) - 2\exp(\frac{-||\mu||N}{2\sigma}) \tag{198}$$

Therefore, the probability of the assumptions in Equations (189), (190), and (191) occuring simultneously is at least $1 - (1+c)\exp(-d/8\sigma^2) - 2\exp(\frac{-||\mu||N}{2\sigma})$.

By using the triangle inequality, Equation (152) and the Assumptions (189) and (190) they can bound $||\boldsymbol{\theta}||_2^2$:

$$||\boldsymbol{\theta}||_2^2 = || \boldsymbol{\mu} + \frac{1}{2} \boldsymbol{\eta} + \frac{1}{2} \boldsymbol{\omega}||_2^2 \tag{199}$$

$$||\boldsymbol{\theta}||_2^2 \leq ||\boldsymbol{\mu}||_2^2 + ||\frac{1}{2} \boldsymbol{\eta}||_2^2 + ||\frac{1}{2} \boldsymbol{\omega}||_2^2 \tag{200}$$

$$||\boldsymbol{\theta}||_2^2 \leq ||\boldsymbol{\mu}||_2^2 + \frac{1}{4}||\boldsymbol{\eta}||_2^2 + \frac{1}{4}||\boldsymbol{\omega}||_2^2 \tag{201}$$

$$||\boldsymbol{\theta}||_2^2 \leq ||\boldsymbol{\mu}||_2^2 + \frac{1}{2}\frac{\sigma^2}{N}(d + \frac{1}{\sigma}) \tag{202}$$

$$||\boldsymbol{\theta}||_2^2 \leq ||\boldsymbol{\mu}||_2^2 + \frac{\sigma^2}{N}(d + \frac{1}{\sigma}) \tag{203}$$

The reverse triangle inequality is defined as

$$|x - y| \geq \big||x| - |y|\big| \tag{204}$$
$$|x - (-y)| \geq \big||x| - |y|\big| \tag{205}$$

Using the reverse triangle inequality, Equations (152), (188) and Assumption (191) we have that

$$|\boldsymbol{\mu}^T \boldsymbol{\theta}| = |\boldsymbol{\mu}^T \boldsymbol{\mu} + \frac{1}{2}\boldsymbol{\mu}^T \boldsymbol{\eta} + \frac{1}{2} \boldsymbol{\mu}^T \boldsymbol{\omega}| \tag{206}$$

$$|\boldsymbol{\mu}^T \boldsymbol{\theta}| \geq \big||\boldsymbol{\mu}^T \boldsymbol{\mu}| - |\frac{1}{2}\boldsymbol{\mu}^T \boldsymbol{\eta}| - |\frac{1}{2} \boldsymbol{\mu}^T \boldsymbol{\omega}|\big| \tag{207}$$

$$|\boldsymbol{\mu}^T \boldsymbol{\theta}| \geq \big|||\boldsymbol{\mu}||_2^2 - \frac{1}{2}\sigma^{1/2}||\boldsymbol{\mu}||^{3/2} - \frac{1}{2}||\boldsymbol{\mu}||_2^2 - \frac{1}{2}\frac{\sigma^2 \epsilon}{2}\big| \tag{208}$$

$$|\boldsymbol{\mu}^T \boldsymbol{\theta}| \geq \big|\frac{1}{2}||\boldsymbol{\mu}||_2^2 - \frac{1}{2}\sigma^{1/2}||\boldsymbol{\mu}||^{3/2} - \frac{1}{2}\frac{\sigma^2 \epsilon}{2}\big| \tag{209}$$

$$|\boldsymbol{\mu}^T \boldsymbol{\theta}| \geq \big|\frac{1}{2}(||\boldsymbol{\mu}||_2^2 - \sigma^{1/2}||\boldsymbol{\mu}||^{3/2} - \frac{\sigma^2 \epsilon}{2})\big| \tag{210}$$

They have assumed that the signal/noise ratio is large: $\frac{||\boldsymbol{\mu}||}{\sigma} = r \gg 1$. Thus, we can drop the absolute value, because we assume that the term inside the $||$ is larger than zero:

$$|\boldsymbol{\mu}^T \boldsymbol{\theta}| \geq \big|\frac{1}{2}(||\boldsymbol{\mu}||_2^2 - \frac{1}{r}||\boldsymbol{\mu}||^{1/2}||\boldsymbol{\mu}||^{3/2} - \frac{||\boldsymbol{\mu}||_2^2 \epsilon}{2r^2})\big| \tag{211}$$

$$|\boldsymbol{\mu}^T \boldsymbol{\theta}| \geq \big|(1 - \frac{1}{r} - \frac{\epsilon}{2r^2})\frac{1}{2}(||\boldsymbol{\mu}||_2^2)\big| \tag{212}$$

We have

$$(1 - \frac{1}{r} - \frac{\epsilon}{2r^2}) \geq 0 \tag{213}$$

if $r \geq 1.36602540378443\ldots$ and $\epsilon \leq 1$, and therefore

$$|\boldsymbol{\mu}^T \boldsymbol{\theta}| \geq \frac{1}{2}(||\boldsymbol{\mu}||_2^2 - \sigma^{1/2}||\boldsymbol{\mu}||^{3/2} - \frac{\sigma^2 \epsilon}{2}) \tag{214}$$

Because of Equation (203) and the fact that if $x \leq y$ and $\text{sgn}(x) = \text{sgn}(y)$ then $x^{-1} \geq y^{-1}$ we have

$$\frac{1}{||\boldsymbol{\theta}||} \geq \frac{1}{\sqrt{||\boldsymbol{\mu}||_2^2 + \frac{\sigma^2}{N}(d + \frac{1}{\sigma})}} \tag{215}$$

We can combine the Equations (214) and (215) to give a single bound:

$$\frac{|\boldsymbol{\mu}^T\boldsymbol{\theta}|}{||\boldsymbol{\theta}||} \geq \frac{||\boldsymbol{\mu}||_2^2 - \sigma^{1/2}||\boldsymbol{\mu}||^{3/2} - \frac{\sigma^2\epsilon}{2}}{2\sqrt{||\boldsymbol{\mu}||_2^2 + \frac{\sigma^2}{N}(d + \frac{1}{\sigma})}} \tag{216}$$

we define $\boldsymbol{\theta}$ such that $\boldsymbol{\mu}^T\boldsymbol{\theta} > 0$ and thus

$$\frac{\boldsymbol{\mu}^T\boldsymbol{\theta}}{||\boldsymbol{\theta}||} \geq \frac{||\boldsymbol{\mu}||_2^2 - \sigma^{1/2}||\boldsymbol{\mu}||^{3/2} - \frac{\sigma^2\epsilon}{2}}{2\sqrt{||\boldsymbol{\mu}||_2^2 + \frac{\sigma^2}{N}(d + \frac{1}{\sigma})}} \tag{217}$$

The false negative rate $\mathrm{FNR}(\boldsymbol{\theta})$ and false positive rate $\mathrm{FPR}(\boldsymbol{\theta})$ are

$$\mathrm{FNR}(\boldsymbol{\theta}) = \int_{-\infty}^{0} \mathcal{N}(x; \frac{\boldsymbol{\mu}^T\boldsymbol{\theta}}{||\boldsymbol{\theta}||}, \sigma^2)\,\mathrm{d}x \tag{218}$$

$$\mathrm{FPR}(\boldsymbol{\theta}) = \int_{0}^{\infty} \mathcal{N}(x; \frac{-\boldsymbol{\mu}^T\boldsymbol{\theta}}{||\boldsymbol{\theta}||}, \sigma^2)\,\mathrm{d}x \tag{219}$$

As $\mathcal{N}(x; \mu, \sigma^2) = \mathcal{N}(-x; -\mu, \sigma^2)$, we have $\mathrm{FNR}(\boldsymbol{\theta}) = \mathrm{FPR}(\boldsymbol{\theta})$. From Equation (217) we can see that as $\epsilon$ decreases, the lower bound of $\frac{\boldsymbol{\mu}^T\boldsymbol{\theta}}{||\boldsymbol{\theta}||}$ will increase. Thus, the mean of the Gaussian distribution in Equation (218) will increase and therefore, the false negative rate will decrease, which shows the benefit of sampling with high boundary scores. This completes the extended proof adapted from (Ming et al., 2022).

