# OpenReview forum: "Energy-based Hopfield Boosting for Out-of-Distribution Detection"
_NeurIPS.cc/2024/Conference — NeurIPS 2024 poster_

### Official Review · Reviewer_BR8F · 2024-07-07

**Soundness:** 3
**Presentation:** 4
**Contribution:** 3
**Rating:** 7
**Confidence:** 4

**Summary:**

The authors introduce Hopfield Boosting for addressing the out-of-distribution detection task. This algorithm trains a model with two heads: one for normal classification and the other for assigning an out-of-distribution score. The OOD score head maintains a list of in-distribution samples and out-of-distribution samples. It uses a modern Hopfield network to leverage the similarity between the inference sample and stored samples to assign the OOD score.
They further improve their performance by first identifying OOD samples that are near the border of the decision boundary using an MHE-based energy function. Then, they use these samples as OOD samples for the OOD score head instead of all auxiliary OOD samples. They evaluated their method on CIFAR-10 and ImageNet-1k and achieved state-of-the-art results.

**Strengths:**

1. Large-scale experiment on ImageNet.
2. Novel use of Hopfield Boosting.
3. Good toy examples that clearly illustrate their point.
4. Low computation requirement.

**Weaknesses:**

1. Only 32x32 images were used.
2. OE methods generally require auxiliary OOD data. As a result, evaluation on a dataset that does not have common classes with the auxiliary OOD dataset is crucial to assess the method's generalization.

**Questions:**

1. What is the relation between ImageNet-21k, which is used as auxiliary OOD data, and the OOD data used for testing? How many semantically similar classes does ImageNet-21k share with each of them? If it shares a considerable number of classes, a test dataset that shares fewer semantically similar classes would be helpful.
2. ImageNet augmentation: Please explain the augmentation used on in-distribution data for ImageNet. Is it the same as mentioned in line 970? Random cropping to 32x32 for images that are very large, like those in ImageNet-1k, seems unreasonable as 32x32 parts of the image 3. may not contain any meaningful object. Could you provide some examples of in-distribution images post-augmentation?
4. Can you provide the accuracy of classification head of the network?
5. What is the reason for cropping images to 32x32 in the ImageNet-1k case? Why not use the more common 224x224 image size?

**Limitations:**

The authors presented the limitations of their work in section H.6, which could suggest that their method lacks generalization in at least some areas.

---

> ### Author Rebuttal · Authors · 2024-08-07
>
> **Response to Weaknesses:**
> 1. For the experiments comprising ImageNet-1K as the ID data set, we use images with size 224x224. This misunderstanding arises because we missed to report the resolutions we use for the ImageNet benchmark. We apologize for this error. We will make sure that the updated version of the manuscript contains this information.
> 2. The reviewer’s analysis is correct and we agree that generalization beyond the classes contained in the auxiliary outlier data set is a highly desirable property for OOD detection methods with OE. It is hard to say with absolute certainty that an OOD test data set does not contain any class overlap with the auxiliary outlier data set, but as far as we know, ImageNet-1K (the auxiliary outlier data set in the CIFAR-10 setting) and SVHN (an OOD test set in the CIFAR-10 setting) do not share common classes.
>
> **Response to Questions:**
>
> 1. For our experiments, we selected auxiliary outlier data sets that cover a large and diverse region of the feature space. Optimally, one would have highly specific auxiliary data that is 100%  indicative of the OOD samples. However, since the OOD samples are generally not known a-priori there is the option to fall back on such large and diverse auxiliary data sets. Such datasets then have the advantage that they likely contain samples that share some similarity with the OOD samples used for testing. This is a realistic setting for many applications since it is often possible to easily mine such auxiliary datasets. It is, however, unrealistic to expect to obtain an auxiliary outlier data set that will completely cover the entire OOD region. Therefore, Hopfield Boosting frequently samples data instances close to the decision boundary between the ID and OOD region. This allows Hopfield Boosting to learn a decision boundary that more tightly encapsulates the ID data.
>   To verify the effectiveness of Hopfield Boosting trained on the ImageNet-1K benchmark with ImageNet-21K as auxiliary outlier data on noticeably different data sets, we evaluated the model on three additional OOD data sets: iCartoonFaces, RPC, and FourShapes (we refer to Appendix H.6 of the original manuscript for examples from these data sets). The results are as follows (all numbers in %):
>
>     |               | FPR95 | AUROC |
>     |---------------|-------|-------|
>     | iCartoonFaces | 3.7   | 98.76 |
>     | RPC           | 79.73 | 92.64 |
>     | FourShapes    | 0.0   | 99.84 |
>
> 2. We apologize for not reporting that we use images with size 224x224 for the ImageNet-1K benchmark. And, indeed, we also did miss to describe the used transformations, which follow Zhu et al. (2023) for comparability. Namely:
>
>     - Resize to 224x224
>     - RandomCrop 224x224, padding 4
>     - RandomHorizontalFlip
>
>     We will include this list of transformations in the Appendix of the final manuscript.
>
> 3. The mean ID accuracy for Hopfield Boosting is 94.02% for CIFAR-10 (ResNet-18), 75.08% for CIFAR-100 (ResNet-18), and 76.3% for ImageNet-1K (ResNet-50). In the PDF attached to the response on the top of this page, we also include an ablation on $\lambda$, the weight of our energy-based OOD loss. The experiment shows that decreasing $\lambda$ to $0.1$ will improve the ID accuracy when using Hopfield Boosting (from 94.02% to 94.98%), while only moderately influencing OOD detection performance (the FPR95 metric increases from 0.92% to 1.08%).
> 4. We will communicate more clearly that we use a resolution of 224x224 in the ImageNet-1K benchmark.
>
> **References:**
>
> Zhu et al., Diversified Outlier Exposure for Out-of-Distribution Detection via Informative Extrapolation, NeurIPS 2023

---

> > ### Comment · Reviewer_BR8F · 2024-08-09
> >
> > Thank you for the rebuttal. My concerns have been addressed, so I will raise my score to 7.
> >
> > I believe it is crucial that you provide augmentation details for both in-distribution (ID) and out-of-distribution (OOD) data, or at least reference a work that explains them in detail. This is important because datasets sometimes require different augmentation techniques for ID and OOD data, which can result in different types of artifacts in the images. Models are very adept at picking up on these artifacts and may base their predictions on them.

---

> > > ### Author Response · Authors · 2024-08-09
> > >
> > > We thank the reviewer for the response/hints. We agree with this sentiment and also think that it is imperative to include a detailed description of the pre-processing steps.

---

### Official Review · Reviewer_UgW1 · 2024-07-12

**Soundness:** 4
**Presentation:** 4
**Contribution:** 4
**Rating:** 8
**Confidence:** 4

**Summary:**

A novel method for identifying OOD samples is presented using Hopfield boosting.  A classifier is trained using a loss function which sharpens the decision boundary between inlier and outlier data samples, where outlier data samples are purposefully drawn from an auxiliary dataset.  By sampling examples close to the decision boundary and determining the modern hopfield energy of the sample, the decision boundary is adjusted to exclude OOD samples from the auxilliary dataset.  The method is demonstrated twice using a low resolution and high resolution example.  In both examples, the Hopfield boosting method outperforms state of the art on average.  An extensive appendix includes additional background, theoretical results, and experimental results.

**Strengths:**

The method is original, theoretically sound, and useful.  The results are clearly presented, to the point where a reader may be able to implement the algorithm without referencing the provided code.  The research question is impactful, and the result presented in this paper is a theoretical and experimental advancement in the field.

**Weaknesses:**

The primary weakness is that the paper does not conclusively show that the method does not lead to overfitting.  The models are trained with a classification loss, but the authors state "the toy example does not consider the inlier classification task that would induce secondary processes, which would obscure the explanations."  However, failure to report these results in the context of the decision boundary refinements shown in Figure 2, 3b and 6 opens the work to questions of model overfitting.

This concern is only partially lifted by near perfect results on the test datasets.  Because the false positive rate (FPR) is reported with respect to an OOD test dataset, there is a question of whether the model is losing in-distribution detections.   Robustness to OOD samples of the original dataset, robustness to adversarial instances, and catastrophic forgetting are all implicated in the sharpening of decision boundaries; this work avoids acknowledging these cases and prevents inference about the implications by omitting the classifier performance for the trained models.  This can be resolved by reporting the model F1 scores in the appendix for the ID classification task. If the models were trained using the Hopfield Boosting alone, then this would not be necessary.  But the presence of the classifier loss means the secondary effects have implications for the semantic meaning of the decision boundary.

**Questions:**

1. How do these results compliment/contrast with results suggesting that a smooth decision boundary is more robust to adversarial samples? See for example [Anderson (2022)](https://proceedings.mlr.press/v168/anderson22a)

2. Most computer vision datasets are expected to have some features in common, which motivates transfer learning techniques.  What justification was used to determine that the test datasets would be strictly OOD w.r.t. the training datasets CIFAR or ImageNet?

3. Is there any analysis which shows that the decision boundaries sharpened during Hopfield Boosting are semantically meaningful, and not based on noise of the dataset?

4. The ResNet-50 model is fine-tuned rather than trained from scratch; what dataset was used to train the pre-initialized weights? Why was the model not trained from scratch?  Because the feature space was already formed (4 epochs is not enough to significantly shift the model), does this imply that the model classification layer is likely identifying noise?

**Limitations:**

Please see the questions above.  The remaining limitations are thoroughly addressed in the work.

---

> ### Author Rebuttal · Authors · 2024-08-07
>
> **Response to Weaknesses:**
> - We agree with this. To demonstrate how Hopfield Boosting can interact in a classification task we created a toy example that resembles the ID classification setting (Figure 4 in the PDF attached to the response on the top of the page): The example shows the decision boundary and the inlier samples organized in two classes. We sample uniformly distributed auxiliary outliers. Then, we minimize the compound objective $\mathcal{L} = \mathcal{L}\_{CE}+\mathcal{L}\_{OOD}$ (applying the gradient updates on the patterns directly). This shows that $\mathcal{L}_{CE}$ is able to separate the two classes well and that $E_b$ still forms a tight decision boundary around the ID data.
> - We agree with the reviewer. The application of Hopfield Boosting influences the ID decision boundary: It decreases the ID classification accuracy on CIFAR-10 on average from 94.80% without Hopfield Boosting to 94.02% with Hopfield Boosting.
> Our work treats samples close to the decision boundary as weak learners. However, the wider ramifications of this behavior remain unclear. For example, the reviewer rightfully points towards a connection to adversarial examples. And indeed, one can also view the sampling of data instances close to the decision boundary as a form of adversarial training in that we search for something like “natural adversarial examples”: Loosely speaking, the usual adversarial example case starts with a given ID sample and corrupts it in a specific way to get the classifier to output the wrong class probabilities. In our case, we start with a large set of potential adversarial instances (the AUX data) and search for the ones that could be either ID or OOD samples. That is, the sampling process will more likely select data instances that are hard to discriminate for the model — for example, if the model is uncertain whether a leaf in an auxiliary outlier sample is a frog or not. Nonetheless, a closer systematic evaluation of the sharpened decision boundary of Hopfield Boosting is important to fully understand the potential implications w.r.t. adversarial examples. We view such an investigation as out-of-scope for this work. However, we consider it as an interesting avenue for future work and will mention so explicitly in the revised manuscript.
>
>   We include the classification results (with the F1 scores) of a model trained with Hopfield Boosting on the CIFAR-10 benchmark in the official comment below. Table 1 in the PDF posted on top of the page also includes the ID accuracies of all compared methods. Figure 1 in the PDF shows the performance when ablating on $\lambda$, the weight of our OOD loss. We will add all those results in the Appendix of the updated manuscript.
>
> **Response to Questions:**
> 1. We would like to thank the reviewer for this very interesting reference. The dichotomy between sharp and smooth decision boundaries is highly relevant to our work. We will discuss it in the updated manuscript. The smooth boundaries discussed in Anderson et al. (2022) help with “classical” adversarial examples. In this framing, our approach would produce different adversarial examples that are not based on noise, but are more akin to “natural adversarial examples” (see our answer to the weakness above). For example, it is perfectly fine for us that an OOD sample close to the boundary does not correspond to any of the ID classes. Furthermore, the noise based smoothing leads to adversarial robustness at the (potential) cost of  degrading classification performance. Similarly, our sharpening of the boundaries leads to better discrimination between ID and OOD region at the (potential) cost of degrading classification performance.
> 2. The test data sets we employ for OOD (Tables 1 and 2 in the original manuscript) can be seen as a de-facto standard benchmark. They have been used in a wide range of other publications (e.g., Hendrycks et al., 2019; Ming et al., 2022; Zhu et al., 2023). We opted to follow the established work for the sake of comparability.
> However, we believe that the reviewer raises an excellent point in that the choice of OOD data sets needs to be more rigorously discussed by the community. Indeed, albeit the selection is not a focus of our work, we did take up upon the resulting problems with regard to benchmarking: In Appendix H.5 (original manuscript), we investigate the Places 365 data set and show that a substantial amount of instances contained in this data set contains semantic overlap with the ID classes of CIFAR-10. And, in Appendix H.6, we evaluate Hopfield Boosting for additional, noticeably different data sets (iCartoonFaces, 4 Shapes, RPC).
> 3. Yes, in Appendix H.5 of the original manuscript, we show that Hopfield Boosting identifies data instances from the OOD test data set Places 365 that contain classes of the ID data set CIFAR-10 (e.g., automobiles, dogs, trucks, and airplanes) as ID. This indicates that Hopfield Boosting learns semantically meaningful features.
> 4. This information is indeed missing in our manuscript and we will add it. In short: We utilize a ResNet-50 that was pre-trained on ImageNet-1K and is provided by Torchvision. We wanted to keep our method comparable to other outlier exposure methods and therefore closely follow the training protocol of Zhu et al. (2023) (hence, we fine-tune the pre-trained ResNet-50 for four epochs). We mainly chose the model because it is well-documented. The model achieves an ID accuracy of 76.3% after fine-tuning with Hopfield Boosting for four epochs. The pre-trained model achieves an ID accuracy of 76.1%.
>
> **References:**
>
> Anderson et al., Certified Robustness via Locally Biased Randomized Smoothing, PMLR 2022
>
> Hendrycks et al., Deep Anomaly Detection with Outlier Exposure, ICLR 2019
>
> Ming et al., POEM: Out-of-Distribution Detection with Posterior Sampling, ICML 2022
>
> Zhu et al., Diversified Outlier Exposure for Out-of-Distribution Detection via Informative Extrapolation, NeurIPS 2023

---

> ### Author Response · Authors · 2024-08-07
> **Hopfield Boosting ID Classification Results CIFAR-10**
>
> We provide the detailed ID classification results (including the F1 scores) of a model trained using Hopfield Boosting on CIFAR-10:
>
> ```
>              precision    recall  f1-score   support
>
>
>            0      0.934     0.947     0.940      1000
>            1      0.961     0.974     0.967      1000
>            2      0.928     0.914     0.921      1000
>            3      0.860     0.873     0.867      1000
>            4      0.933     0.950     0.942      1000
>            5      0.881     0.892     0.886      1000
>            6      0.963     0.955     0.959      1000
>            7      0.978     0.947     0.962      1000
>            8      0.961     0.961     0.961      1000
>            9      0.967     0.949     0.958      1000
>
>
>     accuracy                          0.936     10000
>    macro avg      0.937     0.936     0.936     10000
> weighted avg      0.937     0.936     0.936     10000
> ```

---

> > ### Comment · Reviewer_UgW1 · 2024-08-09
> >
> > I thank the authors for their attention to the weaknesses of the paper and reporting the ID results.  My concerns are addressed.

---

### Official Review · Reviewer_DQmw · 2024-07-12

**Soundness:** 2
**Presentation:** 3
**Contribution:** 3
**Rating:** 6
**Confidence:** 5

**Summary:**

- The paper introduces a novel approach to improve OOD detection by leveraging modern Hopfield energy. The proposed method, called Hopfield Boosting, utilizes auxiliary outlier data to refine the decision boundary between ID and OOD data. By focusing on hard-to-distinguish auxiliary outlier samples near the decision boundary, the method enhances the ability to detect OOD samples.

**Strengths:**

- The incorporation of modern Hopfield energy and auxiliary outlier data for OOD detection is interesting. The method effectively sharpens the decision boundary between ID and OOD data by utilizing the energy measure, which quantifies dissimilarity between data instances and stored patterns. By sampling informative outliers close to the decision boundary, Hopfield Boosting ensures that the model learns a more precise boundary, leading to better OOD detection performance.
- The proposed method achieves significant improvements in OOD detection metrics compared to existing methods.

**Weaknesses:**

- The experimental results in Table 3 show that the effect of weighted sampling is not very significant. Despite this, the proposed method still outperforms existing approaches. It is unclear whether these performance gains are primarily due to the model's structure (two heads for classification and energy computation), the OOD score defined in Equation (13), or the proposed $L\_{OOD}$. This ambiguity makes it difficult to pinpoint the key factors contributing to the improved performance and understand the true efficacy of each component of the proposed method. More detailed ablation studies are needed to disentangle and evaluate the contributions of these individual components.
- The paper lacks a detailed explanation of how the hyperparameter $\beta$ is determined. While it is mentioned that $\beta$ is selected through a validation process, the specific metrics or criteria used to decide the optimal value are not discussed. Providing a clear methodology for $\beta$ selection, including the validation metrics used, would enhance the reproducibility and transparency of the proposed approach.
- There is a need for discussion regarding the impact of the proposed method on in-distribution (ID) accuracy. The proposed method could potentially negatively affect the performance of the classification head, which would undermine its overall utility. Addressing how the method influences ID accuracy and providing empirical evidence to show that it does not significantly degrade classification performance is crucial for establishing the method's efficacy and practical value.

**Questions:**

The questions related to the methodology and experimental results have been addressed in the Weaknesses section.

**Limitations:**

The authors have adequately discussed the limitations of their work.

---

> ### Author Rebuttal · Authors · 2024-08-07
>
> **Response to Weaknesses:**
>
> - Because of the highly competitive nature of the CIFAR-10 benchmark, we view every improvement as important (e.g., the previous methods' AUROC were already above 99%; e.g., POEM achieves an AUROC of 99.21%). That said, we agree that more evaluations w.r.t. the individual components introduced in Hopfield Boosting will shed more light on which factors influence the good performance of Hopfield Boosting, and thank the reviewer for this suggestion. To find the contribution of the individual elements of Hopfield Boosting, we evaluate the performance of the following ablated training procedures on the CIFAR-10 benchmark:
>
>   1. Random sampling instead of weighted sampling
>   2. Random sampling instead of weighted sampling and no projection head
>   3. No application of $\\mathcal{L}_{OOD}$
>
>   The results in the PDF (Table 2) attached to the answer on top of the page show that all of weighted sampling, the projection head, and $\\mathcal{L}_{OOD}$ are important factors that contribute to Hopfield Boosting’s performance.
>   We will include these experimental results in the Appendix.
>
> - We agree with the reviewer that the manuscript will benefit from a more detailed description of the validation process. We will expand the experimental section with additional information regarding the validation and model selection process, as follows:
>
>   For validation and model selection, we evaluate the model on the OOD data sets MNIST and ImageNet-RC with different preprocessing than in training (resize to 32x32 pixels instead of crop to 32x32 pixels), as well as Gaussian and uniform noise. We train models using a grid search with $\lambda$ selected from the set $\\{0.1, 0.25, 0.5, 1.0\\}$, and $\beta$ selected from the set $\\{2, 4, 8, 16, 32\\}$. From said hyperparameter configurations, we select the model with the lowest mean FPR95 metric (where the mean is taken over the validation OOD data sets) and do not consider the ID classification accuracy for model selection.
>
> - We agree with the reviewer that reporting the ID accuracy will help the reader to better understand the influence between outlier exposure methods and the ID classifier’s performance. Therefore, we have compiled the ID accuracies achieved by the individual outlier exposure methods and included results in the PDF attached.
>
>   There is usually an inherent tradeoff between ID accuracy and OOD detection performance when employing outlier exposure methods. We do however not view it as a crucial aspect of our work, since in practice one can always improve the tradeoff by using models with more capacity — in the extreme case practitioners can even train a separate ID network. Hence, the model selection process we employed so far only considered the OOD detection performance and did not take the ID accuracy into account. That said, we still agree with the general sentiment of the reviewer. Thus, we decided to examine the tradeoff by conducting the following experiment:
>
>   We (1) ablate the $\\lambda$ hyperparameter (the weight of the out-of-distribution loss $\\mathcal{L}_{OOD}$) and run Hopfield Boosting on the CIFAR-10 benchmark; (2) select $\\lambda$ from the range $[0, 1]$ with a step size of $0.1$; and (3) record the OOD detection performance (the mean FPR95 where the mean is taken over the OOD test data sets) and the ID classification error for the individual settings of $\\lambda$.
>
>   The results indicate that decreasing the $\\lambda$ hyperparameter improves the ID classification accuracy of Hopfield Boosting (Figure 1). At the same time, the mean OOD AUROC is only moderately influenced: When setting $\\lambda = 0.5$, the hyperparameter setting reported in the original manuscript, the mean ID classification error is 5.98%, and the mean FPR95 is 0.92%. When decreasing $\\lambda$ to 0.1, the mean ID classification error improves to 5.02%. Similarly, the FPR95 only slightly increases to 1.08% (which is still substantially better than the second-best outlier exposed method, POEM, which achieves a mean FPR95 of 2.28%). Hence, practitioners can control the tradeoff between ID classification accuracy and OOD detection performance.
>
>   Our update will include the ID accuracy results and the tradeoff Figure.

---

> > ### Comment · Reviewer_DQmw · 2024-08-09
> >
> > Thank you for the rebuttal. My concerns have been sufficiently addressed, so I raised the score to 6.

---

### Official Review · Reviewer_qRFa · 2024-07-12

**Soundness:** 3
**Presentation:** 3
**Contribution:** 2
**Rating:** 6
**Confidence:** 4

**Summary:**

This paper addresses the crucial task of out-of-distribution (OOD) detection, essential for the safe deployment of machine learning models in real-world scenarios. Within the domain of OOD detection, outlier exposure methods, which use auxiliary outlier data during training, have shown significant improvements in OOD detection performance. In this paper, the authors introduce Hopfield Boosting, a boosting approach that utilizes modern Hopfield energy together with the OOD data to refine the decision boundary between in-distribution (ID) and OOD data. By focusing on hard-to-distinguish auxiliary outlier examples near the decision boundary, Hopfield Boosting enhances the model's ability to differentiate between ID and OOD data. The authors of the paper demonstrates the effectiveness of the proposed method empirically with several benchmark datasets.

**Strengths:**

- The paper is mostly well written and easy to follow.
- The proposed method is technically sound.
- Authors of the paper carry out ample experiments.
- The proposed method achieves good performance.

**Weaknesses:**

- A major complaint I have for the paper is the lack of novelty. As mentioned by the authors of the paper in Section 3.4, the proposed method is an incremental extension of the prior work which also proposed to use Hopfield Energy for OOD detection. The main difference lies in the fact that the proposed method also utilizes auxiliary OOD data. Despite the effectiveness of the proposed method, outlier exposure is a well understood method to enhance the effectiveness of OOD detection, and it is not surprising to me that utilizing OOD data with Hopfield Energy enhances the effectiveness of the proposed method.

**Questions:**

- The paper introduces the MHE-based energy function to combine OOD data with Hopfield Energy. Would it be possible to leverage OOD data in other ways so that it is more similar to HE or SHE in the prior work [1]? If so, how does the proposed method compare to such a naive way of combining HE/SHE with outlier exposure? I think an ablation study like this would help strengthen the novelty of the paper.
- It was stated that 50k samples were used during inference to compute the OOD score. How much better/worse does the proposed method get when we vary the number of samples used?
- What if IN and OOD data are highly imbalanced?

[1] "Out-of-distribution detection based on in-distribution data patterns memorization with modern Hopfield energy"

**Limitations:**

Yes the authors of the paper adequately addressed the limitations.

---

> ### Author Rebuttal · Authors · 2024-08-07
>
> **Response to Weaknesses:**
>
> - We respectfully disagree with the claim that Hopfield Boosting is an incremental extension. We take it as a hint that we did not emphasize the novelty enough. Hopfield Boosting not only combines existing concepts in a novel, principled, and theoretically well-motivated way, but also includes specific innovations that have not been used in previous work: The novel energy function that, to our knowledge, is first introduced in our paper, fits a nonlinear decision boundary into the embedding space of a neural network. Hopfield Boosting samples patterns close to the decision boundary and computes the OOD loss using the same energy function. Our theoretical results are also novel. They show that the novel energy function is well-motivated from a probabilistic perspective. We will make sure that the revised manuscript states this more clearly.
>
> **Response to Questions:**
> - Adapting the setting of Zhang et al. (2023) to leverage OOD data in the way we do is not trivial. Probably, the most straightforward avenues to include AUX data in their setting are: (a) Use an existing OOD loss (e.g., the one from Hendrycks et al., 2019) to perform outlier exposure during training and use HE/SHE on the resulting model. (b) Use a model trained only on the ID data and adapt HE or SHE to include an MHE term that measures the energy of $\boldsymbol{\xi}$ on the AUX data. Due to time constraints, we only conducted experiment (b) for the rebuttal, but we will include both HE/SHE extensions in the camera-ready version of the manuscript.
>
>   In specific, experiment (b) consists of two steps: First, we take CIFAR-10 as a benchmark and use the following score: $s_{\mathrm{mod}}(\boldsymbol{\xi}) \ = \ s_{\mathrm{HE}}(\boldsymbol{\xi}) - \mathrm{lse}(\beta, \boldsymbol{O}^T\boldsymbol{\xi})$, where $\boldsymbol{O}$ contains 50,000 randomly selected encoded patterns from the auxiliary outlier data set. Second, we tune $\beta$ such that $s_{\mathrm{mod}}$ performs well on the OOD Test data to obtain an upper bound on the possible performance. The $\beta$ we selected is $0.001$. The results are as follows (all numbers in %):
>
>   |             | FPR95 | AUROC |
>   |-------------|-------|-------|
>   | SVHN        | 25.02 | 94.90 |
>   | Textures    | 17.42 | 97.08 |
>   | Places365   | 41.24 | 91.16 |
>   | LSUN-Crop   | 7.35  | 98.67 |
>   | LSUN-Resize | 13.69 | 97.68 |
>   | iSUN        | 14.76 | 97.42 |
>   | **Mean**    | 19.91 | 96.15 |
>
>   For comparison, Hopfield Boosting obtains a mean FPR95 of 0.92%. This indicates that the innovations introduced in our work are the main factors for the effectiveness of Hopfield Boosting.
>
> - This question is best answered with an experiment that shows the effect of the memory size:
>
>   We train Hopfield Boosting on CIFAR-10 (ID data) and ImageNet (AUX data). During the weight update process, we store 50,000 patterns in the memories $\boldsymbol{X}$ and $\boldsymbol{O}$, and then ablate the number of patterns stored in the memories for computing the score $s(\boldsymbol{\xi})$ at inference time. We evaluate the discriminative power of $s(\boldsymbol{\xi})$ on SVHN with 1, 5, 10, 50, 100, 500, 1,000, 5,000, 10,000, and 50,000 patterns stored in the memories $\boldsymbol{X}$ and $\boldsymbol{O}$. To investigate the influence of the stochastic process of sampling $N$ patterns from the ID and AUX data sets, we conduct 50 runs for all of the $N$ and create boxplots of the runs.
>
>   The results (attached PDF) show that sampling $N=50,000$ patterns has the lowest variability of the individual trials. We argue that the reason for this is that by the time we reach $N=50,000$ the entire ID data set is stored in the Hopfield memory — which effectively eliminates stochasticity from randomly selecting $N$ patterns from the ID data.
>
>   These are interesting results and we thank the reviewer for bringing the point up. We will include the results in the Appendix of our final paper.
>
> - We see two notions of imbalance that could be relevant here:
>   1. Imbalance w.r.t. the number of samples contained in the ID and AUX data sets: The experiments we conducted in the manuscript show that Hopfield Boosting can handle this notion of imbalance. CIFAR-10 (the ID data set) contains 50,000 training samples, while ImageNet (the AUX data set) contains 1,281,167 training samples. Hopfield Boosting uses all training data from CIFAR-10, but for ImageNet, we sample 50,000 data instances to match the memory size of the ID data set. This is a simple way to ensure that the score $s(\boldsymbol{\xi})$ is not influenced by the number of data instances contained in the ID and AUX data sets.
>   2. Imbalance w.r.t. the number of samples stored in the Hopfield memories: This imbalance is indeed not tackled in the original manuscript, albeit it could be beneficial to store an imbalanced number of patterns in the memories $\boldsymbol{X}$ and $\boldsymbol{O}$. Hence, we do the following experiment to verify that we can use $s(\boldsymbol{\xi})$ when the number of patterns in $\boldsymbol{X}$ and $\boldsymbol{O}$ is imbalanced: We fill $\boldsymbol{X}$ with all 50,000 data instances of CIFAR-10 and fill $\boldsymbol{O}$ with 1, 5, 10, 50, 100, 500, 1000, 5000, 10,000, and 50,000 data instances. Then, we evaluate the discriminative power of $s(\boldsymbol{\xi})$ for the different instances. Our results (attached PDF) show that Hopfield Boosting is robust to an imbalance in the number of samples in $\boldsymbol{X}$ and $\boldsymbol{O}$. The setting with 50,000 samples in both memories (which is the setting we use in the experiments in our original manuscript) incurs the least variability.
> We will also include this experiment in the Appendix of the updated manuscript.
>
> **References:**
>
> Hendrycks et al., Deep Anomaly Detection with Outlier Exposure, ICLR 2019
>
> Zhang et al., Out-of-Distribution Detection based on In-Distribution Data Patterns Memorization with Modern Hopfield Energy, ICLR 2023

---

> > ### Comment · Reviewer_qRFa · 2024-08-09
> >
> > I would like to thank the authors for addressing my concerns. I have updated my score to 6 upon reading other reviews and the rebuttal.

---

### Author Rebuttal · Authors · 2024-08-07

We thank all reviewers for taking the time to provide high-quality feedback. It allowed us to significantly enhance our paper.
In summary, the reviewers appreciated the technical quality and the theoretical depth of our contribution. They also positively acknowledged that Hopfield Boosting obtains good results in OOD detection, achieving substantial improvements to existing methods on three benchmarks (including the large-scale ImageNet-1K benchmark). In terms of questions, the reviewers `DQmw`, `UgW1`, and `BR8F` were interested in the accuracy of the ID classification of Hopfield Boosting. We now report the ID accuracy for Hopfield Boosting and the other methods in our comparison in Table 1 in the document attached to this comment. In short, Hopfield Boosting achieves an ID classification accuracy of 94.02%, while a model trained without Hopfield Boosting obtains 94.80% (averaged over 5 independent training runs). Thanks to the feedback we received from the reviewers, we could also substantially strengthen the experimental evaluation of Hopfield Boosting: In summary, we conducted the following additional experiments:

- We investigate the tradeoff between ID classification accuracy and OOD detection performance of Hopfield Boosting (Figure 1): In the CIFAR-10 benchmark, we ablate on the hyperparameter $\lambda$, which controls the influence of our novel energy-based loss, $\mathcal{L}_{OOD}$, on the total loss. The experiment shows that decreasing $\lambda$ from 0.5 to 0.1 improves the classification error from 5.98% to 5.02%, while only slightly influencing the OOD detection results, increasing the mean FPR95 from 0.92% to 1.08%.

- To measure the contribution of the individual components of Hopfield Boosting (energy-based OOD loss, projection head, weighted sampling) on the total performance, we train models on CIFAR-10 by partially or completely removing said components. The results in Table 2 show that all components contribute considerably to the total performance of Hopfield Boosting, and the mean FPR95 metric worsens from 0.92% to 50.40% when eliminating all components.

- An additional toy experiment visualizes the training process of Hopfield Boosting while also considering the inlier classification task. Figure 4 shows that $\mathcal{L}_{CE}$ is able to separate the two classes well and that $E_b$ still forms a tight decision boundary around the ID data.

- We conduct experiments on how the number of samples stored in the Hopfield memories $\boldsymbol{X}$ and $\boldsymbol{O}$ at inference time influences the OOD detection result on CIFAR-10. Specifically: (a) we measure the performance with 1, 5, 10, 50, 100, 500, 1000, 5000, 10,000, and 50,000 samples in the memory of $\boldsymbol{X}$ and $\boldsymbol{O}$, respectively, and (b) keep the size of $\boldsymbol{X}$ constant at 50,000 while ablating on the number of samples in $\boldsymbol{O}$ (with the same sample sizes as in (a)). The results in Figures 2 and 3 show that Hopfield Boosting is rather robust to the number of samples in the memory, while the setting with 50,000 samples in both memories (which is the setting we use in the experiments in our original manuscript) incurs the least variability.

- We investigate a way to incorporate auxiliary outlier data in the OOD detection methods HE/SHE (Zhang et al., 2023). The results on CIFAR-10 show that extending the OOD detection score of HE, $s_\mathrm{HE}$, by a modern Hopfield energy term containing the auxiliary outlier data is insufficient to obtain OOD detection results that are competitive to Hopfield Boosting (see the response to reviewer `qRFa`).

- We show results of Hopfield Boosting trained on the ImageNet-1K benchmark when evaluated on noticeably different data sets: iCartoonFaces, RPC, and FourShapes (see the response to reviewer `BR8F`; we refer to Appendix H.6 of the original manuscript for examples from these data sets).

The Tables and Figures with the detailed results of the experiments are included in the PDF attached to this comment.

**References:**

Zhang et al., Out-of-Distribution Detection based on In-Distribution Data Patterns Memorization with Modern Hopfield Energy, ICLR 2023

---

### Decision · Program_Chairs · 2024-09-25

**Decision:**

Accept (poster)

**Comment:**

The authors introduce Hopfield Boosting, leveraging modern Hopfield energy to refine the decision boundaries effectively, which enhances the model's ability to differentiate between in-distribution (ID) and OOD data. The method's empirical success is evidenced by substantial improvements in OOD detection metrics across several benchmarks.

Nevertheless, concerns were raised about the paper’s novelty, the impact on ID classification accuracy, and the proposed method's theoretical underpinning. The authors have provided substantial rebuttals, improving the theoretical explanations and demonstrating robustness through additional experiments. The final decision leans towards acceptance, contingent on the authors addressing these remaining concerns in their final revision.